# CDC42 supports HBV entry by NTCP translocation to the plasma membrane and macropinocytosis

Shuzhi Cui[1,2,7], Wei Gao[2,3,4,7], Yuxin Chen[5,7], Yi Xu[1], Zhifang Li[2], Yu Wei [ID][4,6 ✉] & Yaming Jiu [ID][2,3,4 ✉]

## Abstract

**CDC42 is a member of Rho GTPase family that regulates various biological processes and its activity can be hijacked by invading pathogens. Here, we discovered that the level of active CDC42 in hepatocytes positively correlates with the entry capacity of hepatitis B virus (HBV). Mechanistically, CDC42 activation effectively promotes the transport of the viral receptor sodium taurocholate co-transporting polypeptide (NTCP) to the plasma membrane via Rab11 dependent recycling endosomal pathway. NTCP interacts with Rab11 and activation of CDC42 signaling reinforces the interaction between NTCP and Rab11. We further show that clathrin mediated endocytosis (CME), the known HBV entry pathway, is independent of CDC42 activity. Intriguingly, we reveal that CDC42 dependent macropinocytosis is a route for HBV entry, which is equally essential for viral infection as CME. Together, our findings uncover new mechanisms for HBV entry that involve unrecognized functions of CDC42 and suggest that Rho GTPase signaling might represent a potential target for antiviral therapy.**

**Keywords** CDC42; Hepatitis B Virus; Macropinocytosis; Sodium Taurocholate Cotransporting Polypeptide; Virus Entry
**Subject Categories** Membranes & Trafficking; Microbiology, Virology & Host Pathogen Interaction

## Introduction

CDC42 is a member of the Rho GTPase family that constitutes a group of cellular signaling molecules characterized by their capacity to switch between an active GTP-bound form and an inactive GDP-bound form (Coso et al, 1995). Active CDC42 is localized to cellular membranes, binds to its downstream effectors, having diverse effects on actin nucleation and polymerization (Bendezu et al, 2015; Watson et al, 2017). CDC42 plays crucial roles in many cellular processes, including cell polarity, cell survival, adhesion, migration, cell cycle progression, membrane trafficking and phagocytosis

(Coso et al, 1995; Etienne-Manneville and Hall, 2003; Mellman et al, 1999; Nobes and Hall, 1995). It is thus not surprising that CDC42 is involved in viral infection (Swaine and Dittmar, 2015). For instance, CDC42 promotes infections of human immunodeficiency virus type 1 (HIV-1) (Nikolic et al, 2011), oncolytic vaccinia virus (OVV) (Horita et al, 2019), dengue virus type-2 (DENV-2) (Zamudio-Meza et al, 2009) and α-herpesviruses (De Regge et al, 2006) by enhancing cell membrane fluidity. Ebola virus (EBOV), vaccinia virus, herpes simplex virus 1 (HSV-1) and respiratory syncytial virus (RSV) can hijack CDC42-regulated macropinocytosis, an actin-dependent extracellular fluid internalization pathway, for their entry into the host cell (Hetzenecker et al, 2016; Krzyzaniak et al, 2013; Mulherkar et al, 2011; Nanbo et al, 2010; Zhang et al, 2022). CDC42 promotes hepatitis E virus (HEV) infection by directly interacting with its capsid protein (Fan et al, 2021). While CDC42 is also implicated in protein transport and secretion (Choy et al, 1999; Hehnly et al, 2009; Ledoux et al, 2023), its role in these functions in the context of viral infections has not been fully elucidated. Therefore, understanding the explicit function of CDC42 in different viral infections remains an area of active research.

Hepatitis B virus (HBV) is an enveloped virus with a partially double-stranded DNA genome that infects human hepatocytes, causing acute and chronic hepatitis B (Razavi-Shearer et al, 2018; Summers et al, 1975). The identification of sodium taurocholate co-transporting polypeptide (NTCP) as the viral receptor has led to better understanding of the mechanisms for HBV entry (Yan et al, 2012). HBV envelop is composed of three viral surface proteins (HBsAg), namely large (L), middle (M) and small (S) proteins (Li et al, 2016). The preS1 domain of L protein is essential for the interaction with NTCP receptor (Yan et al, 2012). HBV entry into hepatocytes is initiated through attachment to cells via low-affinity interaction between the antigenic loop in the S domain of envelop proteins and heparan sulfate proteoglycans (HSPGs) (Schulze et al, 2007; Sureau and Salisse, 2013). Subsequently, high-affinity binding occurs between the myristoylated preS1 domain and NTCP on the plasma membrane (Asami et al, 2022), triggering virus internalization (Fukano et al, 2021; Konig and Glebe, 2017). Meanwhile, HBV-NTCP interaction induces endocytosis of NTCP (Fukano et al, 2021). NTCP is an important bile acid transporter localized to the basolateral membrane of hepatocytes. It has been shown that

[1]Guangzhou Institute of Pediatrics, Guangzhou Women and Children's Medical Center, Guangzhou Medical University, 510623 Guangzhou, China. [2]Unit of Cell Biology and Imaging Study of Pathogen Host Interaction, Key Laboratory of Molecular Virology and Immunology, Shanghai Institute of Immunity and Infection, Chinese Academy of Sciences, 200031 Shanghai, China. [3]Shanghai Institute of Materia Medica, Chinese Academy of Sciences, 201203 Shanghai, China. [4]University of Chinese Academy of Sciences, Yuquan Road No. 19(A), Shijingshan District, 100049 Beijing, China. [5]Department of Laboratory Medicine, Nanjing Drum Tower Hospital, the Affiliated Hospital of Nanjing University Medical School, 210008 Nanjing, China. [6]Institut Pasteur, Université Paris Cité, 28 rue du Dr. Roux, Paris 75015, France. [7]These authors contributed equally: Shuzhi Cui, Wei Gao, Yuxin Chen. ✉E-mail: yu.wei@pasteur.fr; ymjiu@siii.cas.cn

the transport from intracellular vesicular pool of internalized and newly synthesized NTCP to the cell surface is controlled by adenosine 3′,5′-cyclic monophosphate (cAMP) (Mukhopadhayay et al, 1997), protein kinase B (PKB) and protein kinase C (PKC) (Sarkar et al, 2006).

The endocytic pathways utilized by virus entry mainly include clathrin-mediated endocytosis (CME), caveolae-mediated endocytosis, and macropinocytosis (Mercer et al, 2010). Early studies have shown that CME (Herrscher et al, 2020; Huang et al, 2012) and caveolin-mediated entry are involved in HBV internalization (Macovei et al, 2010). Despite the important advances in the understanding of early steps in HBV infection, the host signaling regulating NTCP-dependent HBV entry remains incompletely understood.

In this study, we investigated CDC42 signaling in HBV infection and found that CDC42 is a key regulator in HBV entry. Mechanistically, we show that CDC42 signaling enhances NTCP trafficking to the plasma membrane and CDC42-dependent macropinocytosis constitutes a pathway to assist the penetration of HBV particles.

# Results

## CDC42 promotes HBV infection

To explore the functions of CDC42 in HBV infection, we used the previously established stable cell line HepG2-NTCP in which the viral receptor NTCP was introduced into HepG2 cells, conferring the susceptibility to HBV infection (Cui et al, 2024; Cui et al, 2023). We stably expressed Flag-tagged constitutively active (CA, G12V) and dominant-negative (DN, T17D) mutants of CDC42 into HepG2-NTCP cells. Neither the expression of CDC42-CA nor CDC42-DN affected growth dynamics of the cells (Fig. EV1A). We first analyzed the levels of active CDC42 by GTP-bound GTPase pull-down assay. As shown in Fig. 1A, expression of CDC42-CA significantly increased the level of CDC42 GTP-bound form in HepG2-NTCP cells, although similar expression of CDC42-CA and CDC42-DN proteins were observed. Notably, HepG2-NTCP cells expressing CDC42-DN showed lower levels of the GTP-bound CDC42 than vector-transfected cells, indicative of the dominant negative effects (Fig. 1A). We infected these cells with HBV for 8 h and examined viral replication at 14 days post-infection (dpi) by Southern blot. As shown in Fig. 1B, viral replication was substantially augmented by the activation of CDC42 signaling and repressed by the expression of CDC42-DN. Quantitative PCR (qPCR) of viral DNA from infected cells and secreted virions collected at 7 dpi further demonstrated positive effects of active CDC42 on viral replication (Fig. 1C). Upon HBV entry, the partially relaxed circular DNA genome (rcDNA) is converted into covalently closed circular DNA (cccDNA) that serves as template for viral transcription. To quantify cccDNA, we collected infected cells at 7 dpi. qPCR analysis showed that the levels of cccDNA were significantly increased in cells expressing CDC42-CA and decreased in cells expressing CDC42-DN (Fig. 1D). Quantification of secreted HBeAg by ELISA showed its increase in the medium from HepG2-NTCP CDC42-CA cells and the decrease from HepG2-NTCP CDC42-DN cells compared to control cells (Fig. 1E).

To confirm the functions of CDC42 in HBV infection, we treated HepG2-NTCP cells with either CDC42 activator bradykinin or inhibitor ML141. We tested cell viability with increasing concentrations of these two reagents and used 100 ng/ml bradykinin and 50 μM ML141 in the following experiments, which showed absence of toxicity to cells (Fig. EV1B,C). Treatment of HepG2-NTCP cells with bradykinin or ML141 for 10 h resulted in the increase or decrease of GTP-bound CDC42, respectively (Fig. 1F), confirming their opposite effects on CDC42 activity. To examine how long the drugs are effective, we treated HepG2-NTCP cells with bradykinin or ML141 for 10 h and measured the kinetics of CDC42 GTP-bound forms by immunoblotting. As shown in Fig. EV1D, the effects of bradykinin and ML141 on CDC42 signaling primarily lasted for 24 h. No difference in the levels of GTP-bound forms was observed between treated- and untreated-cells at 3 days post-treatment (Fig. EV1D), indicating temporary effects of these two reagents on CDC42.

We pre-treated HepG2-NTCP cells with either bradykinin or ML141 for 2 h and inoculated with HBV in the presence of bradykinin or ML141 for another 8 h. We collected cells at 14 dpi for Southern blot analysis and 7 dpi for qPCR and ELISA assays. In agreement with the results with CDC42 mutants, activation of CDC42 by the treatment with bradykinin in cells led to the elevated levels of intracellular HBV DNA, HBV DNA in secreted virions, cccDNA and secreted HBeAg, while treatment with ML141 resulted in opposite effects (Figs. 1G and EV1E–G).

Primary human hepatocytes (PHHs) are regarded as the benchmark model for in vitro studies of HBV replication. To consolidate the above observations, we treated PHHs with bradykinin or ML141 and infected treated PHHs with HBV. To exclude the artifacts of HBV entry linked to the use of polyethylene glycol (PEG) in infection, we used heparin-purified HBV stock to infect PHHs (see "Methods"). Treatment with the CDC42 activator bradykinin significantly augmented intracellular HBV DNA, HBV DNA in secreted virions, cccDNA and secreted HBeAg in PHHs, whereas treatment with the CDC42 inhibitor ML141 decreased viral replication (Fig. 1H–J). Together, these data indicate that CDC42 activation favors HBV infection.

## CDC42 is involved in the regulation of HBV entry

Since CDC42 is associated with the plasma membrane and is a key regulator of membrane morphology and endocytic transport (Ledoux et al, 2023; Mellman et al, 1999; Richnau et al, 2004), we hypothesized that CDC42 may play a role in HBV entry. To address this question, we first performed HBV attachment assay in HepG2-NTCP cells expressing CDC42-CA and CDC42-DN and in HepG2-NTCP cells treated with bradykinin and ML141. Heparin was used as a control reagent to block HBV attachment in HepG2-NTCP cells. Cells were infected with HBV and collected at 3 h post-infection (hpi) (Cui et al, 2023; Watashi et al, 2014). HBV DNA was subsequently measured by qPCR. The results showed no significant difference of HBV binding on the cell surface in these cells (Fig. 2A,B), suggesting that CDC42 has no effect on virus attachment.

Next, we assessed the effect of CDC42 on HBV internalization. HepG2-NTCP cells were infected with HBV and collected at 8 hpi. Myrcludex B, a well-known HBV entry inhibitor that blocks NTCP

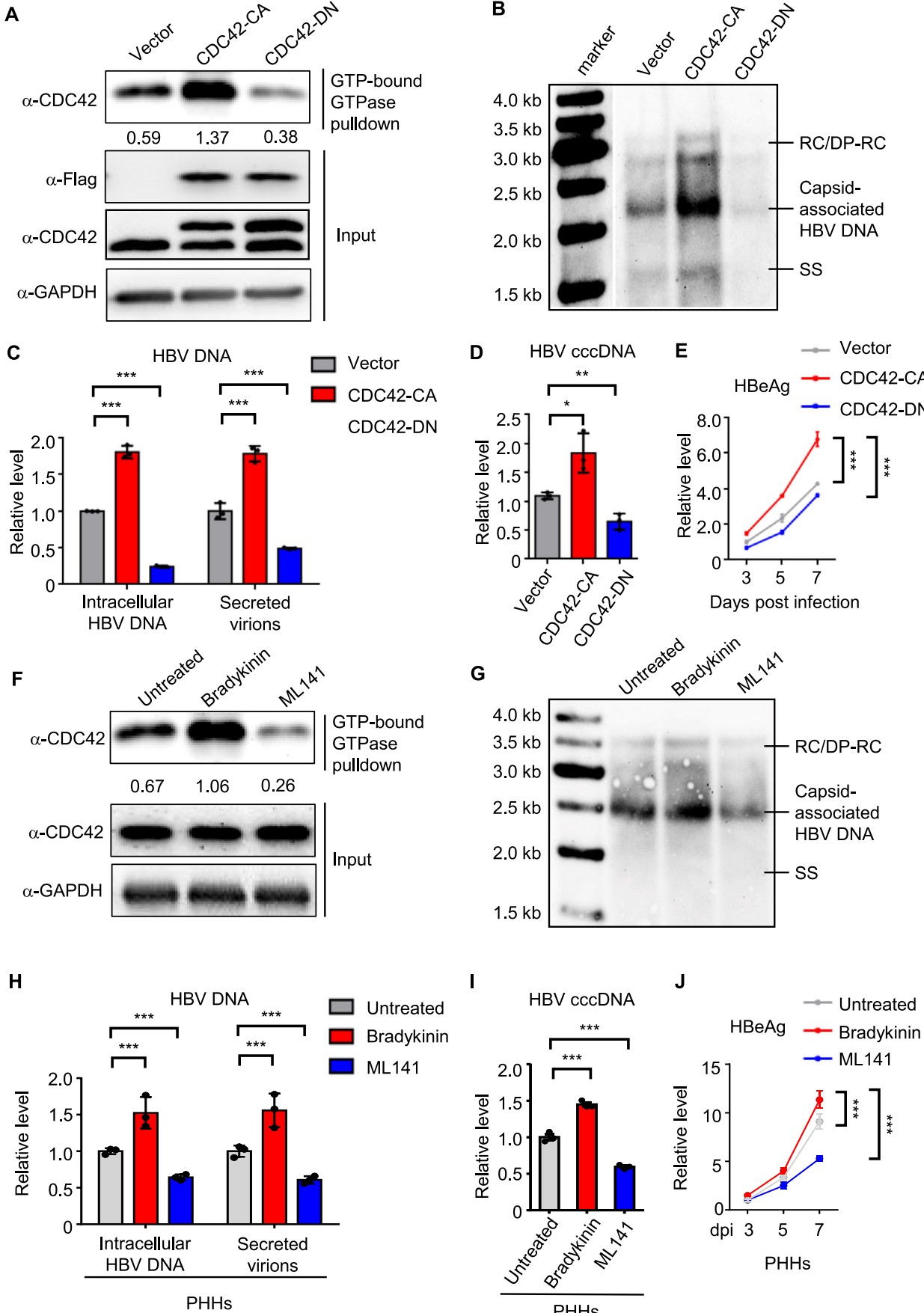

**Figure 1.  Activation of CDC42 promotes HBV infection.**

(A) HepG2-NTCP cells were stably transfected with vector (Vector), Flag-tagged constitutively active CDC42 (CDC42-CA) and dominant-negative CDC42 (CDC42-DN) plasmids. Detection of active CDC42 was assayed by GTP-bound GTPase pulldown and western blotting. GAPDH is used as loading control. The ratio of the GTP-bound active CDC42 versus GAPDH is presented. The experiment was replicated three times and one representative result is displayed. (B) Indicated cells were infected with HBV. Cytoplasmic encapsidated DNA was purified at 14 dpi and detected by Southern blot analysis using an HBV DNA probe. RC/DP-RC relaxed circular DNA, DP-RC deproteinized RC DNA, SS single-stranded (−) DNA. (C) Intracellular HBV DNA and secreted HBV virion DNA from HepG2-NTCP vector, CDC42-CA and CDC42-DN cells were purified at 7 dpi and measured by qPCR. Data are shown as fold changes to the vector group. $n = 3$. Intracellular HBV DNA of CDC42-CA and CDC42-DN: $P < 0.0001$; secreted virions of CDC42-CA: $P = 0.0008$; secreted virions of CDC42-DN: $P = 0.0013$. (D) cccDNA was purified at 7 dpi and measured by qPCR. Data are shown as fold changes to the vector group. $n = 3$. CDC42-CA: $P = 0.0213$; CDC42-DN: $P = 0.0069$. (E) Viral HBeAg in the medium released from infected cells was quantified by ELISA. Data are shown as fold changes to the Vector group at 3 dpi. $n = 3$. CDC42-CA at 7 days post infection: $P = 0.005$; CDC42-DN at 7 days post infection: $P = 0.009$. (F) HepG2-NTCP cells were treated with CDC42 activator bradykinin (100 ng/ml) or inhibitor ML141 (50 μM). Detection of active CDC42 was assayed by GTP-bound GTPase pulldown and western blotting. GAPDH is used as loading control. The ratio of the GTP-bound active CDC42 versus GAPDH is presented. The experiment was repeated three times and one representative result is displayed. (G) HepG2-NTCP cells treated with indicated reagents were infected with HBV. Cytoplasmic encapsidated DNA was purified at 14 dpi and detected by Southern blot analysis using an HBV DNA probe. RC/DP-RC relaxed circular DNA, DP-RC deproteinized RC DNA, SS single-stranded (−) DNA. (H–J) Primary human hepatocytes (PHHs) were treated with CDC42 activator bradykinin (100 ng/ml) or inhibitor ML141 (50 μM) and infected with HBV. Intracellular HBV DNA and secreted HBV virion DNA were purified at 7 dpi and measured by qPCR (H). cccDNA was purified at 3 dpi and measured by qPCR (I). Viral HBeAg in the medium released from infected cells at indicated time points was quantified by ELISA (J). Data are shown as fold changes to the untreated group at 3 dpi. $n = 3$. Intracellular HBV DNA of Bradykinin: $P = 0.0009$; Intracellular HBV DNA of ML141: $P = 0.0014$; secreted virions of Bradykinin: $P = 0.0006$; secreted virions of ML141: $P = 0.0007$; cccDNA of Bradykinin: $P = 0.0007$; cccDNA of ML141: $P = 0.0007$; HBeAg of Bradykinin at 7 days post infection: $P = 0.0005$; HBeAg of ML141 at 7 days post infection: $P = 0.0002$. These experiments were repeated three times. Data are represented as mean ± SEM. $^*P < 0.05$; $^{**}P < 0.01$; $^{***}P < 0.001$ (unpaired $t$ test). Source data are available online for this figure.

(Schulze et al, 2010), was tested for HBV entry inhibition in HepG2-NTCP cells with increasing concentrations and 200 nM Myrcludex B was used in this study (Fig. EV2A). Analysis of HBV DNA by qPCR showed that activation of CDC42 either by the expression of the constitutively active mutant or by bradykinin treatment significantly increased HBV internalization (Fig. 2C,D). Conversely, inactivation of CDC42 drastically inhibited virus entry (Fig. 2C,D). CDC42-DN and ML141 showed similar suppressive potency as Myrcludex B (Fig. 2C,D), suggesting a crucial role of CDC42 in HBV entry. We then employed PHHs for HBV internalization assay. PHHs were treated with bradykinin or ML141 and infected with HBV for 8 h. Quantification of intracellular HBV DNA by qPCR demonstrated the increase of HBV DNA in PHHs treated with bradykinin and its decrease in cells treated with ML141 (Fig. 2E). In addition, we used a synchronized infection protocol to validate these observations (Chakraborty et al, 2020). HepG2-NTCP cells expressing CDC42-CA or CDC42-DN were incubated with HBV for 1 h at 4 °C and then transferred to 37 °C for 2 h, 4 h or 6 h. Internalized HBV DNA was quantified by qPCR. We observed a time-dependent increase in HBV DNA and significant difference in viral DNA levels between HepG2-NTCP CDC42-CA or CDC42-DN and their parental cells (Fig. EV2B).

To further confirm these observations, we collected cells at 2, 4, 6 and 8 hpi, and performed immunofluorescence with anti-preS1 antibody for staining viral particles inside the cells. Cells expressing CDC42-CA or treated with bradykinin showed significantly higher numbers of preS1-positive signals than control cells, whereas abrogation of CDC42 activity by CDC42-DN or ML141 treatment resulted in decreased entry of viral particles, with the effect becoming significant at 4 hpi (Figs. 2F–I and EV2C). To further exclude artifacts in HBV entry assay, HepG2 cells were employed as negative control. As showed in Fig. 2J, HBV entry was solely dependent on the expression of NTCP in HepG2 cells and neither the treatment with bradykinin nor with ML141 did not significantly alter the intracellular HBV DNA levels in HepG2 cells.

While HepG2-NTCP CDC42-CA and CDC42-DN cells stably express CDC42 mutants, which may affect post-entry steps in viral replication, treatment of HepG2-NTCP cells with CDC42 activator bradykinin or inhibitor ML141 is temporary (Fig. EV1D), suggesting that CDC42 signaling mainly regulates early steps in HBV infection. To assess the effects of CDC42 signaling on post-entry steps of viral life cycle, we first infected HepG2-NTCP cells with HBV, washed cells to eliminate viral particles and treated infected cells with bradykinin or ML141. Cells were collected either at 3 dpi for cccDNA analysis or at 5 dpi for pgRNA analysis. No significant differences in the levels of cccDNA or pgRNA were observed between treated- and untreated-cells (Fig. 3A,B). We then used cell models for HBV replication bypassing the entry step, namely HepAD38 cells and HepG2 cells transfected with plasmid containing 1.3-mer HBV genome (pHBV1.3). HepAD38 cells were treated with bradykinin or ML141. Intracellular encapsidated viral DNA was prepared for qPCR analysis, which excluded integrated HBV DNA (see Methods). No difference in the levels of encapsidated viral DNA was observed between treated- and untreated-samples (Fig. 3C). Similarly, HepG2 cells transfected with pHBV1.3 and treated with bradykinin or ML141 showed comparable levels of intracellular encapsidated viral DNA (Fig. 3D). Furthermore, we performed immunostaining of HBcAg at 7 dpi in infected HepG2-NTCP vector, CDC42-CA and CDC42-DN cells. Similar frequency of HBcAg-positive cells was detected among the three cell lines, although the fluorescence intensity was more pronounced in CDC42-CA cells and weaker in CDC42-DN cells compared to the parental cell line (Fig. 3E), which suggests that CDC42 may influence the number of viral particles entering each cell rather than the number of cells susceptible to HBV.

Together, these data indicate that CDC42 primarily promotes HBV entry via NTCP receptor for effective viral infection.

## CDC42 fosters the recruitment of NTCP to the cell surface

To investigate the mechanisms of the regulation of HBV entry by CDC42, we focused on the potential roles of CDC42 in NTCP expression and its cellular localization. First, we examined NTCP mRNA in HepG2-NTCP vector, CDC42-CA and CDC42-DN cells

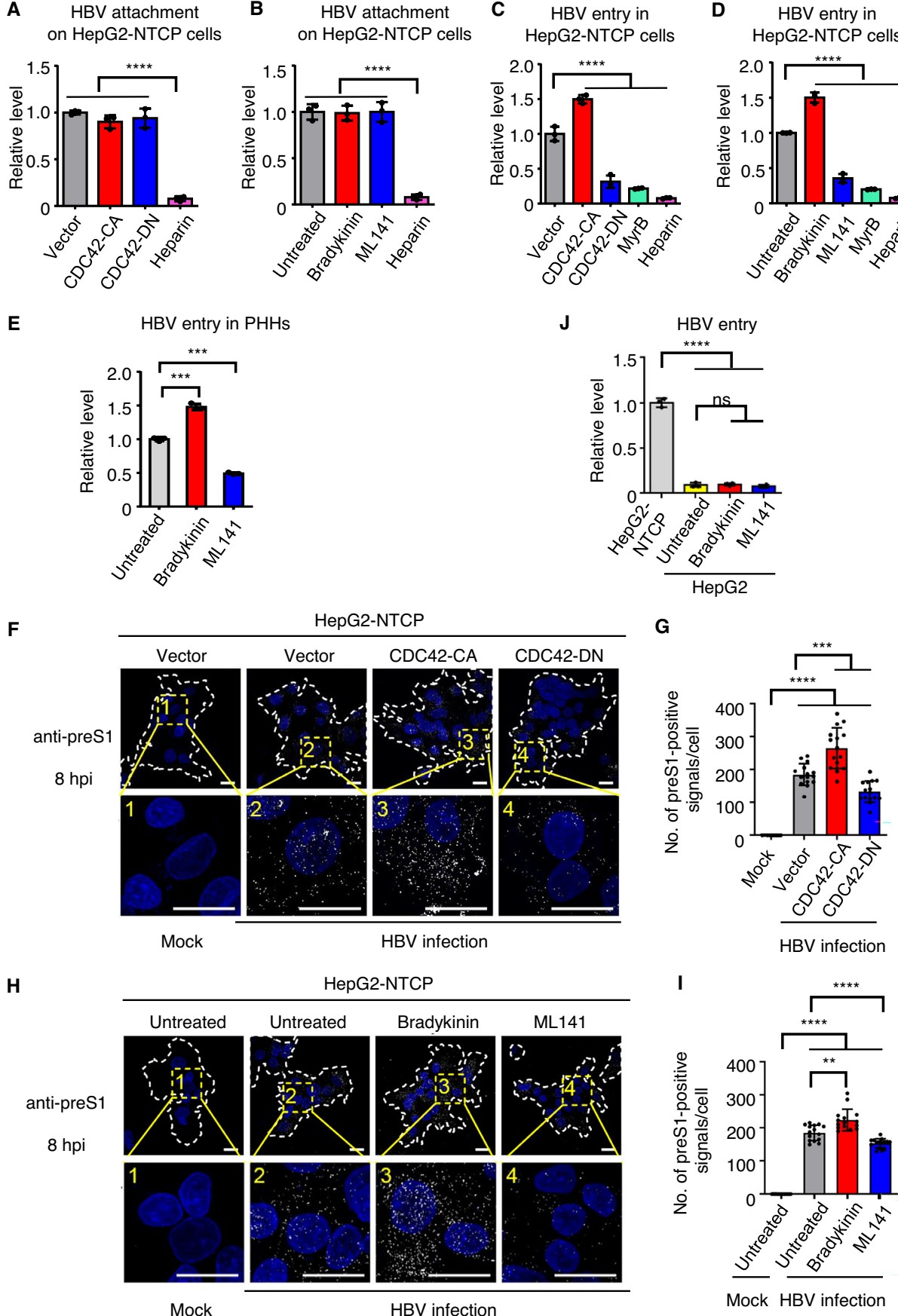

© The Author(s)

◄ **Figure 2. CDC42 signaling promotes HBV entry.**

(A) Quantification of HBV DNA in HepG2-NTCP vector, CDC42-CA and CDC42-DN cells at 3 hpi by qPCR. Data are shown as fold changes to the vector group. Treatment with heparin blocked HBV attachment. $n = 3$. Vector, CDC42-CA and CDC42-DN: $P < 0.0001$. (B) Quantification of HBV DNA in HepG2-NTCP cells treated with indicated reagents at 3 hpi by qPCR. $n = 3$. Untreated, Bradykinin and ML141: $P < 0.0001$. (C) Quantification of internalized HBV DNA in HepG2-NTCP vector, CDC42-CA, CDC42-DN cells and HepG2-NTCP cells treated with 200 nM Myrcludex B (MyrB) at 8 hpi by qPCR. Data are shown as fold changes to the vector group. $n = 3$. CDC42-CA, CDC42-DN, MyrB and heparin: $P < 0.0001$. (D) Quantification of internalized HBV DNA in HepG2-NTCP untreated or treated with 100 ng/ml bradykinin, 50 µM ML141 or 200 nM MyrB by qPCR. Data are shown as fold changes to the untreated group. $n = 3$. Bradykinin, ML141, MyrB and heparin: $P < 0.0001$. (E) Primary human hepatocytes (PHHs) were treated with bradykinin (100 ng/ml) or ML141 (50 µM) and infected with HBV. Intracellular HBV DNA was quantified by qPCR at 8 hpi. $n = 3$. Bradykinin: $P = 0.0001$; ML141: $P = 0.0001$. (F) Representative immunofluorescence images of HepG2-NTCP uninfected cells (Mock), HepG2-NTCP vector, CDC42-CA and CDC42-DN cells infected with HBV at 8 hpi. HBV particles were stained with anti-preS1 antibody. Nuclei were stained with DAPI. The cell edges are outlined by white dotted lines. Scale bar = 10 µm. (G) Quantification of preS1-positive signals per cell in (F). $n = 15$ views (100×/1.5 oil objective). Vector, CDC42-CA and CDC42-DN versus mock: $P < 0.0001$; CDC42-CA versus vector: $P = 0.0002$; CDC42-DN versus vector: $P = 0.0003$. (H) Representative immunofluorescence images of HepG2-NTCP uninfected cells (Mock) and infected HepG2-NTCP cells treated as indicated at 8 hpi. HBV particles were stained with anti-preS1 antibody. Nuclei were stained with DAPI. The cell edges are outlined by a white dotted line. Scale bar = 10 µm. (I) Quantification of preS1-positive signals per cell in (H). $n = 15$ views (100×/1.5 oil objective). Untreated, Bradykinin and ML141 versus mock: $P < 0.0001$; Bradykinin versus untreated: $P = 0.0024$; ML141 versus untreated: $P < 0.0001$. (J) Quantification of internalized HBV DNA in HepG2-NTCP cells and HepG2 cells treated as indicated at 8 hpi. Data are shown as fold changes to the untreated group. $n = 3$. HepG2-NTCP versus HepG2: $P < 0.0001$; Bradykinin versus untreated in HepG2: $P = 0.7329$; ML141 versus untreated in HepG2: $P = 0.4328$; These experiments were repeated three times. Data are represented as mean ± SEM. ns, no significant difference; $**P < 0.01$; $***P < 0.001$; $****P < 0.0001$ (unpaired $t$ test). Source data are available online for this figure.

and found that NTCP transcription levels were similar regardless of CDC42 activity (Fig. EV3A), which is in agreement with the fact that the regulation of NTCP mRNA is mainly achieved by bile acids (Appelman et al, 2021).

We then analyzed NTCP protein expression by immunoblotting. Total NTCP protein was expressed at similar levels in HepG2-NTCP control cells and CDC42 mutants-expressing cells (Fig. 4A). However, when plasma membrane (PM) proteins were isolated from these cells and analyzed for NTCP expression, we found an enrichment of NTCP on the plasma membrane in CDC42-CA-expressing cells and its attenuation in CDC42-DN-expressing cells (Fig. 4A). The membrane protein transferrin receptor (TfR) was used as control for PM protein expression, which was not influenced by CDC42 (Fig. 4A). Immunoblotting analysis of PM proteins from HepG2 cells, which barely express NTCP, showed negative signals with anti-NTCP antibody, but positive signals with anti-TfR antibody (Fig. EV3B), validating the specificity of the detection.

We further measured the cell surface-associated NTCP by immunostaining. Since intracellular signal interference occurred when using anti-NTCP antibody for immunofluorescence labeling, we established HepG2-NTCP-EGFP line by stably expressing NTCP fused with enhanced green fluorescent protein (EGFP) in HepG2 cells. We confirmed the expression of the fusion protein NTCP-EGFP by immunoblotting (Fig. EV3C). Previous report demonstrated that NTCP-EGFP fusion protein retains bile acid transport activity comparable to that of NTCP (Stross et al, 2010). To test whether EGFP would influence NTCP function as HBV receptor, we compared HBeAg expression between HepG2-NTCP cells and HepG2-NTCP-EGFP cells infected with HBV. The levels of HBeAg at 7 dpi from HepG2-NTCP-EGFP cells were slightly lower than those from HepG2-NTCP cells (Fig. EV3D), indicating that fusion with EGFP has limited negative effects on the function of NTCP as HBV receptor and HepG2-NTCP-EGFP cells can be used for HBV infection.

We then constructed stable cell lines expressing CDC42-CA and CDC42-DN in HepG2-NTCP-EGFP cells. We imaged with Z axis scanning and calculated the immunofluorescent intensity of EGFP from the orthogonal view of HepG2-NTCP-EGFP vector control, CDC42-CA and CDC42-DN cells, which represents the expression

level of NTCP on the plasma membrane. We observed a significant increase in PM-NTCP signals in HepG2-NTCP-EGFP CDC42-CA cells and a decrease in HepG2-NTCP-EGFP CDC42-DN cells (Fig. 4B). We further used the XZ orthogonal data to analyze the averaged fluorescence intensity distribution of PM-NTCP in these cells. Cells expressing CDC42-CA showed higher PM-NTCP fluorescent intensity along cell contour than control cells, while cells expressing CDC42-DN showed much lower intensity than control cells (Fig. 4B). Consistently, HepG2-NTCP-EGFP cells treated with bradykinin had high levels of PM-NTCP and these cells treated with ML141 showed low levels of PM-NTCP, as demonstrated by immunoblotting and immunofluorescence assays (Fig. 4C,D).

We tested whether CDC42 influences taurocholate (TC) bile acid uptake activity of NTCP. Stable expression of CDC42-CA in HepG2-NTCP cells significantly enhanced TC uptake, contrasting to the cells expressing CDC42-DN in which TC uptake was inhibited (Fig. 4E). These findings are consistent with the observations that CDC42 markedly upregulates NTCP membrane expression. It is known that CDC42 plays a role in the regulation of cell polarity. To assess the effects of CDC42 on cell polarity of HepG2-NTCP cells, we stained MRP2, a bile acid transporter that is expressed on the apical membrane of hepatocytes, in HepG2-NTCP vector, CDC42-CA and CDC42-DN cells. We found that activation of CDC42 increased MRP2 expression, whereas its inactivation decreased the number of MRP2-positive signals (Fig. EV3E), confirming the role of CDC42 signaling in cell polarity. NTCP is located on the basolateral membrane of hepatocytes in the liver. To address the question of whether CDC42 signaling impact NTCP distribution on HepG2-NTCP cells, we quantified NTCP-EGFP intensity on the apical and basal surfaces of HepG2-NTCP-EGFP vector, CDC42-CA and CDC-42-DN cells and calculated the ratio of NTCP-EGFP intensity at the basal surface to that at the apical surface. No difference was observed in the distribution of NTCP between these cell types (Fig. EV3F).

Previous report showed that HBV infection down-regulated its receptor in infected cells (Breiner et al, 2001). To examine HBV infection on NTCP expression in the context of CDC42 activation or inactivation, we analyzed total NTCP expression in HepG2-NTCP vector, CDC42-CA and CDC42-DN cell lines infected or

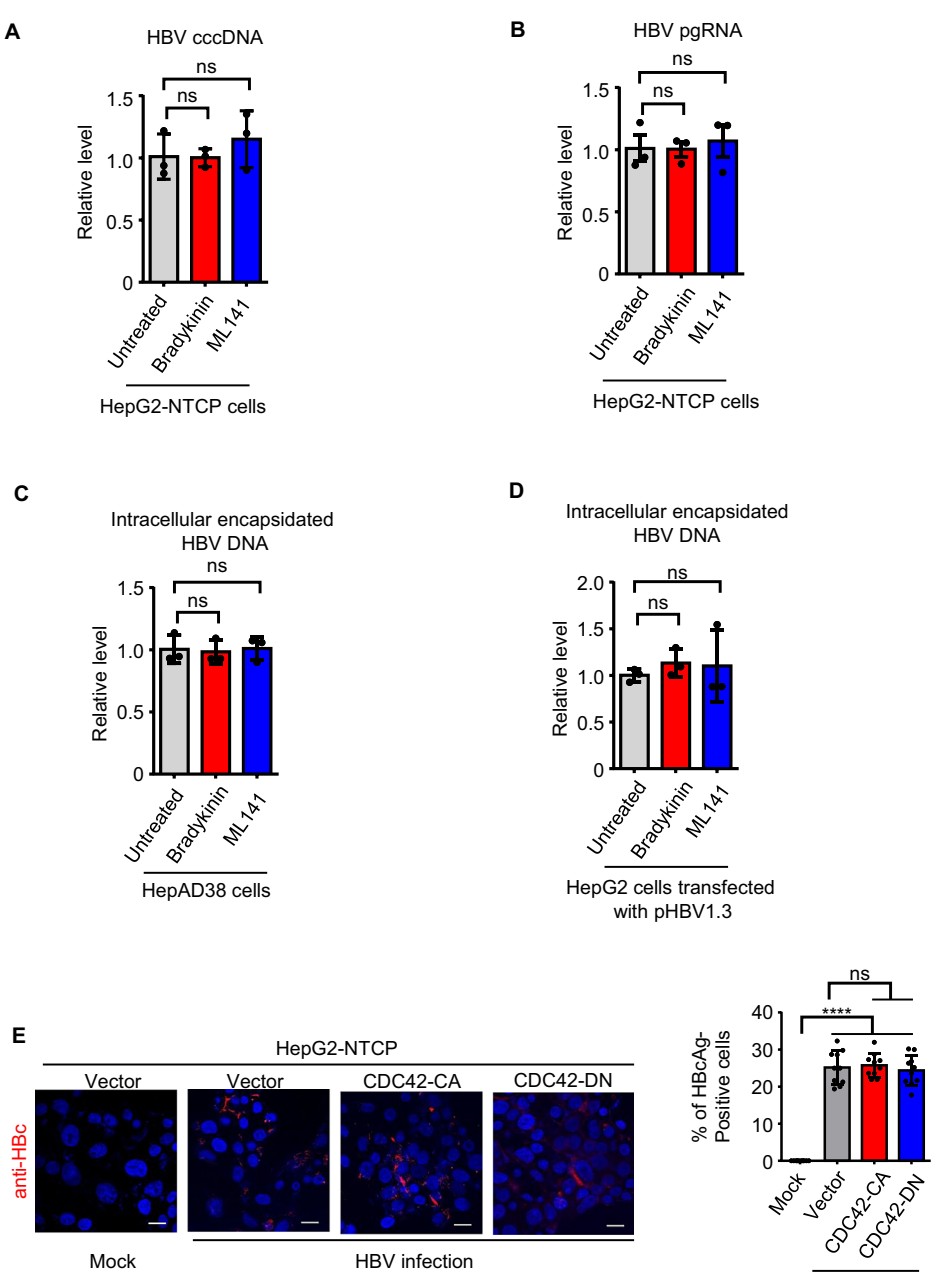

**Figure 3. CDC42 has no effect on late steps of HBV replication.**

(A, B) HepG2-NTCP cells were first infected with HBV for 8 h, then treated with bradykinin (100 ng/ml) or ML141 (50 μM) for 8 h. Cells were collected at 3 dpi for cccDNA analysis (A) or at 5 dpi for pgRNA analysis (B) by qPCR. Data are shown as fold changes to the untreated group. n = 3. (C) HepAD38 cells were first treated with bradykinin (100 ng/ml) or ML141 (50 μM) for 2 h in the presence of doxycycline, then in the absence of doxycycline for 8 h. Cells were collected 7 days later for the preparation of encapsidated viral DNA. Encapsidated viral DNA was quantified by qPCR. Data are shown as fold changes to the untreated group. n = 3. (D) HepG2 cells were treated with bradykinin (100 ng/ml) or ML141 (50 μM) for 2 h. Cells were transfected with the plasmid pHBV1.3, followed by the treatment with bradykinin or ML141 for 8 h. Cells were collected at 48 h post-transfection for the preparation of encapsidated viral DNA. Encapsidated viral DNA was quantified by qPCR. Data are shown as fold changes to the untreated group. n = 3. (E) HepG2-NTCP cells were infected with HBV. Cells were stained with anti-HBc antibody at 7 dpi. Quantification of HBcAg-positive cells is presented. Scale bar = 10 μm. n = 10 views (100×/1.5 oil objective). Vector, CDC42-CA and CDC42-DN versus mock: P < 0.0001. Data are represented as mean ± SEM. ns no significant difference; ****P < 0.0001 (unpaired t test). Source data are available online for this figure.

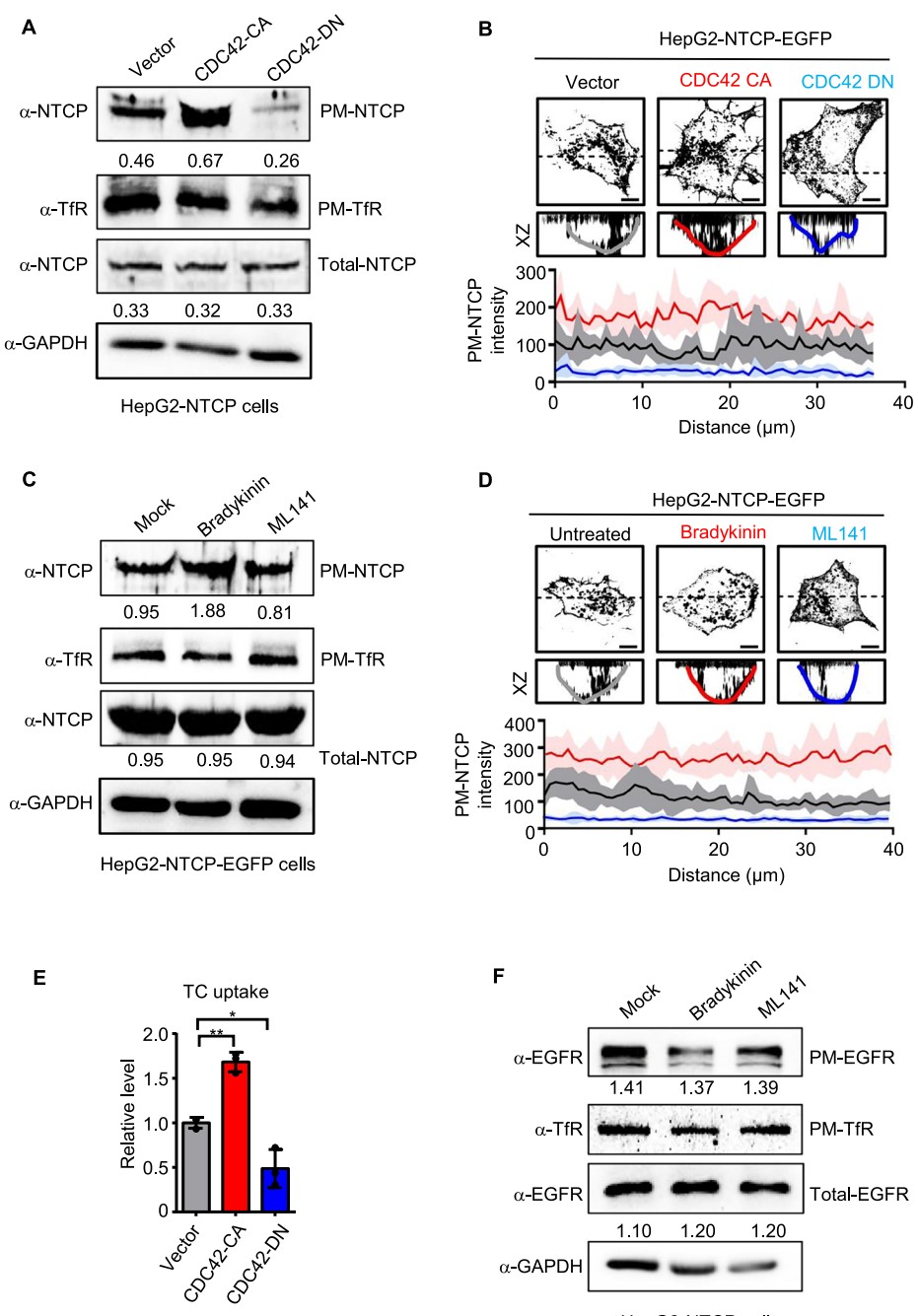

uninfected with HBV by western blot. Consistent with the previous report (Breiner et al, 2001), HBV infection reduced total NTCP expression, regardless of CDC42 status (Fig. EV3G). These results further confirm that CDC42 signaling mainly regulates NTCP expression at the plasma membrane.

Epidermal growth factor receptor (EGFR) has been shown as a co-receptor for HBV infection (Iwamoto et al, 2019). We investigated whether CDC42 had influence on the membrane localization of EGFR. In contrast to NTCP, treatment of HepG2-NTCP cells with bradykinin and ML141 had no significant effect on EGFR plasma membrane expression (Fig. 4F). Together, we

conclude that CDC42 fosters specifically the recruitment of NTCP to the plasma membrane.

## CDC42 positively regulates recycling endosomal-dependent NTCP transportation

Next, we investigated the mechanisms of CDC42-mediated NTCP translocation on the plasma membrane. It has been reported that intracellular NTCP co-localizes with the recycling endosome marker Rab11 (Sarkar et al, 2006), suggesting that recycling endosomes may represent a route of internalized NTCP traffic to

**Figure 4. CDC42 enhances the recruitment of NTCP on the cell surface.**

(A) Plasma membrane (PM) proteins were purified from HepG2-NTCP vector, CDC42-CA and CDC42-DN cells. PM and total proteins were subjected to western blotting analysis with indicated antibodies. The plasma membrane protein transferrin receptor (PM-TfR) and GAPDH were used as loading control of plasma membrane proteins and total proteins, respectively. The ratio of the PM-NTCP versus PM-TfR and total NTCP versus GAPDH are presented. (B) Orthogonal views show NTCP distribution in HepG2-NTCP-EGFP vector, CDC42-CA and CDC42-DN cells. An XZ orthogonal image was established at the thickest portion of the cells at z axis (dot lines). Line profiles of the fluorescence intensity of PM-NTCP-EGFP correspond to the color lines in the bottom panel of orthogonal views (XZ panel). The start point (0 μm) corresponds to the left start point of the lines. Scale bar = 10 μm. $n = 7$. (C) Plasma membrane proteins were purified from HepG2-NTCP cells treated with bradykinin (100 ng/ml) or ML141 (50 μM). PM and total proteins were subjected to western blotting analysis with indicated antibodies. PM-TfR and GAPDH were used as loading control of plasma membrane proteins and total proteins, respectively. The ratio of the PM-NTCP versus PM-TfR and total NTCP versus GAPDH are presented. (D) Orthogonal views show NTCP distribution in HepG2-NTCP-EGFP treated with bradykinin (100 ng/ml) or ML141 (50 μM). An XZ orthogonal image was established at the thickest portion of the cells at z axis (dot lines). Line profiles of the fluorescence intensity of PM-NTCP-EGFP correspond to the color lines in the bottom panel of orthogonal views (XZ panel). The start point corresponds to the left start point of the lines. Scale bar = 10 μm. $n = 11$. (E) Taurocholate (TC) uptake assay in HepG2-NTCP vector, CDC42-CA and CDC42-DN cells by ELISA. Data are shown as fold changes to the vector group. $n = 3$. CDC42-CA: $P = 0.0071$; CDC42-DN: $P = 0.0161$. (F) Plasma membrane proteins were purified from HepG2-NTCP cells treated with bradykinin (100 ng/ml) or ML141 (50 μM). Plasma membrane EGFR (PM-EGFR) was detected by western blotting analysis. PM-TfR and GAPDH were used as loading control of plasma membrane proteins and total proteins, respectively. The ratio of the PM-EGFR versus PM-TfR and total EGFR versus GAPDH are presented. Data are represented as mean ± SEM. These experiments were repeated three times. *$P < 0.05$; **$P < 0.01$ (unpaired *t* test). Source data are available online for this figure.

the plasma membrane. To assess the impact of recycling endosomes on the transport of NTCP and HBV infection, we depleted Rab11 in HepG2-NTCP cells using short hairpin RNA (shRNA) targeting Rab11 and confirmed the downregulation of Rab11 by immuno-blotting (Fig. 5A). We observed that Rab11 knockdown specifically reduced PM-NTCP but not PM-TfR in cells, while the levels of total NTCP was not affected (Fig. 5A). By qPCR, we confirmed that depletion of Rab11 had no effect on NTCP transcription (Fig. EV4A). We depleted Rab11 in HepG2-NTCP-EGFP cells and performed immunofluorescence assay. Orthogonal views showed that the fluorescence intensity of PM-NTCP in Rab11 knockdown cells was significantly decreased (Fig. 5B), which led to the decrease of HBV entry (Fig. 5C). Consequently, deficiency of Rab11 in HepG2-NTCP cells drastically diminished HBV replication (Fig. 5D) and HBeAg secretion (Fig. 5E). These results provide evidence that Rab11 participates in NTCP transport to the cell membrane.

To investigate the role of CDC42 in the recycling endosomal pathway, we expressed mCherry-tagged Rab11 (Rab11-mCherry) in HepG2-NTCP-EGFP cells and performed live-cell imaging. NTCP carried by mCherry-positive recycling endosomes gradually fused with plasma membrane (Fig. 5F), indicating that NTCP can be recycled to the plasma membrane by recycling endocytosis. Immunofluorescence assays in HepG2-NTCP-EGFP CDC42-CA and CDC42-DN cells showed that activation of CDC42 increased co-localization of NTCP with Rab11 and suppression of CDC42 decreased their co-localization (Fig. 5G). In contrast, co-localization of NTCP with the late endosome/lysosome marker Lamp2 was at lower levels in CDC42-CA cells, but at higher levels in CDC42-DN cells (Fig. EV4B), suggesting that CDC42 signaling may alienate NTCP from late endosome/lysosome compartments to favor NTCP plasma membrane localization. It is noted that CDC42 did not influence Rab11 and Lamp2 expression (Fig. EV4C). To elucidate the relationship between NTCP, Rab11, and CDC42, we performed immunoprecipitation assays. We generated HepG2-NTCP cells stably expressing wild-type CDC42 (CDC42-WT) as control. Cellular extract from HepG2-NTCP CDC42-WT, CDC42-CA and CDC42-DN cells was immunoprecipitated with anti-Rab11 antibody and immune complexes were analyzed by western blotting with anti-NTCP and anti-CDC42 antibodies. We found that Rab11 formed protein complex with NTCP but not with CDC42 (Fig. 5H).

Indeed, immunoprecipitation of CDC42 failed to precipitate neither NTCP nor Rab11, confirming that CDC42 was absent in NTCP-Rab11 complex (Fig. 5I). Notably, we observed a significant increase of the interaction between NTCP and Rab11 in HepG2-NTCP CDC42-CA cells compared to HepG2-NTCP CDC42-WT cells and a decrease in HepG2-NTCP CDC42-DN cells (see Fig. 5H), indicating that CDC42 signaling modulates NTCP-Rab11 interaction. We treated HepG2-NTCP shRab11 cells with brady-kinin and ML141 and prepared plasma membrane proteins. Immunoblotting analysis showed that PM-NTCP levels were reduced in Rab11 knockdown cells (Fig. 5J). Furthermore, when Rab11 was knocked down, NTCP on the plasma membrane was much less sensitive to the treatment with bradykinin and ML141 (Fig. 5J), suggesting that CDC42 acts upstream of Rab11 to regulate NTCP translocation to the membrane. In conclusion, our data indicate that CDC42 facilitates the expression of NTCP on the plasma membrane by modulating Rab11-dependent recycling endosomal trafficking of NTCP.

## The enhancement of HBV internalization by CDC42 is independent of clathrin-mediated endocytosis

It has been shown that CME is an important pathway for HBV entry (Herrscher et al, 2020; Huang et al, 2012; Umetsu et al, 2018). To investigate the relationship between CDC42 and CME, we examined the CME activity in HepG2-NTCP CDC42-CA and CDC42-DN cells by testing the uptake of fluorescein isothiocyanate (FITC)-labelled transferrin, which is mediated through constitutive endocytosis via a clathrin-dependent pathway (Daniels et al, 2006). There was no significant difference in CME activity among HepG2-NTCP control, CDC42-CA and CDC42-DN cells (Fig. 6A,B). Similarly, treatment of HepG2-NTCP cells with bradykinin and ML141 did not induce any change in CME capability (Fig. 6C,D). These results indicate that CME activity is not regulated by CDC42.

We then examined the relationship between CDC42 and HBV-related CME. We employed CME inhibitors pitstop2 and dynasore (Kirchhausen et al, 2008; von Kleist et al, 2011) to treat HepG2-NTCP cells and infected the cells with HBV (Fig. EV4D,E). Attachment assay showed that CME inhibitors did not have any effect on HBV binding to the cell surface (Fig. EV4F). In contrast, the treatment with CME inhibitors reduced the level of internalized

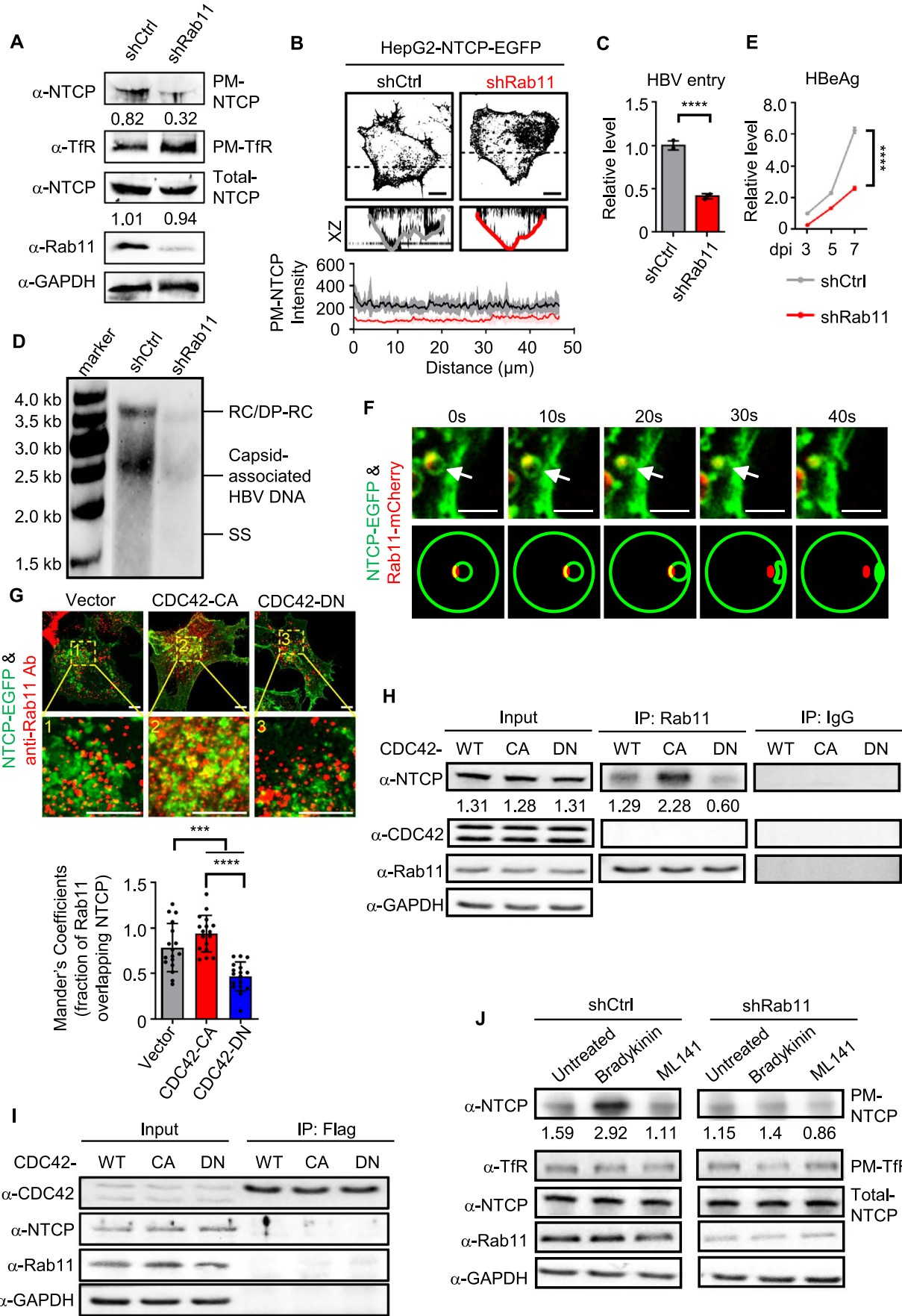

**Figure 5.  CDC42 signaling increases recycling endosomal-dependent NTCP translocation to the plasma membrane.**

(A) Plasma membrane proteins were purified from HepG2-NTCP control (shCtrl) and Rab11 knockdown cells (shRab11). PM and total proteins were subjected to western blotting analysis with indicated antibodies. PM-TfR and GAPDH were used as loading control of plasma membrane proteins and total proteins, respectively. The ratio of the PM-NTCP versus PM-TfR and total NTCP versus GAPDH are presented. (B) Orthogonal views show NTCP distribution in HepG2-NTCP-EGFP shCtrl and shRab11 cells. An XZ orthogonal image was established at the thickest portion of the cells at z-axis (dot lines). Line profiles of the fluorescence intensity of PM-NTCP-EGFP correspond to the color lines in the bottom panel of orthogonal views (XZ panel). The start point corresponds to the left start point of the lines. Scale bar = 10 μm. $n = 11$. (C) Quantification of internalized HBV DNA in HepG2-NTCP shCtrl and shRab11 cells by qPCR at 8 hpi. Data are shown as fold changes to the shCtrl group. $n = 3$. $P < 0.0001$. (D) HepG2-NTCP shCtrl and shRab11 cells were infected with HBV. Cytoplasmic encapsidated DNA was purified at 14 dpi and detected by Southern blot analysis using an HBV DNA probe. RC/DP-RC: relaxed circular DNA, DP-RC: deproteinized RC DNA; SS: single-stranded (−) DNA. (E) Viral HBeAg in the medium released from infected cells was quantified by ELISA. Data are shown as fold changes to the shCtrl group at 3 dpi. $n = 3$. Seven days post infection: $P < 0.0001$. (F) Translocation of NTCP-positive recycling endosomes to the plasma membrane. Top panel: representative immunofluorescence images of NTCP-positive vesicle transport. Bottom panel: schematic diagram of NTCP-positive vesicle transport. The white arrow indicates a representative vesicle. (G) Representative immunofluorescence images of NTCP-EGFP co-localization with recycling endosomes in HepG2-NTCP vector, CDC42-CA and CDC42-DN cells. The yellow box indicates the magnified area shown in the corresponding bottom panel. Recycling endosomes were stained with anti-Rab11 antibody. Quantification of Mander's coefficients of NTCP and recycling endosomes. $n = 20$ views (100×/1.5 oil objective). Scale bar = 10 μm. CDC42-CA versus vector: $P = 0.0009$; CDC42-DN versus vector: $P = 0.0005$; CDC42-CA versus CDC42-DN: $P < 0.0001$. (H) Co-immunoprecipitation assay with anti-Rab11 antibody and control IgG in HepG2-NTCP CDC42-WT, CDC42-CA and CDC42-DN cells. Immunoprecipitates were detected by indicated antibodies. The ratio of NTCP versus Rab11 is presented. (I) Co-immunoprecipitation assay with anti-Flag antibody in HepG2-NTCP CDC42-WT, CDC42-CA and CDC42-DN cells. Flag-coprecipitated proteins were detected with indicated antibodies. (J) Plasma membrane proteins were purified from HepG2-NTCP shCtrl and shRab11 cells treated with bradykinin (100 ng/ml) or ML141 (50 μM). PM and total proteins were subjected to western blotting analysis with indicated antibodies. PM-TfR and GAPDH were used as loading control of plasma membrane proteins and total proteins, respectively. The ratio of the PM-NTCP versus PM-TfR is presented. These experiments were repeated three times. Data are represented as mean ± SEM. ***$P < 0.001$; ****$P < 0.0001$ (unpaired $t$ test). Source data are available online for this figure.

HBV DNA (Fig. 6E), which is in accordance with previous reports (Herrscher et al, 2020; Huang et al, 2012; Umetsu et al, 2018).

We next treated HepG2-NTCP CDC42-DN cells with pitstop2 and dynasore and evaluated HBV internalization at 8 hpi. Analysis of HBV DNA by qPCR showed that CME inhibitors further inhibited HBV internalization (Fig. 6E) and decreased the levels of HBeAg in CDC42-DN cells (Fig. 6F). These results suggest that HBV-related CME is independent of CDC42 signaling. To evaluate whether the function of CDC42 on HBV relies on CME, we inhibited CME in HepG2-NTCP cells by treating them with pitstop2 or dynasore, inactivated CDC42 with the addition of ML141 and then infected the cells with HBV. The CDC42 inhibitor was capable to further block HBV internalization and viral replication in the context of CME suppression (Fig. 6G,H). These results suggest that CDC42-mediated HBV internalization and clathrin-mediated endocytic uptake of HBV may represent independent pathways for HBV entry.

## Macropinocytosis-mediated HBV internalization is a CDC42-dependent process

CDC42 plays an important role in macropinocytosis, which leads to the engulfment of fluid, solute and membrane under normal physiological conditions (Koivusalo et al, 2010). To examine whether CDC42 regulates HBV internalization through the macropinocytosis pathway, we first wanted to confirm that macropinocytosis is dependent on CDC42 activity. We used the well-known marker for macropinocytosis the high-molecular-weight dextran (70 kDa or higher) (Le AH and Machesky, 2022) to measure macropinocytosis capacity of HepG2-NTCP control, CDC42-CA and CDC42-DN cells, as well as HepG2-NTCP cells treated with bradykinin and ML141. Quantification of the fluorescence signals of dextran confirmed that the activity of macropinocytosis in cells correlated with the activity of CDC42 (Fig. EV5A–D).

To explore the role of macropinocytosis in HBV entry, we treated HepG2-NTCP cells with two macropinocytosis inhibitors,

wortmannin and 5-(N-Ethyl-N-isopropyl)-Amiloride (EIPA) (Shpetner et al, 1996; Swanson and Watts, 1995). We determined that 50 μM for wortmannin and EIPA is a working concentration without any cell cytotoxicity (Fig. EV5E,F). Using dextran uptake assay, we confirmed the inhibition capacity of wortmannin and EIPA for macropinocytosis (Fig. EV5G,H). Analysis by attachment assay showed that the inhibition of macropinocytosis did not hinder HBV binding to the cell surface (Fig. EV5I). HBV internalization was then assayed in cells treated with wortmannin and EIPA. Myrcludex B was used as control for HBV entry inhibition. The results showed that inhibition of macropinocytosis led to a significantly decrease in HBV internalization (Fig. 7A). To confirm this observation, we infected cells with HBV particles prepared with heparin to rule out potential effects of PEG and naked HBV capsid on macropinocytosis and assayed HBV internalization at different time points post infection. Whereas treatment with wortmannin and EIPA had no effect on HBV entry at 2 hpi, inhibition of macropinocytosis significantly decreased HBV entry at 6 hpi (Fig. EV5J). Immunofluorescence with anti-preS1 antibody further evidenced that macropinocytosis inhibitors blocked HBV entry into host cells (Fig. 7B,C). The levels of intracellular HBV DNA, HBV DNA in secreted virions and HBeAg were significantly decreased (Fig. 7D–F). Of note, macropinocytosis inhibition did not influence NTCP membrane expression (Fig. EV5K). These results suggest that macropinocytosis is an avenue for HBV internalization and its inhibition leads to decreased HBV entry.

To dissect the relationship between CDC42 and HBV-related macropinocytosis, we treated HepG2-NTCP CDC42-DN cells with wortmannin and performed HBV internalization assays. Analysis of HBV DNA by qPCR showed that treatment with wortmannin did not further decrease HBV internalization in HepG2-NTCP CDC42-DN cells (Fig. 7G), nor in ML141-treated HepG2-NTCP cells (Fig. 7H). Cells and media were collected at 7 dpi for qPCR and ELISA assays, respectively. The levels of intracellular HBV DNA, HBV DNA in secreted virions and secreted HBeAg were not further inhibited by wortmannin treatment in HepG2-NTCP

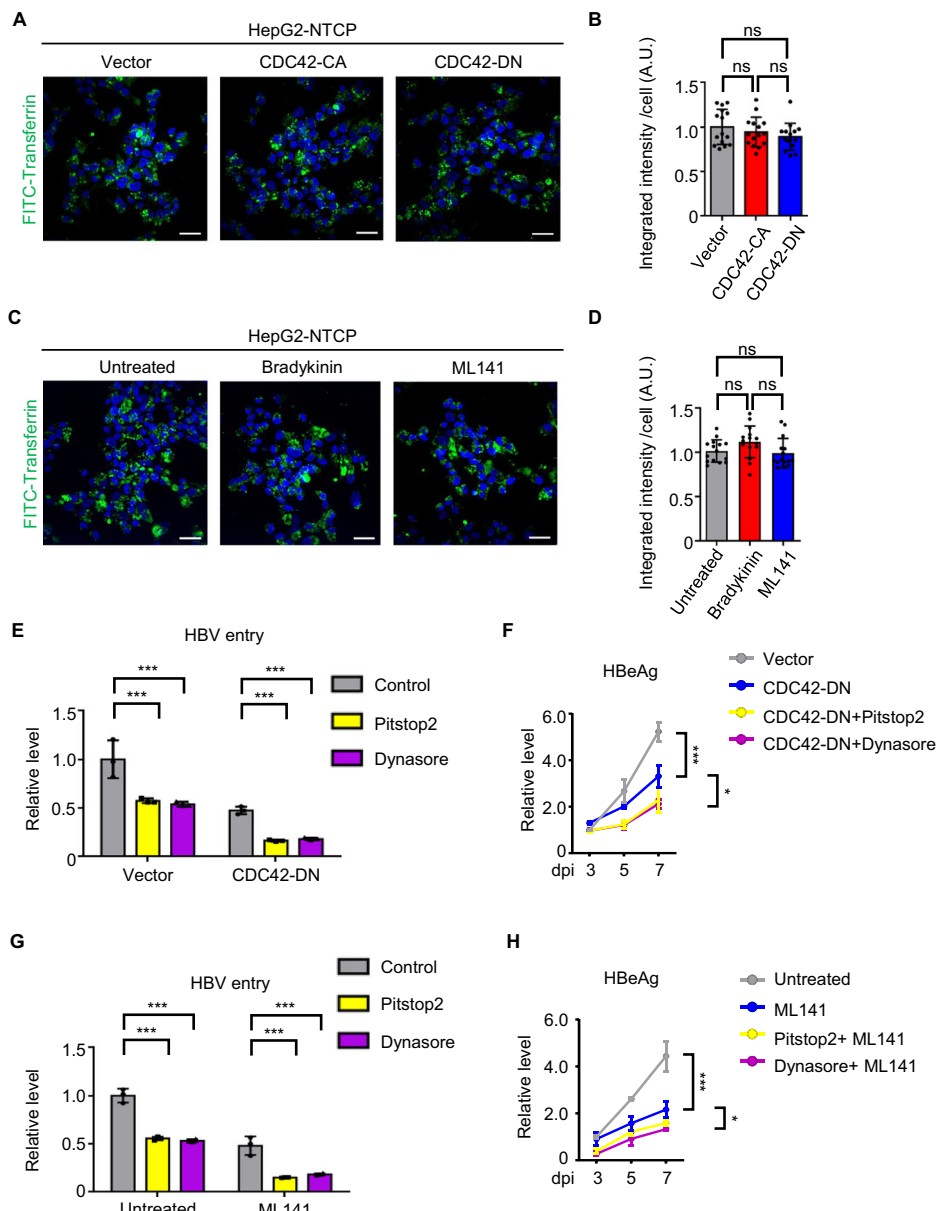

**Figure 6.  The effects of CDC42 on HBV internalization is independent of clathrin-mediated endocytosis.**

(**A**) Transferrin uptake assay in HepG2-NTCP vector, CDC42-CA and CDC42-DN cells. Transferrin is shown in green and DAPI in blue. Scale bar = 10 μm.
(**B**) Quantification of fluorescence intensity of FITC-Transferrin within each cell in A. Data are shown as fold changes to the vector group. $n = 15$ views (100×/1.5 oil objective). (**C**) Transferrin uptake assay in HepG2-NTCP cells treated with bradykinin (100 ng/ml) or ML141 (50 μM). Transferrin is shown in green and DAPI in blue. Scale bar = 10 μm. (**D**) Quantification of fluorescence intensity of FITC-Transferrin within each cell in (C). Data are shown as fold changes to the untreated group. $n = 15$ views (100×/1.5 oil objective). (**E**) HepG2-NTCP vector and CDC42-DN cells treated with CME inhibitors pitstop2 (40 μM) or dynasore (100 μM) were infected with HBV and collected at 8 hpi. Intracellular HBV DNA was quantified by qPCR. Data are shown as fold changes to the control group in HepG2-NTCP vector cells. $n = 3$. Pitstop2 in vector: $P = 0.0009$; Dynasore in vector: $P = 0.0007$; pitstop2 in CDC42-DN: $P = 0.0002$; Dynasore in CDC42-DN: $P = 0.0003$. (**F**) HepG2-NTCP vector and CDC42-DN cells treated with CME inhibitors were infected with HBV. Cell culture media were collected at indicated time points post-infection for the quantification of HBeAg by ELISA. $n = 3$. CDC42-DN versus vector at 7 days post infection: $P = 0.0006$; CDC42-DN+Pitstop2 versus CDC42-DN at 7 days post infection: $P = 0.0173$; CDC42-DN +Dynasore versus CDC42-DN at 7 days post infection: $P = 0.0151$. (**G**) HepG2-NTCP cells treated with indicated reagents were infected with HBV and collected at 8 hpi. Intracellular HBV DNA was quantified by qPCR. Data are shown as fold changes to the control group in HepG2-NTCP untreated cells. $n = 3$. Pitstop2 in untreated: $P = 0.0005$; Dynasore in untreated: $P = 0.0004$; pitstop2 in ML141: $P = 0.0004$; Dynasore in ML141: $P = 0.0006$. (**H**) HepG2-NTCP cells treated with indicated reagents were infected with HBV. Cell culture media were collected at indicated time points post-infection for the quantification of HBeAg by ELISA. $n = 3$. ML141 versus untreated at 7 days post infection: $P = 0.0002$; Pitstop2 + ML141 versus ML141 at 7 days post infection: $P = 0.0452$; Dynasore+ML141 versus ML141 at 7 days post infection: $P = 0.0118$. These experiments were repeated three times. Data are represented as mean ± SEM. A.U. arbitrary unit, ns no significant difference; *$P < 0.05$; ***$P < 0.001$ (unpaired $t$ test). Source data are available online for this figure.

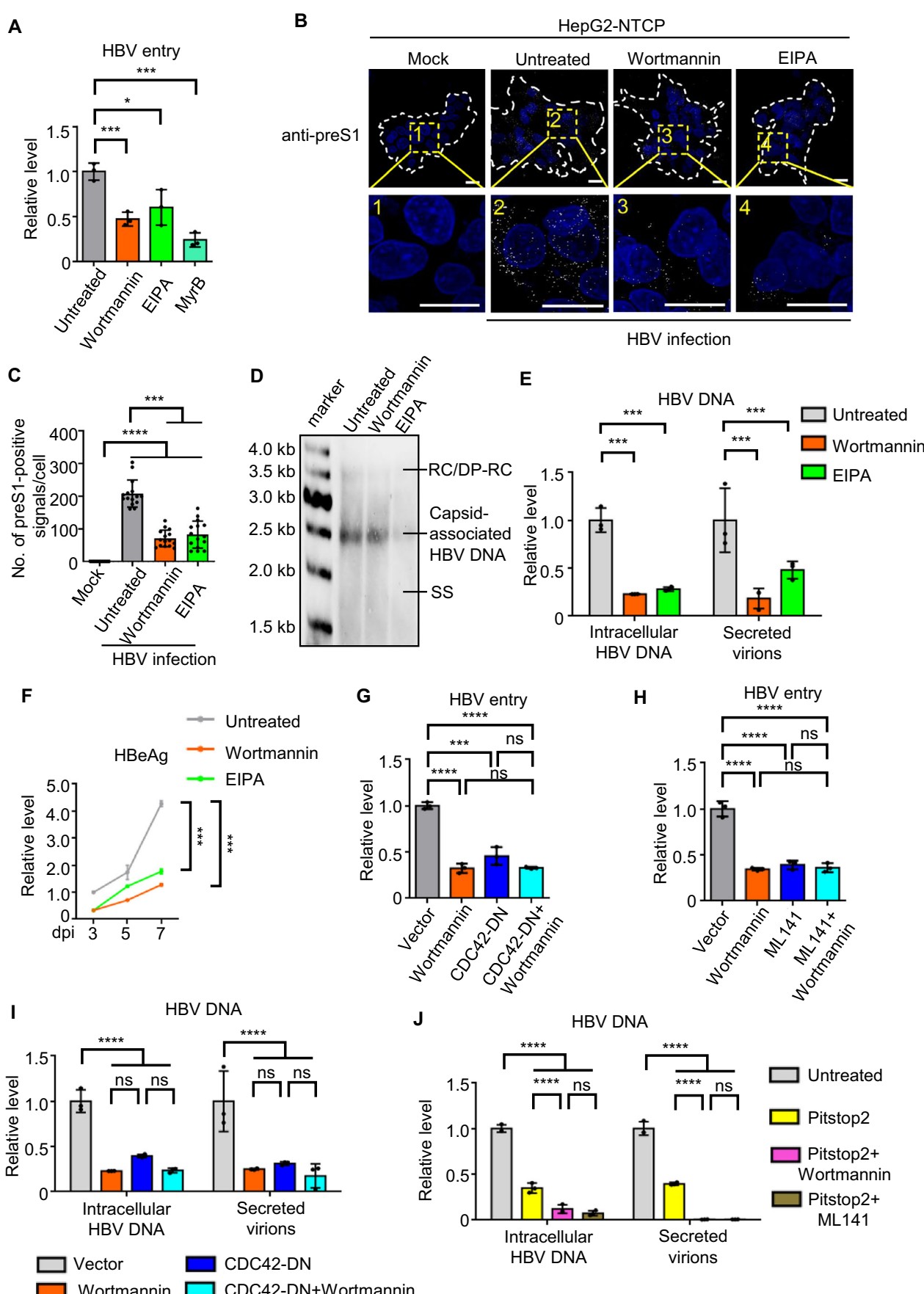

**Figure 7. Macropinocytosis is involved in CDC42-mediated HBV internalization.**

(A) HepG2-NTCP cells treated with 50 µM wortmannin, 50 µM 5-(N-Ethyl-N-isopropyl)-Amiloride (EIPA) and 200 nM Myrcludex B (MyrB) were infected with HBV and collected at 8 hpi. Intracellular HBV DNA was quantified by qPCR. Data are shown as fold changes to the untreated group. $n = 3$. Wortmannin: $P = 0.0002$; EIPA: $P = 0.0182$; MyrB: $P = 0.0001$. (B) HepG2-NTCP cells treated with wortmannin (50 µM) and EIPA (50 µM) were infected with HBV. At 8 hpi, cells were stained with anti-preS1 antibody. Nucleus was stained with DAPI. The cell edges are outlined by a white dotted line. Scale bar = 10 µm. (C) Quantification of preS1-positive signals per cell in (B). $n = 15$ views (100×/1.5 oil objective). Untreated, Wortmannin and EIPA versus mock: $P < 0.0001$; Wortmannin versus untreated: $P = 0.0007$; EIPA versus untreated: $P = 0.0009$. (D) HepG2-NTCP cells treated with indicated reagents were infected with HBV. Cytoplasmic encapsidated DNA was purified at 14 dpi and detected by Southern blot analysis using an HBV DNA probe. RC/DP-RC relaxed circular DNA, DP-RC deproteinized RC DNA, SS single-stranded (−) DNA. (E) Intracellular HBV DNA and secreted HBV virion DNA from HepG2-NTCP untreated cells, wortmannin (50 µM) or EIPA (50 µM) treated cells were purified at 7 dpi and measured by qPCR. Data are shown as fold changes to the untreated group. $n = 3$. Intracellular HBV DNA of Wortmannin: $P = 0.0001$; intracellular HBV DNA of EIPA: $P = 0.0002$; secreted virions of Wortmannin: $P = 0.0007$; secreted virions of EIPA: $P = 0.0009$. (F) Viral HBeAg released from infected cells was quantified by ELISA. Data are shown as fold changes to the untreated group at 3 dpi. $n = 3$. Wortmannin at 7 days post infection: $P = 0.0001$; EIPA at 7 days post infection: $P = 0.0001$. (G) HepG2-NTCP vector and HepG2-NTCP CDC42-DN cells treated with wortmannin (50 µM) were infected with HBV and collected at 8 hpi. Intracellular HBV DNA was quantified by qPCR. Data are shown as fold changes to the vector group. $n = 3$. Wortmannin and CDC42-DN + Wortmannin versus vector: $P < 0.0001$; CDC42-DN versus vector: $P = 0.0007$. (H) HepG2-NTCP cells treated with indicated reagents were infected with HBV and collected at 8 hpi. Intracellular HBV DNA was quantified by qPCR. Data are shown as fold changes to the untreated group. $n = 3$. Wortmannin, ML141 and ML141 + Wortmannin versus vector: $P < 0.0001$. (I) Intracellular HBV DNA and secreted HBV virion DNA from HepG2-NTCP vector and CDC42-DN cells upon wortmannin treatment were purified at 7 dpi and measured by qPCR. Data are shown as fold changes to the vector group. $n = 3$. Intracellular HBV DNA and secreted virions of Wortmannin, CDC42-DN and CDC42-DN + Wortmannin versus vector: $P < 0.0001$. (J) Intracellular HBV DNA and secreted HBV virion DNA from HepG2-NTCP cells treated with indicated reagents were purified at 7 dpi and measured by qPCR. Data are shown as fold changes to the untreated group. $n = 3$. Intracellular HBV DNA and secreted virions of Pitstop2, Pitstop2 + Wortmannin and Pitstop2 + EIPA versus untreated: $P < 0.0001$; Intracellular HBV DNA and secreted virions of Pitstop2 + Wortmannin versus pitstop2: $P < 0.0001$. These experiments were repeated three times. Data are represented as mean ± SEM. ns, no significant difference; $*P < 0.05$; $***P < 0.001$; $****P < 0.0001$ (unpaired $t$ test). Source data are available online for this figure.

CDC42 DN cells (Figs. 7I and EV5L). These results support the notion that inactivation of CDC42 invalidates macropinocytosis-mediated HBV entry.

To dissect the relationship between CDC42-macropinocytosis and CME, we treated HepG2-NTCP cells with pitstop2 with the addition of ML141 or wortmannin and then infected the cells with HBV. Analysis of intracellular HBV DNA, secreted HBV virion DNA and HBeAg showed that the CME inhibitor further inhibited HBV infection in CDC42-inactivated and macropinocytosis-inhibited cells (Figs. 7J and EV5M). Together, we conclude that CDC42 regulates HBV entry through macropinocytosis, and HBV-related macropinocytosis and HBV-related CME are two independent venues for HBV entry.

## Discussion

In this study, we demonstrate that the small GTPase CDC42 is a key positive regulator of HBV infection and reveal the mechanisms by which CDC42 regulates HBV entry. We show that CDC42 promotes translocation of the viral receptor NTCP to the plasma membrane and macropinocytosis-mediated virion internalization. Our findings provide insights into the regulation of NTCP membrane localization in hepatocytes and identify CDC42 as a novel regulator in the early step of HBV infection.

NTCP is recognized as the receptor for HBV (Asami et al, 2022; Yan et al, 2012) and targeted by the preS1 domain of the HBV large envelop protein. Exogenous expression of NTCP in cell lines that are normally not susceptible to HBV, such as HepG2 and Huh7 cells, can confer their susceptibility to HBV infection (Ni et al, 2014). The physiological function of NTCP in the liver is to uptake conjugated bile acids into hepatocytes, providing hepatics influx of bile salts (Anwer and Stieger, 2014; Appelman et al, 2021). NTCP is present at the plasma membrane and in endocytic vesicles, where it transports to the plasma membrane via recycling endosomes (Sarkar et al, 2006). Recent studies have unveiled the roles of other cellular factors in the regulation of NTCP in the context of HBV entry. EGFR has been shown capable of interacting with NTCP to initiate its oligomerization, thus serving as an HBV entry co-factor (Fukano et al, 2021; Iwamoto et al, 2019). It has been reported that interferon-induced transmembrane proteins 3 (IFITM3) is involved in HBV internalization by interacting with NTCP (Palatini et al, 2022). Here, we show that CDC42 does not interact with NTCP, but signals NTCP translocation to the plasma membrane. This is achieved via enhancement of recycling endosome activity by CDC42, as knockdown of the recycling endosome marker Rab11 strongly affects the level of NTCP at the plasma membrane, HBV entry and viral replication. Previous reports demonstrated that Rab11 is involved in HBV infection (Chu et al, 2022; Inoue et al, 2019). We show that NTCP interacts with Rab11 and its interaction is greatly strengthened by the activation of CDC42 signaling. Although it has been reported that CDC42 is enriched on recycling endosomes in HeLa cells (Balklava et al, 2007), our immunoprecipitation assays show absence of CDC42 in NTCP/Rab11 complex.

The CDC42 pathway plays a pivotal role in modulating cellular actin dynamics via a multiplex network. Currently, the downstream effectors that connect GTPase-bound CDC42 to NTCP/Rab11 complex are unknown. Previous studies have showed that PI3K/PKB/PKC can trigger the translocation of NTCP from intracellular compartments towards the plasma membrane (McConkey et al, 2004; Mukhopadhayay et al, 1997; Webster et al, 2002). CDC42 can promote the PI3K pathway in several contexts, including integrin/immune complex-stimulated neutrophil, melanoma cells, and mouse oocytes (Poku et al, 2023; Rodal et al, 2008; Yan et al, 2018). Thus, CDC42 may promote NTCP trafficking by regulating the PI3K pathway. Our findings indicate that the plasma membrane translocation mediated by CDC42 signaling is specific to NTCP, as membrane recycling of EGFR does not respond to CDC42. The mechanisms involved in this fine selection of downstream targets

by CDC42 signaling remain to be further elucidated. We observed that NTCP is almost equally expressed on the basal and apical membranes in HepG2-NTCP cells. Previous study found that the polarized basolateral localization of human NTCP is cell-type specific (Sun et al, 2001). The reason why NTCP overexpressed in HepG2 lacks the polarity pattern seen in the liver is not known. We speculate that this patterning may require other cell components and microenvironmental cues in the liver but absent in the cell culture system.

Detailed mechanisms for how HBV enters into hepatocytes are not completely understood. Cellular endocytic pathways are frequently hijacked by viruses for productive entry. Macropinocytosis is an actin-dependent endocytic process that is closely regulated by CDC42 and is used as entry route for several viruses (Koivusalo et al, 2010). We show that cells treated with reagents inhibiting macropinocytosis are much less susceptible to HBV infection. The same reagents lost their effects on HBV entry when they were administrated in cells where CDC42 was inactivated, indicating that HBV entry through macropinocytotic process is regulated by CDC42 signaling. In previous reports, macropinocytosis is dispensable for HBV entry (Chakraborty et al, 2020; Herrscher et al, 2020). This discrepancy may reside in the differences of the duration of the treatment with macropinocytosis inhibitors and HBV infection protocol. We treated cells with wortmannin or EIPA for 2 h and incubated with HBV for 8 h in the presence of wortmannin or EIPA (see "Methods"), in contrast to 1 h treatment and then infected with HBV for 2 h (Herrscher et al, 2020) or 6 h treatment during HBV inoculation (Chakraborty et al, 2020) in the previous reports. The fact that HepG2 cells are resistant to HBV infection suggests that macropinocytosis-mediated HBV entry in HepG2-NTCP cells is dependent on the receptor NTCP. It is yet unknown whether the attachment of HBV with HSPGs on the hepatocyte induces CDC42 activation, which would favor NTCP membrane transport and macropinocytosis.

It has been reported that clathrin-mediated endocytosis consists of a pathway for HBV entry (Herrscher et al, 2020; Huang et al, 2012). Our study confirmed this observation and reveals that the CME-mediated HBV internalization is independent of CDC42 signaling. We show that inhibition of CME or macropinocytosis results in reduced internalization of HBV and decreased levels of HBeAg, indicating that both pathways can lead to productive viral infection. These data indicate that HBV involves various cellular endocytic pathways to ensure the success of its infection. Which pathway is more efficient to establish HBV infection? Does HBV randomly utilize the endocytic pathways for its entry, or does it selectively use specific ones? What are the factors that influence the selection of the entry pathway by the virus? These questions remain unanswered. It appears that successfully complete endosomal escape is a key step in HBV infection. Both clathrin-coated vesicles (CCVs) derived from CME and macropinosomes generated via macropinocytosis undergo vesicular fusion to form late endosomes and proceed through maturation (Donaldson, 2019). Further studies on the processes of CCVs and macropinosomes maturation may provide valuable insights into the detailed mechanisms of the endocytic pathways involved in HBV entry.

Together, our data uncover novel functions of CDC42 in HBV-receptor NTCP recycling and macropinocytosis-mediated HBV internalization, placing CDC42 in an important position as an antiviral target. Efforts to develop selective small molecule drugs for the family of GTPases have mainly focused on the Ras subfamily in the treatment of cancers. With the advent of the development of molecules targeting the large GTPase superfamily (Morstein et al, 2024), we may hope in the near future the discovery of CDC42 inactivator that blocks HBV infection.

## Methods

**Reagents and tools table**

| Reagent/resource | Reference or source | Identifier or catalog number |
|---|---|---|
| **Experimental models** | | |
| HepG2-NTCP | Cui et al, 2023 | N/A |
| Primary human hepatocytes (PHH) | Liver-Biotechnology (Shenzhen) | LV-WEM001 |
| HBV | This study | N/A |
| **Recombinant DNA** | | |
| Plasmid: Flag-CDC42-WT pCDH-CMV-puro | This study | N/A |
| Plasmid: Flag-CDC42-CA pCDH-CMV-puro | This study | N/A |
| Plasmid: Flag-CDC42-DN pCDH-CMV-puro | This study | N/A |
| Plasmid: mCherry-Rab11 pCDH-CMV-puro | This study | N/A |
| Plasmid: NTCP-EGFP pCDH-CMV-Blast | This study | N/A |
| Plasmid: shRab11 pLKO.1 puro | This study | N/A |
| Plasmid: pHBV1.3 | Deng et al, 2009 | N/A |
| **Antibodies** | | |
| Anti-Cdc42 Polyclonal antibody (for WB) | Proteintech | 10155-1-AP |
| anti-Flag clone M2 (for WB) | Sigma-Aldrich | F3165 |
| anti-GAPDH Rabbit antibody (for WB) | Sigma-Aldrich | G9545 |
| Anti-GFP (for WB) | Sigma-Aldrich | G6795 |
| anti-HBc [C1] (for IF) | Abcam | ab8637 |
| Anti-LAMP-2(H4B4) (for IF) | Santa Cruz | sc-18822 |
| Anti-MRP2 (for IF) | Med Chem Express | HY- P81118 |
| Anti-NTCP Polyclonal Antibody (for IF) | Invitrogen | PA5-25614 |
| Anti-preS1(AP1) (for IF) | Santa Cruz | sc-57761 |
| Anti-Rab11 (D4F5) XP® Rabbit mAb (for WB and IF) | Cell Signaling Technology | 5589P |
| Anti-SLC10A1/NTCP1 Antibody (for WB) | Abcam | ab131084 |
| Anti-Transferrin Receptor Monoclonal Antibody (H68.4) (for WB) | Invitrogen | 13-6800 |
| Alexa Fluor 488 goat anti-rabbit IgG (H + L) | Invitrogen | A11008 |

| Reagent/resource | Reference or source | Identifier or catalog number |
|---|---|---|
| Alexa Fluor 488 goat anti-mouse IgG (H + L) | Invitrogen | A11001 |
| Alexa Fluor 568 goat anti-rabbit IgG (H + L) | Invitrogen | A11011 |
| Alexa Fluor 555 goat anti-mouse IgG (H + L) | Invitrogen | A21422 |
| Horseradish peroxidase-linked anti-mouse IgG antibody | Cell Signaling Technology | 7076 V |
| Horseradish peroxidase -linked anti-rabbit IgG antibody | Cell Signaling Technology | 7074 V |
| **Oligonucleotides and other sequence-based reagents** | | |
| Primers for qPCR | This study | N/A |
| **Chemicals, enzymes and other reagents** | | |
| 2×Color SYBR Green qPCR Master Mix | EZBioscience | A0012-R2 |
| 5-(N-ethyl-N-isopropyl)-Amiloride | Cayman chemical | 14006 |
| Active Cdc42 Detection Kit | Cell Signaling Technology | #8819 |
| Bradykinin | TargetMol | TP1277 |
| Cell Counting Kit-8 | Beyotime Biotechnology | C0039 |
| Color Reverse Transcription Kit | EZBioscience | A0010CGQ |
| DAPI | Beyotime | C1006 |
| Dynasore | MedChemExpress | HY-15304/CS-1340 |
| Enhanced BCA Protein Assay Kit | Beyotime | P0010 |
| EZ-press RNA Purification Kit | EZBioscience | B0004DP |
| FITC-Transferrin | Ruixibio | R-F8802 |
| heparin | MedChemExpress | HY-17567 |
| Heparin HiTrap columns, | Cytiva™ | 17040601 |
| HBeAg enzyme-linked immunosorbent assay kit | Shanghai Kehua | N/A |
| LysoTracker™ Deep Red | Invitrogen | L12492 |
| ML141 | Selleckchem | S7686 |
| Myrcludex B | MedChemExpress | HY-P3465 |
| Pitstop2 | Cayman chemical | 23885 |
| Rhod-Dextran | Sigma-Aldrich | 9379 |
| Sulfo-NHS-SS-Biotin | APExBIO | A8005 |
| Streptavidin Beads T3 | APExBIO | K1301 |
| taurocholate (TC) | Aladdin | T336942-5mg |
| Total bile acid (TBA) kits | Nanjing Jiancheng Bioengineering Institute | E003-2-1 |
| Triton X-100 | Sangon Biotech | T0694 |
| Trizol reagent | Tiangen | DP405 |
| Wortmannin | Medbio | MED19561 |
| **Software** | | |
| Excel, 2016 | N/A | N/A |

| Reagent/resource | Reference or source | Identifier or catalog number |
|---|---|---|
| ImageJ Fiji | N/A | N/A |
| Prism, v. 8.4.3 | Graphpad Prism | N/A |
| **Other** | | |

## Cell culture

The human hepatoma cell line HepG2 (ATCC #HB-8065) and HEK293T (ATCC#CRL-3216) cell line were cultured at 37 °C with 5% $CO_2$ in Dulbecco's modified Eagle's medium (DMEM) (Biological Industries,01-055-1 A) supplemented with 10% fetal bovine serum (FBS) (YLESA, S211201T) and 1% penicillin and streptomycin (Biological Industries, 03-031-1B). HBV genome stably transfected HepAD38 cell line was maintained in DMEM containing 10% FBS, 1% penicillin and streptomycin, 200 µg/mL G418 (Invivogen, ant-gn-1) and 1 µg/mL doxycycline (Sigma-Aldrich, D9891). Viral replication is induced when doxycycline is removed (LADNER et al, 1997). The generation of HepG2-NTCP cell line has been reported previously (Cui et al, 2023). HepG2-NTCP-EGFP cells were generated using lentiviral vectors encoding human NTCP fused to EGFP (NTCP-EGFP), followed by clonal selection in medium containing 5 µg/mL blasticidin (Invivogen, ant-bl-1).

Primary human hepatocytes (PHH) were purchased from Liver-Biotechnology (Shenzhen) and cultured at 4 °C with 5% $CO_2$ in PHH Maintenance Medium (Liver-Biotechnology, LV-WEM001) supplemented with 1% penicillin and streptomycin (Biological Industries, 03-031-1B).

## HBV production and titration

HepAD38 cells were cultured in DMEM supplemented with MEM non-essential amino acids (NEAA) (Gibco, 11140–035), 10% FBS, 1% penicillin and streptomycin, 5 µg/mL insulin, 25 µg/mL hydro-cortisone (Cayman, 18226-1), 2% dimethyl sulfoxide (DMSO) (Cayman, 18226-1). We used two methods to produce HBV particles from HepAD38 culture. Method 1: HBV stock was concentrated from HepAD38 cell culture supernatant using PEG8000 (Sigma, 89510) to a final concentration of 8% PEG, and incubated overnight at 4 °C. Viral particles were collected by centrifugation at 10,000 $\times g$ for 1 h at 4 °C. The viral pellets were suspended in hepatocyte maintenance medium (PMM). Virus titers were quantified by qPCR. Method 2: HBV particles were also prepared with Heparin HiTrap columns (Cytiva™,17040601). In brief, viral particles in HepAD38 cell culture supernatant was purified by passing over Heparin HiTrap columns (5 mL) and bound virus was eluted with NaCl (390 mM) and purified by sucrose gradient centrifugation (3 mL 60%, 7 mL 25% and 9 mL 15%) at 32,000 rpm. The resulting gradient was fractionated in 2 mL aliquots and infectivity of the virus-rich fraction evaluated in HepG2-NTCP cells and aliquots stored at −80 °C (Chakraborty et al, 2020; Wettengel et al, 2021). We compared PEG-purified particles with heparin-purified particles infecting HepG2-NTCP cells in

HBV internalization assays and found no significant difference. PEG-prepared HBV stock was used in the infection experiments shown in Figs. 1B–E,G, EV1E–G, 2F–J, EV3D, 5C–E, EV5L,M, 6E–H, and 7A–J. Heparin-prepared HBV stock was used in the infection experiments shown in Figs. 1H–J, 2A–E, EV2A–C, 3A,B,E, EV3G, and EV5I.

## Cloning

Flag-tagged human wild-type (WT), constitutively active (CA, substitution of Gly$^{12}$ to Val) and dominant-negative (DN, substitution of Thr$^{17}$ to Asp) CDC42 mutant fragments (kindly provided by Prof. Xueliang Zhu, State Key Laboratory of Cell Biology, CAS Center for Excellence in Molecular Cell Science, Shanghai Institute of Biochemistry and Cell Biology, Chinese Academy of Sciences) were cloned into pCDH-CMV-puro. NTCP fused with EGFP was cloned into pCDH-CMV-blasticidin (the puromycin resistance sequence in the pCDH-CMV-puro plasmid was replaced with the blasticidin resistance sequence). mCherry-tagged Rab11 fragment was amplified from pHR-FKBP:mCherry-Rab11a (Addgene, #72902) and cloned into pCDH-CMV-puro.

## Production of lentiviruses

In total, $10^7$ HEK293T cells were seeded 18 h earlier in 10-cm tissue culture dishes. Cells were transfected with 14 μg lentivirus plasmids along with 7 μg pCMV-VSVG and 3.5 μg pCMV-delta 8.9. Sixteen hours later, medium was changed. Every 24 h, culture supernatants were collected for two times, filtered and clarified by brief centrifugation. The supernatants were then mixed with PEG 6000 (Sigma, 81260) to a final concentration of 8% PEG and incubated overnight at 4 °C. Viral particles were collected by centrifugation at $1500 \times g$ for 30 min. The viral pellets were suspended in complete DMEM medium.

## Generation of cell lines

HepG2-NTCP vector, HepG2-NTCP CDC42-WT, HepG2-NTCP CDC42-CA and HepG2-NTCP CDC42-DN cells were generated by transduction of lentiviruses expressing Flag vector, Flag-tagged WT CDC42, CDC42-CA and CDC42-DN into HepG2-NTCP cells. HepG2-NTCP-EGFP CDC42-WT, HepG2-NTCP-EGFP CDC42-CA and HepG2-NTCP-EGFP CDC42-DN cells were generated by transduction of lentiviruses expressing Flag-tagged WT CDC42, CDC42-CA and CDC42-DN into HepG2-NTCP-EGFP cells. HepG2-NTCP-EGFP mcherry-Rab11 cells were generated by transduction of lentiviruses expressing mcherry-Rab11. Cells were selected in medium containing 2 μg/mL puromycin (Invivogen, ant-pr-1). HepG2-NTCP Rab11 stable knockdown cell line is generated by the delivery of cassettes expressing hairpin RNA using pLKO.1 puro vector (Addgene #8453) and selection with 2 μg/ml puromycin. The sequence of hairpin RNAs used for Rab11 knockdown is 5′-CCGG-CGAGCTATAACATCAGCATAT-CTCG AG-ATATGCTGATGTTATAGCTCG-TTTTTG-3′.

The growth curve of HepG2-NTCP vector, HepG2-NTCP CDC42-CA and HepG2-NTCP CDC42-DN cells were obtained by seeding $2.5 \times 10^4$ cells into 12-well plates in triplicate. Cell numbers were determined every day for a total of 5 days.

## CDC42 activity assay

CDC42 activity assay was performed following the manufacturer's protocol (CST, #8819) and has been reported previously (Zhao et al, 2023). Briefly, $1 \times 10^7$ cells were collected, washed with cold PBS and incubated for 5 min on ice in lysis buffer. The insoluble fraction of cell lysates was removed by centrifugation at $16,000 \times g$ and 4 °C for 15 min. Lysate protein concentration was determined using BCA protein assay (Beyotime Biotechnology, P0010) and adjusted to the same concentration by adding lysis buffer. Cell lysates (equivalent volume for each experimental group) were incubated with GST-PAK1-PBD fusion protein bound to glutathione resin at 4 °C for 30 min. The GTP-bound GTPase binds to the glutathione resin through GST-linked binding protein. The resin was washed three times with Wash Buffer and eluted in SDS Sample Buffer. Eluted proteins were then analyzed by western blotting using antibody against human CDC42.

## Cell treatments and transfection

Bradykinin (TargetMol, TP1277), ML141 (Selleckchem, S7686), Myrcludex B (MedChemExpress, HY-P3465), pitstop2 (Cayman Chemical, 23885), dynasore (MedChemExpress, HY-15304/CS-1340), wortmannin (Medbio, MED19561), EIPA (Cayman Chemical, 14006) and heparin (MedChemExpress, HY-17567) were purchased. For the tests of the functions of CDC42 on HBV infection, seeded HepG2-NTCP cells were treated with bradykinin (100 ng/ml) or ML141 (50 μM) for 2 h before HBV infection. The cells were then infected with HBV for 8 h in the presence of the reagents. To block HBV attachment to the cell surface, HepG2-NTCP cells were treated with heparin (50 IU/mL) during incubation with HBV at 4 °C. To block HBV entry, HepG2-NTCP cells were treated with Myrcludex B (200 nM) for 2 h and then infected with HBV for 8 h in the presence of Myrcludex B.

To test whether bradykinin and ML141 have effects on HBV replication after entry-step, HepAD38 cells were first treated with either bradykinin (100 ng/ml) or ML141 (50 μM) for 2 h in the presence of doxycycline to inhibit pgRNA transcription, then for another 8 h in the absence of doxycycline to allow pgRNA transcription. Cells were collected for the preparation of encapsidated HBV DNA 7 days later. We used this treatment protocol because we wanted to be in line with HepG2-NTCP/HBV infection treatment. For transfection with the plasmid containing 1.3-mer HBV genome (pHBV1.3), HepG2 cells were first treated with bradykinin (100 ng/ml) or ML141 (50 μM) for 2 h, then transfected with pHBV1.3. Cells were treated with bradykinin or ML141 for another 8 h. Cells were collected at 48 h post-transfection for the preparation of encapsidated HBV DNA.

To test the role of macropinocytosis in HBV infection, HepG2-NTCP cells were treated with macropinocytosis inhibitors wortmannin (50 μM) and EIPA (50 μM) for 2 h and then infected with HBV for 8 h in the presence of wortmannin and EIPA.

## Cytotoxicity assay

Cytotoxicity assay was performed using Cell Counting Kit-8 (CCK8, Beyotime Biotechnology, C0039) according to the manufacturer's instructions. HepG2-NTCP cells were plated in 96-well plate ($3 \times 10^3$ cells/well) and treated by indicated reagents for 10 h.

CCK-8 solution was added to the wells for 4 h and the absorbance was measured at 450 nm.

## Western blot

Cells were lysed in RIPA lysis buffer (Beyotime, P0013D) supplemented with protease (Beyotime, P1009) and phosphatase inhibitors (Beyotime, P1082). Protein concentrations were determined using the BCA Protein Assay kit. After boiling, protein samples were subjected to SDS-polyacrylamide gel electrophoresis and immunoblotting. The blots were blocked with 5% dry milk in Tris-buffered saline containing 0.1% Tween 20 (TBST) and incubated with primary antibodies overnight at 4 °C. Primary antibodies were diluted in Primary Antibody Dilution Buffer (Beyotime, P0023A). Following antibodies were used: anti-Cdc42 Polyclonal antibody (Proteintech, 10155-1-AP), anti-Flag clone M2 (Sigma-Aldrich, F3165), anti-GAPDH Rabbit antibody (Sigma-Aldrich, G9545), anti-GFP (Sigma-Aldrich, G6795), anti-Lamp2 (H4B4) (Santa cruz, sc-18822), anti-SLC10A1/NTCP1 Antibody (Abcam, ab131084), anti-Transferrin Receptor Monoclonal Antibody (H68.4) (Invitrogen, 13-6800), anti-EGF Receptor (D38B1) XP® Rabbit mAb (Cell Signaling Technology, 4267S), anti-Rab11 (D4F5) XP® Rabbit mAb (Cell Signaling Technology, 5589P). After washing with TBST, membranes were incubated with corresponding secondary antibodies (horseradish peroxidase-linked anti-mouse IgG antibody, Cell Signaling Technology, 7076V; horseradish peroxidase-linked anti-rabbit IgG antibody, Cell Signaling Technology, 7074V). Immunoreactive bands were visualized by an enhanced chemiluminescence system (Beyotime Biotechnology, P0018FM). The levels of protein expression were quantified with ImageJ software. Intensities of bands of interests were quantified with background subtraction and normalization to loading controls.

## Co-immunoprecipitation

For co-immunoprecipitation, cells were plated in 6-cm dishes. After washing by PBS, cells were lysed in NP-40 lysis buffer (50 mM Tris-HCl pH 8.0, 150 mM NaCl, 1% NP-40, protease inhibitors and phosphatase Inhibitors). Cell lysate was centrifuged for 10 min at 4 °C. Targeted proteins were either precipitated by antibodies or Flag beads (AlpalifeBio, KTSM1308) at 4 °C overnight. After washes in NP-40 lysis buffer, the precipitates were subjected to WB analysis.

## Cell surface protein isolation assay

HepG2-NTCP cells were seeded in a 10-cm dish for 80% confluency. Cells were washed three times with a buffer containing 0.5 mM $CaCl_2$ and 1 mM $MgCl_2$ in PBS (PBS-CM). 5 mL of 0.5 mg/mL Sulfo-NHS-SS-Biotin (APExBIO, A8005) was added and incubated at 4 °C for 30 min. After removing the liquid, cells were incubated in 10 mL 0.1% BSA in PBS-CM for 20 min to remove unbound biotin. Cells were washed three times with PBS-CM. Cells were lysed for 30 min on ice using 1 ml of lysis buffer containing 150 mM NaCl, 50 mM Tris pH 7.4, 5 mM EDTA, 10% (w/v) sucrose, 1% NP40, and protease inhibitors. The lysate was centrifuged at 12,000 rpm for 30 min at 4 °C and the supernatant was retained for protein quantification and subsequent purification. Subsequently, 20 μL of Streptavidin Beads T3 (APExBIO, K1301) was added to the lysate and incubated at 4 °C overnight. Beads were collected by magnetic stand. The supernatant

was removed. Beads were washed 3–4 times with 1% BSA in PBS-CM. 2×SDS-PAGE loading buffer was added to beads and incubated at 100 °C for 10 min. Samples were then cooled on ice. Beads were eliminated using magnetic stand. The supernatant, which contained membrane proteins, was collected. Samples were analyzed by western blot.

## Bile acids uptake assay

HepG2 NTCP vector, CDC42 CA and CDC42 DN cells were seeded in a 24-well plate for 80% confluency. Cells were washed twice with PBS and cultured in complete culture medium containing 100 μM taurocholate (TC) (Aladdin, T336942-5mg) at 37 °C for 30 min. After the incubation period, the TC solution was collected and the remaining TC concentration was measured using Total bile acid (TBA) kits (Nanjing Jiancheng Bioengineering Institute, E003-2-1). The amount of bile acid taken up by the cells was obtained by subtracting the remaining concentration from the initial concentration.

## Transferrin uptake assay

FITC-Transferrin (Ruixibio, R-F8802) was utilized for the uptake assay. HepG2-NTCP cells were inoculated on coverslips in a 24-well plate. Cells were pre-treated with indicated reagents for 2 h and then incubated with 50 μg/mL FITC-Transferrin at 37 °C for 45 min in the presence of indicated reagents. Cells were washed three times with cold PBS and then fixed with 4% PFA and permeabilized in 0.1% Triton X-100. Coverslips were mounted onto glass slides using DAPI Fluoromount-G Mounting Medium (Southernbiotech, 0100-20) after three washes with 1×PBS. Imaging data were obtained using an Olympus SpinSR10 Ixplore spinning disk confocal microscope with UplanApo 60×/1.2 oil objective. Images were analyzed with ImageJ software.

## Dextran uptake assay

Rhodamine B-Dextran (Sigma-Aldrich, 9379) was used for the uptake assay. HepG2-NTCP cells were inoculated on coverslips in 35 mm glass-bottom dishes. Cells were pre-treated with indicated reagents for 2 h and then incubated with 0.25 mg/ml Rhodamine B-Dextran at 37 °C for 1 h in the presence of indicated reagents. Cells were washed with PBS. Medium without phenol red (Gibco, 21063029) was added, and imaging data were obtained using an Olympus SpinSR10 Ixplore spinning disk confocal microscope with UplanApo 100×/1.5 oil objective. Images were analyzed with ImageJ software.

## Immunofluorescent staining

Cells were inoculated in a 24-well plate on coverslips. Cells were washed with PBS and fixed with 4% PFA for 30 min at room temperature (RT), followed by another PBS wash. Permeabilization was achieved with 0.1% Triton X-100 in PBS for 5 min. Subsequently, cells were blocked in 5% bovine serum albumin (BSA) in PBS (blocking buffer) for 30 min. The primary antibody was diluted in blocking buffer and incubated with cells at RT for 1 h. The secondary antibodies (Invitrogen, A11008, A11001, A11011, A21422) were then incubated with cells at RT for 1 h.

After three washes with PBS, Coverslips were mounted onto glass slides using DAPI Fluoromount-G Mounting Medium. Primary antibodies are: anti-preS1(AP1) (Santa Cruz, sc-57761), anti-Rab11 (D4F5) XP® Rabbit mAb (Cell Signaling Technology, 5589 P), anti-Lamp2 (H4B4) (Santa cruz, sc-18822), anti-MRP2 (MedChemExpress, HY- P81118), anti-HBc [C1] (Abcam, ab8637).

## Microscopy

Imaging data were obtained using an Olympus SpinSR10 Ixplore spinning disk confocal microscope with UplanApo 60×/1.2 oil objective and a 4-channel 15 mW laser light source. For confocal reconstruction, we used 210 nm z-step size for overlapping z-stacks. A series of images were captured from the top to the bottom of cells at different z-axis positions to record the cell panorama.

For living cell imaging, HepG2-NTCP-EGFP-Rab11-mCherry stable cell lines was seeded in 35-mm glass-bottom dishes (MatTek Corporation, P35G-1.5-14-C) at the density of $2 \times 10^4$ cells per dish. Next day, the supernatant was discarded, washed once with PBS. 2 mL of medium without phenol red containing 10% FBS was added. The time-lapse images were acquired with Olympus CellSens Dimension system, consisting of an Olympus SpinSR10 Ixplore spinning disk confocal and a Yokogawa CSU-W1 confocal scanner. Samples were placed in the heated (37 °C) and 5% $CO_2$ environment. In all, 488 nm and 555 nm filters, and UplanApo 100×/1.5 oil objective (Olympus Corporation) were used. The recording was set as every 1 s for 5 min. One focal plane was recorded for all live cell movies.

## Quantitative analysis of images

Cross-sections of cells were reconstructed using ImageJ. The maximum intensity projection was applied to the reconstructed images, which generated the corresponding orthogonal view. The Freehand Line mode was used to draw a curve that fits with the plasma membrane in the orthogonal view. After conducting layer-by-layer scanning of cell samples, an XZ orthogonal image was established at the thickest portion of the cells at z axis. The plasma membrane curve in the orthogonal image was fitted using the Plot Profile tool in ImageJ, and the fluorescence intensity distribution of NTCP-EGFP on the plasma membrane was analyzed. The fluorescence intensity data were obtained with the analysis of 15 views (100×/1.5 oil objective) in each experiment and used to generate the corresponding fluorescence intensity distribution graph in GraphPad Prism 9 software.

For quantitative assessment of co-localization of the dual-color signals obtained in double labeling experiments, Manders Overlap Coefficient values were calculated using Just Another Colocalisation Plugin in ImageJ.

## HBV infection

HBV infection was described previously (Cui et al, 2023). Briefly, antibiotics were removed from the medium at least 48 h before viral infection. HepG2-NTCP cells were seeded in 24-well plates at the density of 15,000 cells per well. Cells were then incubated with HBV at a MOI of 100 genome equivalents (GEq) per cell for 8 h. Based on published studies (Cui et al, 2023; Watashi et al, 2014),

HBV DNA isolated from cells collected at 8 hpi was considered as entered HBV DNA from input virus. HBV DNA isolated on the 7th and 14th days post-infection was designated as the progeny DNA, resulting from viral replication after the establishment of infection. For immunofluorescence with anti-preS1 antibody, cells were infected with HBV at a MOI of 200 GEq per cell. Subsequently, the cells were washed with PBS three times and maintained in hepatocyte maintenance medium (PMM).

## HBV attachment assay and internalization assay

HBV attachment and internalization assay was performed according to the previous descriptions with modifications (Cui et al, 2023). For the attachment assay, HepG2-NTCP cells were incubated with HBV at 4 °C for 3 h and then collected after washing out the free virus.

For the internalization assay, HepG2-NTCP cells were cultured with HBV at 4 °C for 3 h, then changed the medium. Cells were cultured with HBV at 37 °C for 8 h. Alternatively, cells were inoculated with heparin-purified HBV for 1 h at 4 °C and then transferred to 37 °C for 2 h, 4 h or 6 h. Cells were then trypsinized and extensively washed prior to quantifying the cellular HBV DNA.

## Isolation and analysis of viral DNA

HBV virion DNA was extracted from culture supernatants using the TIANamp Virus DNA/RNA Kit (Tiangen, DP315) following the manufacturer's instructions. Intracellular HBV DNA was extracted using the QIAamp DNA kit (Qiagen, 51306).

Encapsidated viral DNA was prepared as follows. Cells were lysed with lysis buffer (10 mM Tris-HCl (pH 8.0), 1 mM EDTA, 1% NP-40, and 2% sucrose) at 37 °C for 30 min. After removing the nuclear pellet by centrifugation, the supernatant was incubated with 7.2% PEG 8000 containing 1.5 M NaCl at 4 °C overnight. Viral nucleocapsids were pelleted by centrifugation at $10,000 \times g$ for 30 min at 4 °C, followed by digestion in digestion buffer (0.5 mg/ml protease K, 0.5% sodium dodecyl sulfate (SDS), 150 mM NaCl, 25 mM Tris-HCl (pH 8.0), and 10 mM EDTA) at 37 °C overnight. DNA released from nucleocapsid was purified by phenol-chloroform extraction, ethanol precipitation and resuspended in TE buffer (10 mM Tris-HCl, pH 8.0, 1 mM EDTA).

Samples were quantified by qPCR (EZBioscience, A0012-R2). The relative HBV DNA copy number was expressed as the ratio of the copy number of HBV DNA to the β-actin DNA. Primer sequences are following:

HBV rcDNA forward 5′- ATCCTGCTGCTATGCCTCATCTT-3′;
HBV rcDNA reverse 5′-ACAGTGGGGGAAAGCCCTACGAA-3′;
HBV cccDNA forward 5′-TGCACTTCGCTTCACCT-3′;
HBV cccDNA reverse: 5′-AGGGGCATTTGGTGGTC-3′;
β-actin forward 5′-GGG AAATCGTGCGTGACAT-3′;
β-actin reverse 5′-GTCAGGCAGCTCGTAGCTCTT-3′.

## Southern blot analysis for viral DNA

Southern blot analysis was performed as previously described (Cui et al, 2023; Cai et al, 2013). Fourteen days post HBV infection, encapsidated viral DNA was prepared. Samples were resolved in 1% agarose gel. The gel was subjected to denaturation in 0.5 M HCl, then in a solution containing 0.5 M NaOH and 1.5 M NaCl,

followed by neutralization in neutralization buffer (1 M Tris-HCl (pH 7.4) and 1.5 M NaCl). DNA was then blotted onto Hybond-XL membrane (GE Healthcare) in 20× SSC (1× SSC is 0.15 M NaCl plus 0.015 M sodium citrate) buffer. HBV DNA probe labeling and hybridization were carried out by using DIG High Prime DNA Labeling and Detection Stater Kit II (Roche, 11585614910).

### RNA extraction and real-time PCR analysis

Total RNA was extracted with TRIzol Reagent (Tiangen, DP405) according to the manufacturer's protocol, and cDNA was synthesized using the Hifair III 1st Strand cDNA Synthesis SuperMix for qPCR (gDNA digester plus) (Yeasen, 11141ES10). Samples were quantified by qPCR (EZBioscience, A0012-R2).

The amplified PCR products were quantified and normalized using β-actin as a control. Primer sequences are following:
NTCP forward 5′-GGAGGGAACCTGTCCAATGTC-3′;
NTCP reverse 5′-CATGCCAAGGGCACAGAAG-3′;
β-actin forward 5′-GGG AAATCGTGCGTGACAT-3′;
β-actin reverse 5′-GTCAGGCAGCTCGTAGCTCTT-3′.

For HBV pgRNA analysis, HBV2268F: 5′-GAGTGTGGATTCG CACTCC-3′ and HBV2372R: 5′-GAGGCGAGGGAGTTCTTCT-3′ were used.

### ELISA

Culture supernatant of HBV-infected cells was collected at 3-, 5- and 7-day post infection, respectively. The HBeAg levels in the culture supernatant were quantified using an enzyme-linked immunosorbent assay (ELISA) kit (Shanghai Kehua) according to the manufacturer's instructions. The supernatant was mixed with the enzyme conjugate in a reaction plate and incubated at 37 °C for 1 h. After washing, reaction solution A and B were added to the reaction plate and incubated at 37 °C for 30 min. Following the addition of the termination solution, the absorbance was measured.

### Statistics and reproducibility

Statistical analyses were performed using GraphPad Prism 9. Parametric Student *t* test was used. Data are presented as the mean and standard error of the mean. *P* value < 0.05 was considered statistically significant. The experiments were repeated three times. Similar results were obtained.

## Data availability

This study includes no data deposited in external repositories.

The source data of this paper are collected in the following database record: biostudies:S-SCDT-10_1038-S44319-025-00581-8.

## Peer review information

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

## Acknowledgements

We thank Fengmin Lu and Xiangmei Chen (Beijing University, China), Youhua Xie and Qiang Deng (Fudan University, China), and Xinpeng Hong (Rockefeller University, USA) for their helpful suggestions. This work was supported by National Natural Science Foundation of China (32222022, 92354301, 92054104); R&D Program of Guangzhou National Laboratory (GZNL2023A03004); Natural Science Foundation of Shanghai (23ZR1470900); Key Research and Development Program, Ministry of Science and Technology of China (2022YFC2303502).

## Author contributions

**Shuzhi Cui**: Writing—original draft; Project administration. **Wei Gao**: Project administration. **Yuxin Chen**: Resources. **Yi Xu**: Resources. **Zhifang Li**: Data curation; Methodology. **Yu Wei**: Supervision; Writing—review and editing. **Yaming Jiu**: Conceptualization; Supervision; Funding acquisition; Writing—original draft; Writing—review and editing.

Source data underlying figure panels in this paper may have individual authorship assigned. Where available, figure panel/source data authorship is listed in the following database record: biostudies:S-SCDT-10_1038-S44319-025-00581-8.

## Disclosure and competing interests statement

The authors declare no competing interests.

# Expanded View Figures

**Figure EV1.  Effects of exogenous expression of CDC42-CA and CDC42-DN in HepG2-NTCP cells on cell growth and HBV infection.**

(A) Equal numbers of HepG2-NTCP vector, CDC42-CA and CDC42-DN cells were plated in triplicate. Data are shown as fold changes to the vector group. $n = 3$.
(B, C) Determination of the effect of bradykinin ((B), $n = 3$) and ML141 ((C), $n = 5$) on HepG2-NTCP cell viability by cytotoxicity assay. Data are shown as fold changes to the untreated group. (D) HepG2-NTCP cells were treated with bradykinin (100 ng/ml) or ML141 (50 µM) for 10 h. Cells were subjected to GTP-bound GTPase pulldown at indicated time points post treatment. CDC42 GTP-bound forms were analyzed by western blot. (E) Intracellular HBV DNA and secreted HBV virion DNA from HepG2-NTCP cells treated with bradykinin (100 ng/ml) or ML141 (50 µM) were purified at 7 dpi and measured by qPCR. Data are shown as fold changes to the untreated group. $n = 3$. Intracellular HBV DNA of Bradykinin: $P = 0.0002$; Intracellular HBV DNA of ML141: $P = 0.0005$; secreted virions of Bradykinin: $P = 0.0006$; secreted virions of ML141: $P = 0.0007$. (F) cccDNA was purified at 7 dpi and measured by qPCR. Data are shown as fold changes to the untreated group. $n = 3$. cccDNA of Bradykinin: $P = 0.0026$; cccDNA of ML141: $P = 0.0193$. (G) Viral HBeAg in the medium released from infected cells was quantified by ELISA. Data are shown as fold changes to the untreated group. $n = 3$. HBeAg of Bradykinin at 7 days post infection: $P = 0.0001$; HBeAg of Bradykinin at 7 days post infection: $P = 0.0007$; These experiments were repeated three times. Data are represented as mean ± SEM. A.U. Arbitrary Unit, ns no significant difference; $*P < 0.05$; $**P < 0.01$; $***P < 0.001$ (unpaired $t$ test).

▶

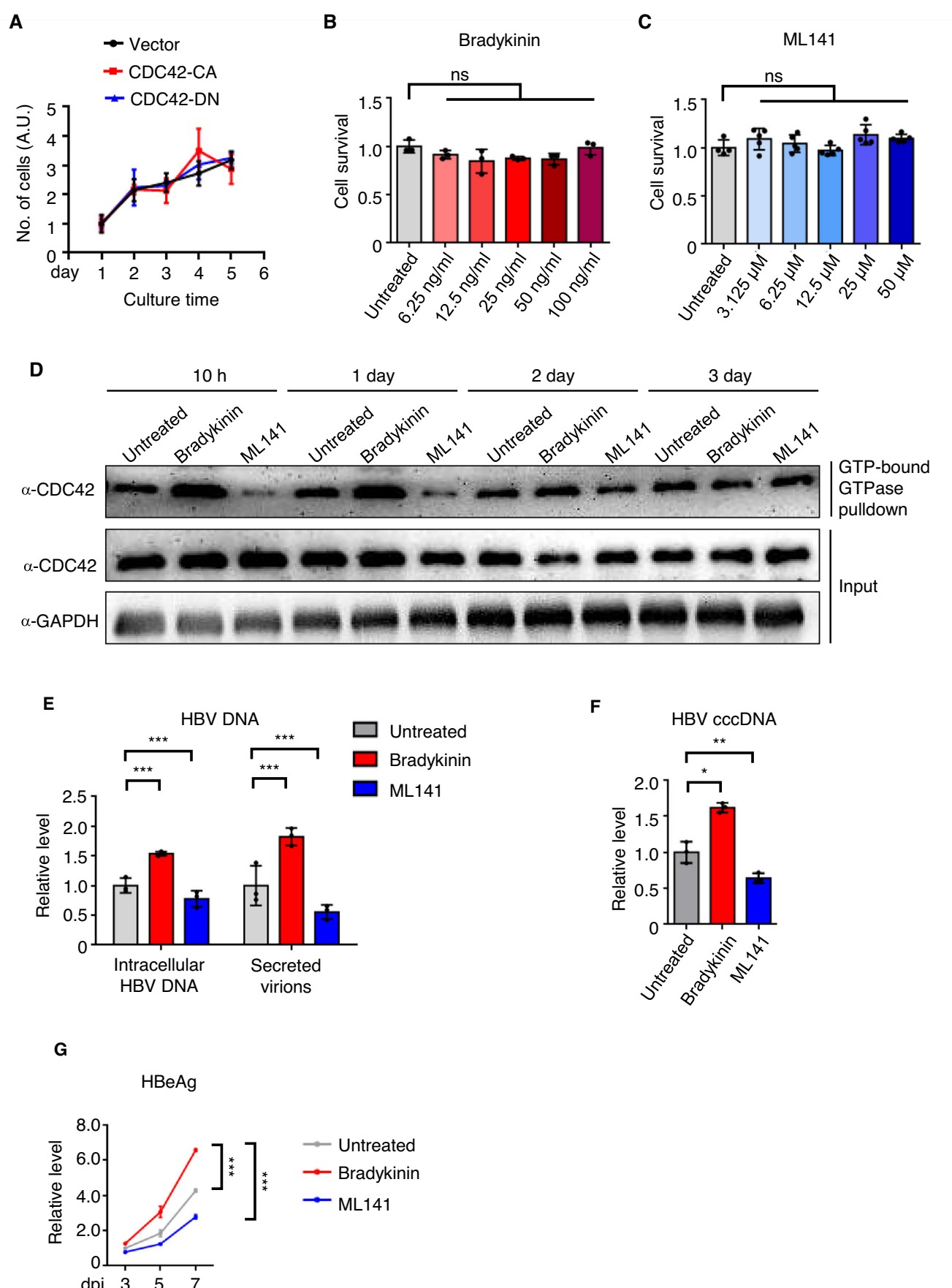

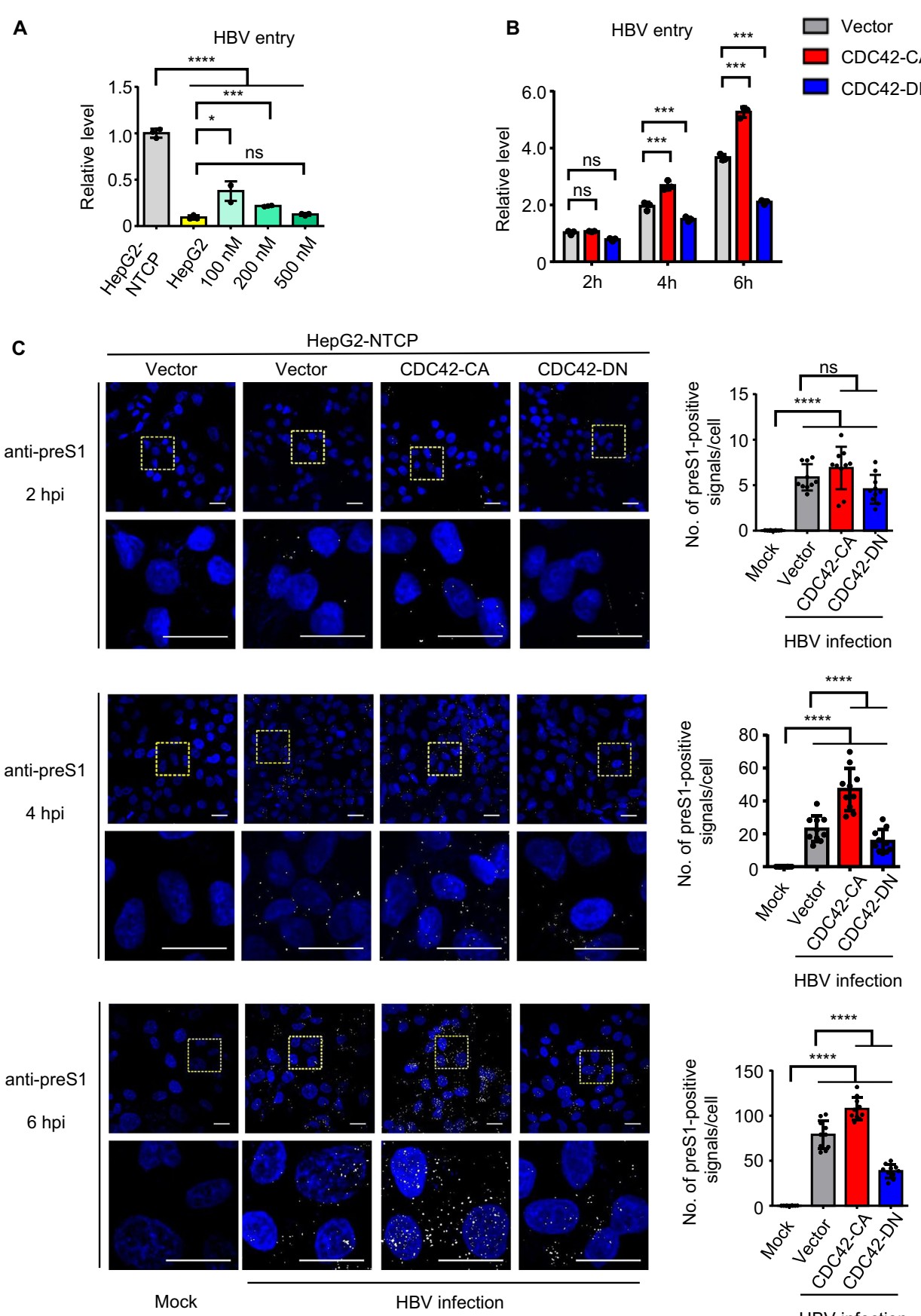

◀ **Figure EV2. Effect of CDC42 on HBV entry and replication.**

(A) HepG2-NTCP cells were infected with HBV in the presence of increasing concentrations of Myrcludex B (MyrB). HepG2 cells were used as negative control for infection. $n = 3$. Data are shown as fold changes to HepG2-NTCP group. HepG2, 100 nM, 200 nM and 500 nM MyrB versus HepG2-NTCP: $P < 0.0001$; 100 nM MyrB versus HepG2: $P = 0.0108$; 200 nM MyrB versus HepG2: $P = 0.0003$. (B) HepG2-NTCP vector, CDC42-CA and CDC42-DN cells were inoculated with heparin-purified HBV for 1 h at 4 °C and then transferred to 37 °C for 2 h, 4 h or 6 h. Intracellular HBV DNA was quantified by qPCR. $n = 3$. Data are shown as fold changes to vector group at 2 hpi. CDC42-CA at 4 h post infection: $P = 0.0009$; CDC42-DN at 4 h post infection: $P = 0.0007$; CDC42-CA at 6 h post infection: $P = 0.0002$; CDC42-DN at 6 h post infection: $P = 0.0001$. (C) Representative immunofluorescence images of HepG2-NTCP uninfected cells (Mock), HepG2-NTCP vector, CDC42-CA and CDC42-DN cells infected with HBV at 2, 4 and 6 hpi. HBV particles were stained with anti-preS1 antibody. Nucleus were stained with DAPI. Quantification of preS1-positive signals per cell is presented. $n = 10$ views (100×/1.5 oil objective). Vector, CDC42-CA and CDC42-DN versus mock at 2, 4 and 6 h post infection: $P < 0.0001$; CDC42-CA and CDC42-DN versus vector at 4 and 6 h post infection: $P < 0.0001$; Scale bar = 10 μm. Data are represented as mean ± SEM. ns, no significant difference; *$P < 0.05$; ***$P < 0.001$; ****$P < 0.0001$ (unpaired $t$ test). Source data are available online for this figure.

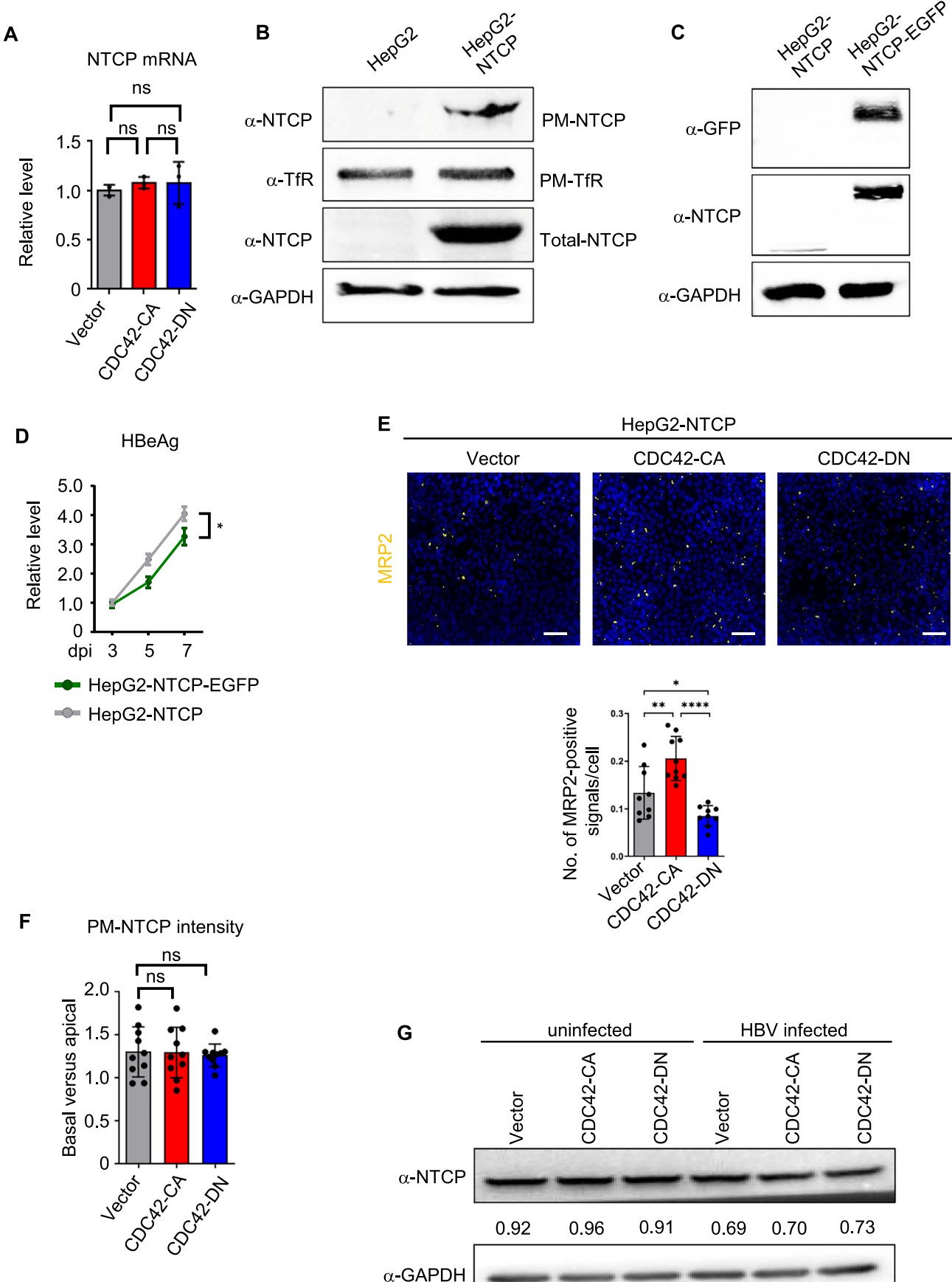

◄ **Figure EV3. Detection of NTCP expression and the effects of NTCP-EGFP fusion protein on HBV replication.**

(A) Quantification of NTCP mRNA by qPCR in HepG2-NTCP vector, CDC42-CA and CDC42-DN cells. Data are presented as fold changes to the vector group. $n = 3$. (B) Plasma membrane (PM) proteins were purified from HepG2 and HepG2-NTCP cells by biotin labeling. PM and total proteins were subjected to Western blotting analysis. The plasma membrane transferrin receptor (PM-TfR) and GAPDH were used as loading control of plasma membrane proteins and total proteins, respectively. (C) Indicated proteins in HepG2-NTCP and HepG2-NTCP-EGFP cells were detected by Western blotting. (D) HepG2-NTCP and HepG2-NTCP-EGFP cells were infected with HBV for 8 h. Culture supernatant was collected at indicated time points for HBeAg analysis by ELISA. These experiments were repeated three times. $n = 3$. HepG2-NTCP-EGFP at 7 days post infection: $P = 0.0231$. (E) Immunostaining of MRP2 in HepG2-NTCP vector, CDC4-CA and CDC42-DN cells. Representative images are presented. MRP2-positive signals were quantified. $n = 10$ views (40×/1.5 air objective). Scale bar = 10 μm. CDC42-CA versus vector: $P = 0.0035$; CDC42-DN versus vector: $P = 0.0264$; CDC42-DN versus CDC42-CA: $P < 0.0001$. (F) NTCP-EGFP fusion protein was quantified at the basal and apical surfaces of HepG2-NTCP-EGFP vector, CDC42-CA and CDC42-DN cells. The values are presented as the ratio of EGFP intensity at the basal surface to that at the apical surface. $n = 10$. (G) Total NTCP expression was detected by Western blot in HepG2-NTCP vector, CDC42-CA and CDC42-DN cell lines infected or uninfected with HBV. Cells were collected at 7 dpi. GAPDH was used as loading control. The ratio of NTCP versus GAPDH is presented. Data are represented as mean ± SEM. ns no significant difference; *$P < 0.05$; **$P < 0.01$; ****$P < 0.0001$ (unpaired $t$ test). Source data are available online for this figure.

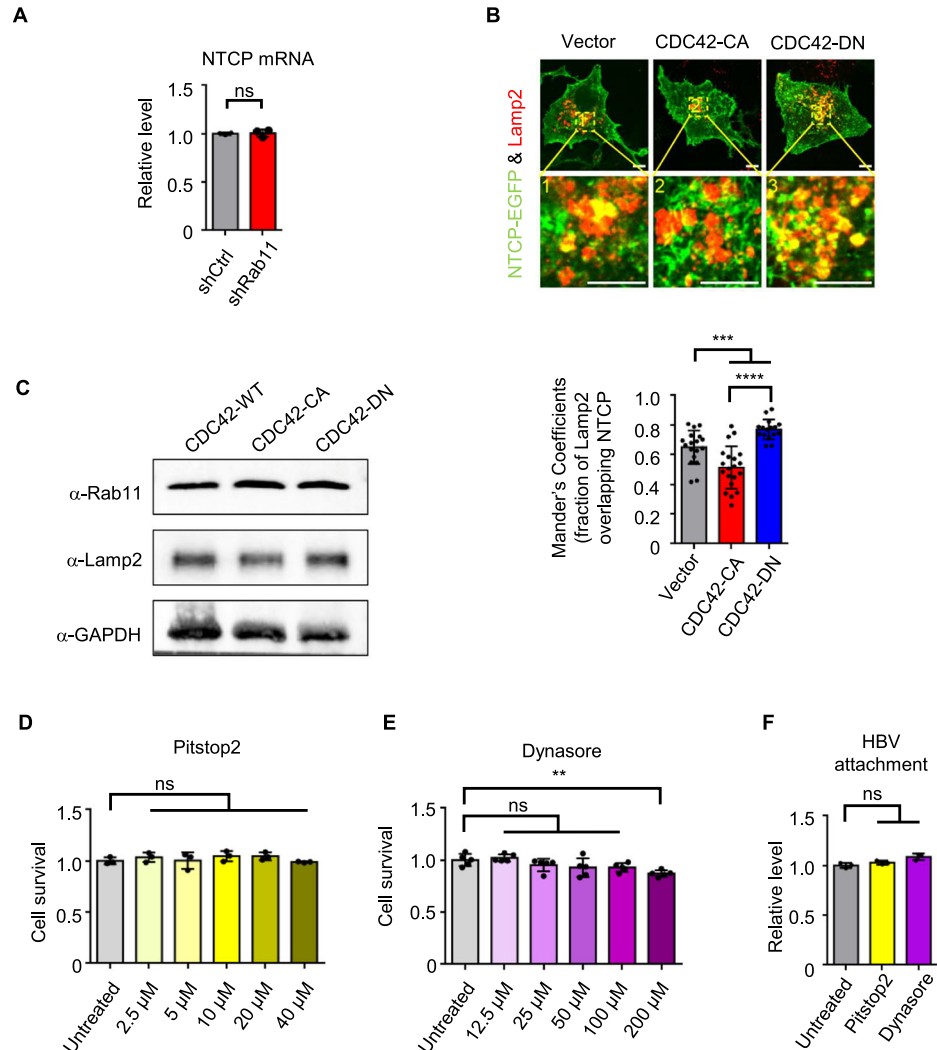

**Figure EV4. Effects of knockdown of Rab11 on NTCP mRNA expression and cytotoxicity assays of Pitstop2 and Dynasore.**

(A) Quantification of NTCP mRNA by qPCR in HepG2-NTCP shControl (shCtrl) and shRab11 cells. Data are shown as fold changes to the shCtrl group. $n = 3$. (B) Representative immunofluorescence images of NTCP-EGFP co-localization with late endosomes/lysosomes in HepG2-NTCP vector, CDC42-CA and CDC42-DN cells. The yellow box indicates the magnified area shown in the corresponding bottom panel. Late endosomes/lysosomes were stained with anti-Lamp2 antibody. Quantification of Mander's coefficients of NTCP and late endosomes/lysosomes. $n = 20$ views (100×/1.5 oil objective). CDC42-CA versus vector: $P = 0.0009$; CDC42-DN versus vector: $P = 0.0008$; CDC42-DN versus CDC42-CA: $P < 0.0001$; Scale bar = 10 μm. (C) Western blotting analysis of indicated proteins from cellular extract of HepG2-NTCP CDC42-WT, CDC42-CA and CDC42-DN cells. GAPDH is used as loading control. (D, E) Cytotoxicity assays of pitstop2 ((D), $n = 3$) and dynasore ((E), $n = 5$) at different concentrations on HepG2-NTCP cell viability. 200 μM dynasore: $P = 0.0024$; Data are shown as fold changes to the untreated group. (F) Quantification of HBV DNA in HepG2-NTCP cells treated with indicated reagents at 3 hpi by qPCR. Data are shown as fold changes to the untreated group. $n = 3$. These experiments were repeated three times. Data are represented as mean ± SEM in all quantification panels. ns, no significant difference; $**P < 0.01$; $***P < 0.001$; $****P < 0.0001$ (unpaired $t$ test). Source data are available online for this figure.

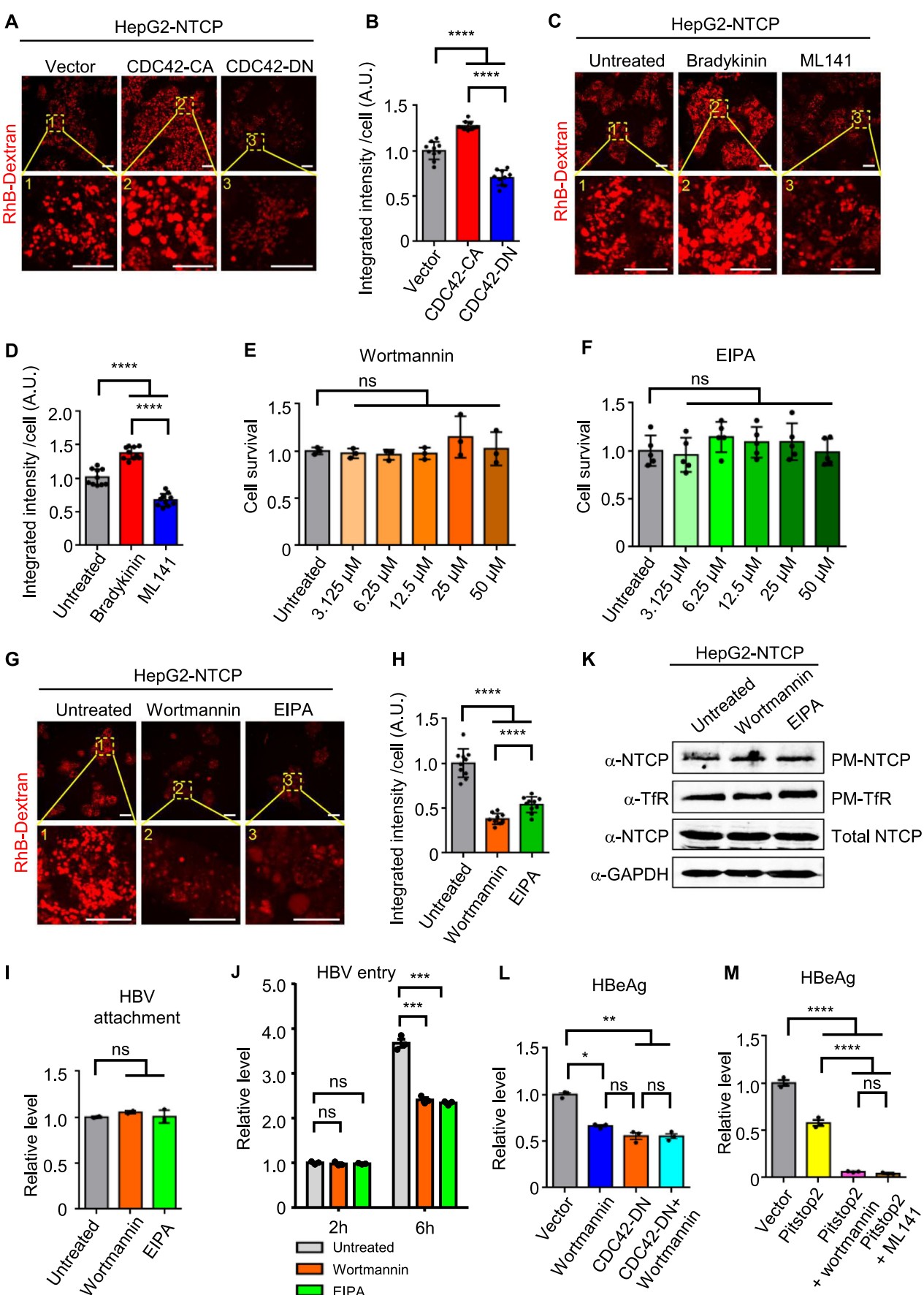

**Figure EV5.  CDC42 regulates macropinocytosis in HBV infection.**

(A) Dextran uptake assay in HepG2-NTCP vector, CDC42-CA and CDC42-DN cells. Cells were incubated with Rhodamine B Dextran (RhB-Dextran) for 1 h and then washed. Dextran staining is shown in red. Scale bar = 10 μm. (B) Quantification of fluorescence intensity of Rhodamine B-Dextran within each cell in A. Data are shown as fold changes to the vector group. $n$ = 15 views (100×/1.5 oil objective). CDC42-CA and CDC42-DN versus vector, CDC42-DN versus CDC42-CA: $P < 0.0001$. (C) Dextran uptake assay in HepG2-NTCP cells treated with bradykinin or ML141. Dextran staining is shown in red. Scale bar = 10 μm. (D) Quantification of fluorescence intensity of Rhodamine B-Dextran within each cell in C. Data are shown as fold changes to the untreated group. n = 15 views (100×/1.5 oil objective). Bradykinin and ML141 versus untreated, ML141 versus Bradykinin: $p < 0.0001$. (E, F) Cytotoxic assays of wortmannin ((E), $n = 3$) and EIPA ((F), $n = 5$) at different concentrations on HepG2-NTCP cell viability. Data are shown as fold changes to the untreated group. (G) Dextran uptake assay in HepG2-NTCP cells treated with wortmannin (50 μM) or EIPA (50 μM). Dextran staining is shown in red. Scale bar = 10 μm. (H) Quantification of fluorescence intensity of Rhodamine B-Dextran within each cell in (G). Data are shown as fold changes to untreated group. $n$ = 15 views (100×/1.5 oil objective). Wortmannin and EIPA versus untreated, EIPA versus Wortmannin: $P < 0.0001$. (I) Quantification of HBV DNA in HepG2-NTCP cells treated with indicated reagents at 3 hpi by qPCR. Data are shown as fold changes to the untreated group. $n = 3$. (J) Quantification of internalized HBV DNA in HepG2-NTCP cells untreated or treated with wortmannin (50 μM) or EIPA (50 μM) at 2 hpi and 6 hpi by qPCR. Data are shown as fold changes to the untreated group at 2 hpi. $n = 3$. Wortmannin at 6 hpi: $P = 0.0003$; EIPA at 6 hpi: $P = 0.0002$. (K) Plasma membrane proteins were purified from HepG2-NTCP cells treated with wortmannin (50 μM) and EIPA (50 μM) by biotin labeling. PM and total proteins were subjected to Western blotting analysis. The plasma membrane transferrin receptor (PM-TfR) and GAPDH were used as loading control of plasma membrane proteins and total proteins, respectively. (L, M) Culture supernatant from HBV-infected HepG2-NTCP vector or HepG2-NTCP CDC42-DN cells treated with indicated reagents was collected at 7 dpi. HBeAg was quantified by ELISA. Data are shown as fold changes to the vector group. These experiments were repeated three times. $n = 3$. Wortmannin versus vector: $P = 0.0175$; CDC42-DN versus vector: $P = 0.0012$; CDC42-DN + wortmannin versus vector: $P = 0.0010$; Pitstop2, Pitstop2 + Wortmannin and Pitstop2 + ML141 versus vector, Pitstop2 + Wortmannin and Pitstop2 + ML141 versus Pitstop2: $P < 0.0001$; Data are represented as mean ± SEM. A.U. Arbitrary Unit, ns no significant difference; $*P < 0.05$; $**P < 0.01$; $***P < 0.001$, $****P < 0.0001$ (unpaired $t$ test). Source data are available online for this figure.

