## [Peer Review File · EMBO Reports]

CDC42 supports hepatitis B virus entry by fostering NTCP to plasma membrane and macropinocytosis

Yaming Jiu, Shuzhi Cui, Wei Gao, YuXin Chen, Zhifang Li, Yu Wei, and Yi Xu

Corresponding author(s): Yaming Jiu (ymjiu@siii.cas.cn) , Yu Wei (yu.wei@pasteur.fr)

Review Timeline:

Submission Date:	10th Dec 24
Editorial Decision:	24th Feb 25
Revision Received:	30th May 25
Editorial Decision:	9th Jul 25
Revision Received:	17th Jul 25
Accepted:	25th Aug 25

Editor: Deniz Senyilmaz Tiebe

Transaction Report:

Dear Yaming,

Thank you for submitting your research manuscript to our journal, which was now seen by four referees, whose reports are copied below. Please accept my apologies for the delay in getting back to you, it took longer than anticipated to receive the referee reports, as previously communicated.

Referees express interest in the proposed role of CDC42 in hepatitis B virus entry. However, they also raise concerns that need to be addressed for publication here. In particular,

- Purity/status of the viral inoculum (viral vs. subviral particles) is currently unclear and this may influence the results (referees #1 and #2)
- Some of the key findings need to be replicated in primary human hepatocytes (PHHs) (referee #2).
- The effect of CDC42 mutant form on cell polarity and number of HBV positive cells needs to be explored (referee #3)
- Existing data currently cannot differentiate virus taken up and virus bound to the membrane (referee #4).
- Referees #1, #2 and #4 raise concerns regarding anti-preS1 stainings.
- Up or down-regulation using plasmid transfection with an HBV1.3 plasmid would strengthen the study (referee #4).
- Referees raise concern about data presentation and missing controls.

Should you be able to address these concerns, we would like to invite you to revise your manuscript with the understanding that the referee concerns (as in their reports) must be fully addressed and their suggestions taken on board.

I would be happy to discuss your plans on how to address these points with a video chat. Should that be of interest for you, please respond to this email to set an appointment for such a video chat.

Please address all referee concerns in a complete point-by-point response. Acceptance of the manuscript will depend on a positive outcome of a second round of review. It is EMBO reports policy to allow a single round of major experimental revision only and acceptance or rejection of the manuscript will therefore depend on the completeness of your responses included in the next, final version of the manuscript.

We realize that it is difficult to revise to a specific deadline. In the interest of protecting the conceptual advance provided by the work, we recommend a revision within 3 months. Please discuss the revision progress ahead of this time with me if you require more time to complete the revisions, or if you have questions or comments regarding the revision (also by video chat).

1. A data availability section providing access to data deposited in public databases is missing (where applicable).
2. Your manuscript contains statistics and error bars based on $n=2$. Please use scatter plots in these cases.

You can submit the revision either as a Scientific Report or as a Research Article. For Scientific Reports, the revised manuscript can contain up to 5 main figures and 5 Expanded View figures, and it should not exceed 27000 characters. If the revision leads to a manuscript with more than 5 main figures it will be published as a Research Article. In this case the Results and Discussion section should be separate. If a Scientific Report is submitted, these sections have to be combined. This will help to shorten the manuscript text by eliminating some redundancy that is inevitable when discussing the same experiments twice. In either case, all materials and methods should be included in the main manuscript file.

<<https://www.embopress.org/page/journal/14693178/authorguide#expandedview>>

4) a .docx formatted letter INCLUDING the reviewers' reports and your detailed point-by-point responses to their comments. As part of the EMBO publication's Transparent Editorial Process, EMBO reports publishes online a Review Process File (RPF) to accompany accepted manuscripts. This File will be published in conjunction with your paper and will include the referee reports, your point-by-point response and all pertinent correspondence relating to the manuscript.

<https://www.embopress.org/page/journal/14693178/authorguide#transparentprocess>

5) a complete author checklist, which you can download from our author guidelines

<https://www.embopress.org/page/journal/14693178/authorguide>. Please insert information in the checklist that is also reflected in the manuscript. The completed author checklist will also be part of the RPF.

6) Please note that all corresponding authors are required to supply an ORCID ID for their name upon submission of a revised manuscript (<<https://orcid.org/>>). Please find instructions on how to link your ORCID ID to your account in our manuscript tracking system in our Author guidelines

<<https://www.embopress.org/page/journal/14693178/authorguide#authorshipguidelines>>

7) Before submitting your revision, primary datasets produced in this study need to be deposited in an appropriate public database (see <https://www.embopress.org/page/journal/14693178/authorguide#datadeposition>). Please remember to provide a reviewer password if the datasets are not yet public. The accession numbers and database should be listed in a formal "Data Availability" section placed after Materials & Method (see also

<https://www.embopress.org/page/journal/14693178/authorguide#datadeposition>). Please note that the Data Availability Section is restricted to new primary data that are part of this study. * Note - All links should resolve to a page where the data can be accessed. *

Additional information on source data and instruction on how to label the files are available:

<https://www.embopress.org/page/journal/14693178/authorguide#sourcedata>

9) Our journal encourages inclusion of *data citations in the reference list* to directly cite datasets that were re-used and obtained from public databases. Data citations in the article text are distinct from normal bibliographical citations and should directly link to the database records from which the data can be accessed. In the main text, data citations are formatted as follows: "Data ref: Smith et al, 2001" or "Data ref: NCBI Sequence Read Archive PRJNA342805, 2017". In the Reference list, data citations must be labeled with "[DATASET]". A data reference must provide the database name, accession number/identifiers and a resolvable link to the landing page from which the data can be accessed at the end of the reference. Further instructions are available at <http://www.embopress.org/page/journal/14693178/authorguide#referencesformat>

10) Regarding data quantification (see Figure Legends:

<https://www.embopress.org/page/journal/14693178/authorguide#figureformat>)

- the name of the statistical test used to generate error bars and P values,

- the number (n) of independent experiments (please specify technical or biological replicates) underlying each data point,

- the nature of the bars and error bars (s.d., s.e.m.),

- If the data are obtained from n Program fragment delivered error ``Can't locate object method "less" via package "than" (perhaps you forgot to load "than"?) at //ejpvfs23/sites23b/embor_www/letters/embor_decision_revise_and_review.txt line 56.' 2,

use scatter blots showing the individual data points.

12) Please also note our reference format:

13) All Materials and Methods need to be described in the main text using our 'Structured Methods' format, which is required for all research articles. According to this format, the Methods section includes a Reagents and Tools Table (listing key reagents, experimental models, software and relevant equipment and including their sources and relevant identifiers) followed by a Methods and Protocols section describing the methods using a step-by-step protocol format. The aim is to facilitate adoption of the methodologies across labs. More information on how to adhere to this format as well as a downloadable template (.docx) for the Reagents and Tools Table can be found in our author guidelines:

I look forward to seeing a revised version of your manuscript when it is ready. Please let me know if you have questions or comments regarding the revision.

Kind regards,

Deniz Senyilmaz Tiebe

Deniz Senyilmaz Tiebe, PhD
Senior Scientific Editor
EMBO Reports

Referee #1:

Cui and colleagues study the role of CDC42 in hepatitis B virus entry and conclude a role for the regulation of micropinocytosis-dependent internalization. When conducting virus internalization and trafficking studies it's essential to consider the purity of the viral inoculum and how this may influence the results. On reading the Methods section the authors used a PEG concentrated stock of HBV for their experiments that will by definition aggregate the particles and this is likely to bias the results to favor macropinocytic uptake pathways.

General comment: to ensure replicate sample numbers and independent experiments are listed in figure legends. Overlaid symbols on bar charts are difficult to discern and suggest increasing their size.

Specific comments on the experimental design and conclusions are listed below:

Fig.1. In panel (A) the authors state that CDC42-CA expressing cells show a significant increase in GTP-bound CDC42 and the inverse phenotype with CDC42-DN cells. However, no information is provided on the replicate sample numbers that were used for quantitative analysis or stat test used to support this conclusion. Visual inspection of the blots does not support this conclusion.

Fig.2. Suggest authors present HBV cell binding data that is currently plotted as supp in main figure as this supports a role for CDC42 in the viral internalization process. The authors currently use an 8h inoculation and trypsinization protocol to measure internalized HBV particles by qPCR measurement of HBV DNA presented in (A). 8h is a long time window in which to study virus

trafficking and this is particularly relevant given the asynchronous nature of the infection. Recommend using a synchronised temp-shift protocol as published in Chakraborty 2020 DOI: 10.1111/cmi.13250 and to perform time-dependent assays with heparin purified HBV that would increase the robustness of the data and define the role of CDC42 in regulating HBV internalization rate. (C/E) Similar critique to single 8h time window to image intracellular preS1 as evidence of internalization in asynchronous infection assay.

Fig.3/4: for all blots please ensure replicate values are provided in figure legend and how quantitative values are determined. Panels (B/D) the authors need to evaluate the functional activity of NTCP-EGFP in terms of bile acid uptake or HBV infection as this large fusion protein may show different localization and trafficking properties to the parental protein. Of note when the authors show HBV infection data they use HepG2-NTCP cells and not those expressing NTCP-EGFP and yet the data sets are presented back to back in the same figures prompting the reader to compare the read-outs. The authors could have used a fluorescent labelled preS1 to track NTCP as previously reported.

Fig.5. (E) Did the authors evaluate the antiviral activity of Dynasore and Pitstop in the CDC42-DN line this may provide more reliable data given the potential off-target effects of ML141.

Fig.6. Earlier critique of HBV internalization protocol mentioned above applies here. Encourage the authors to read the Chakraborty 2020 DOI: 10.1111/cmi.13250 paper that showed limited evidence for a macropinocytotic uptake of purified HBV particles in two independent hepatoma cell lines.

Referee #2:

CDC42 supports 1 hepatitis B virus entry by fostering NTCP plasma membrane location and macropinocytosis-dependent internalization
Cui-S et al.,

The authors used the HBV-susceptible HepG2-NTCP cell line to investigate modes of HBV binding and entry.

Key Findings:

CDC42 positively regulates HBV infection through two distinct mechanisms
Promoting NTCP receptor trafficking to the plasma membrane
Enhancing macropinocytosis-mediated viral internalization

Points of concern.

The authors used only the human hepatoma cell line HepG2, artificially overexpressing NTCP, while HBV is adapted to highly differentiated hepatocytes, relying both on differentiation-dependent cellular factors for entry (NTCP) and expression of episomal cccDNA (hepatic nuclear factors, e.g. HNF4) for infection and replication. Either the authors use primary human hepatocytes (PHH) to show the observed effects in a differentiated cell, known as the gold standard for HBV binding and infectivity in vitro, or they should extend the title by including the cell line HepG2 in the title.

Fig. 1b: the higher replicative intermediates for HBV-infected HepG2-NTCP cells 12 days after the infection, still overexpressed CDC42-CA could be derived from either increased HBV uptake, but also from increased activity of cccDNA (e.g. increased transcription, see increased HBeAg expression in Figure 1e). How do the authors really distinguish between these things in experimental results?

Even the little increased amount of cccDNA in presence of constitutively active (CA) CDC42 (and decrease during presence of dominant-negative CDC42) shown in Fig. 1d could be a result of increased or decreased activity of cellular repair mechanism in the nucleus acting on incoming HBV rcDNA after uptake (via whichever HBV route of uptake). This is not clearly shown in the manuscript.

Why did the authors have not used an inducible expression vector of regulated expressing CDC42 and its variants? Using this kind of regulation, the authors could have activated the CDC42 activity during infection, and shut it down after the infection (8 hours) till analysis of HBV replicative intermediates 12 days later (and vice versa)?

The effects of chemical activators (bradykinin) or inhibitors (ML141) on HBV replicative forms (fig. 1g), secreted virions (fig. 1h) and HBV cccDNA (fig. 1i) for only 10 hours during infection is a nice approach to differentiate between uptake of HBV and replication of HBV under CDC42, but the controls are missing that it has no effect on HBV replication after 12 days, the day the above mentioned assays on HBV replication have been done. In this regard, the figure 1f is not convincing. An upper band, half visible (but half cut) in the untreated anti-CDC42 control is cut in the Bradykinin- and ML141-treated bands. Is it not there? What is it? The authors should explain and show larger section of the blot here.

Fig. 2c-f and Fig. 6b/c: the authors used an anti-preS1 to show viral particles within cells using immunofluorescence. This statement cannot be made. The preS1-domain is part of the large HBV surface protein, LHBs and is indeed essential for HBV binding of hepatocytes via the NTCP. But the LHBs is also present in HBV subviral particles that are present in large excess both in sera of HBV-infected patients and also in any HBV-containing inoculate from cell culture derived HBV that is not purified by ultracentrifugation via density gradient, e.g. sucrose. Did the authors distinguish between HBV virions and subviral particles in

their uptake assay using preS1-specific antibodies in IF? How can the authors argue that the preS1-dependent intracellular signal is derived from viral particles and not subviral particles? According to the material and methods section, the authors used HepAD38 cells that produce HBV but also large excess of subviral particles. The authors did not purify the viral particles in a way to separate virions from subviral particles, but used PEG-precipitation and low-speed centrifugation to pellet the virions-subviral particle precipitates. From the solubilized precipitates, only virions were quantified, but not the amount of subviral particles that contaminate the preparation and will for sure affect the preS1-dependent assay on virus (and subviral particle) uptake. The authors should determine the amount of preS1-containing subviral particles in their preparation and the ratio of virions to subviral particles in the inoculum.

Minor:

Spelling:

Line 34: Rab11-dependent, instead of Rab11-depedent.

Citation:

Line 185: A citation is needed for the argument that regulation of NTCP mRNA is mainly achieved by bile acids.

Referee #3:

In their manuscript entitled "CDC42 supports hepatitis B virus entry by fostering NTCP plasma membrane location and macropinocytosis-dependent internalization" Shuzi Cui and coworker investigate the impact of CDC42 on HBV life cycle. They describe that CDC42 is a prominent factor affecting HBV entry. In this context they provide evidence that CDC42-dependent macropinocytosis is relevant for HBV entry.

The authors present interesting observations but at the present stage of the manuscript there are still a number of open points, which must be addressed.

Specific points:

Is there a bias in the chosen assay system as GTP-loaded and GDP-loaded CDC42 differ with respect to their localization and due to association with membranes and cytoskeleton components in their solubility.

The authors should include data describing the impact of CDC42-CA and CDC42-DN on cell polarity.

Information about the mutations leading to CA or DN-activity of CDC42 should be included in the manuscript.

The authors should provide a quantification of fig 1a (GTP-bound CDC42).

IF or FACS data should be included investigating the impact of CDC42-CA or CDC42-DN overexpression on the number of HBV positive cells.

line 140: "Together, these data indicate that activation of CDC42 signaling facilitates HBV infection." At this point of the analysis this statement is an overinterpretation. Here the authors can conclude that CDC42 activation favours HBV replication fig2A. It is amazing that a moderate change in the level of GTP versus -GDP-bound CDC42 (fig1) is associated with a comparable on HBV entry effect as myrcludex. The authors should comment on this.

What is the impact of CDC42-CA and CDC42-DN on taurocholate/bile acid uptake?

The authors provide evidence that CDC42-CA leads to an enhanced localization of NTCP on the plasma membrane. In differentiated cells NTCP-localization shows polarity-is this observed in CDC42-CA overexpressing cells?

HBV-infection is characterized by "receptor down modulation" in infected cells. Is this affected by CDC42-activation?

The colocalization of CDC42 with Rab11 should be characterized in more detail with respect to the subcellular compartment(s) in which both proteins colocalize.

What is the impact of CDC42-CA/DN in the experimental setting described here on membrane composition, lipid raft formation, especially cholesterol content and cholesterol uptake?

Minor points :

There are some typing errors and style and grammar could be slightly modified.

Referee #4:

The manuscript presents compelling evidence that the level of active CDC42 in hepatocytes positively correlates with the entry capacity of the hepatitis B virus (HBV). This occurs via the promotion of the transport of the viral receptor, sodium taurocholate co-transporting polypeptide (NTCP), to the plasma membrane through a CDC42-driven, Rab11-dependent recycling endosomal

pathway. Additionally, it demonstrates that micropinocytosis is essential for viral uptake.

General Comments:

The experiments are generally well designed, but some controls are needed and further experiments should be included to confirm the claims.

It is widely recognized that high multiplicities of infection (MOIs) of HBV are required in cell culture to achieve infection of a small number of cells with minimal amounts of covalently closed circular DNA (cccDNA), indicating that only a few particles reach the nucleus and convert relaxed circular DNA (rcDNA) to cccDNA. This suggests that most HBV particles taken up by cells do not lead to actual infection. It would be interesting to consider which entry pathways result in uptake and which lead to real infection. Observing that HepG2 cells without NTCP expression cannot be infected by HBV at all, likely due to reduced uptake and membrane fusion, the observed HBV uptake in these cells requires further explanation.

Major Comments:

Figure 1:

What is the transduction rate of the HepG2 cells? A Flag staining of the cells would be beneficial since the CDC42 constructs express a Flag-tag. The authors could then show flag staining in parallel to HBV-PreS1 staining of positively and not-transfected cells next to each other.

Southern blot should be labeled with each individual band correctly rather than "replication forms".

Explanation is needed for the significant difference between the CA and DN controls in Fig. 1b, while Fig. 1d only shows a three-fold difference.

Figure 2:

While claiming the quantification of HBV entry, it is essential to discriminate between virus taken up and virus bound to the membrane. How do the authors differentiate this just by using PCR?

In Fig. 2c, the fluorescence image of anti-PreS1 labeled particles shows PreS1 in the nucleus, which is unexpected since PreS1 is a membrane protein. Confocal microscopy should be used to show PreS1 staining in the endosomes and not the nucleus.

A heparin control should be included since it would prevent PreS1 attachment and thus any particle uptake.

In Fig. 2g, HepG2 cells do not take up any HBV compared to HepG2-NTCP cells. This result needs clarification, given that MyrB-treated HepG2-NTCP should yield similar results in HBV uptake, which seems to contradict to Fig. 2b.

Figure 3:

The quantification of the Western blot in the figure is unclear. It should be clarified to which blot the numbers 0.46, 0.67, etc., correspond, as they do not seem to correlate with PM-NTCP or PM TfR.

Overall, the claim that CDC42 and Rab11 inhibit HBV uptake is supported by showing HBV infection parameters, Southern blot, and qPCR. Including experiments for up or down-regulation using plasmid transfection with an HBV1.3 plasmid would strengthen the study. Since HBV cannot spread in cell culture, all experiments should show no effect on the HBV parameters for HBV markers. Another potential experiment could be the knockdown after HBV infection at 1 dpi. Since this should not show any reduction in infectio parameters. Same could be done by using HBV expressing cell lines (HepG2-HBV, HepAD38).

Dear Deniz,

Thank you very much for handling our manuscript (MS# EMBOR-2024-60966V1) entitled "CDC42 supports hepatitis B virus entry by fostering NTCP to plasma membrane and macropinocytosis". We appreciate all the critical comments and suggestions from you and four reviewers. We have revised the manuscript accordingly. Please find all the specific changes in point-by-point answers.

We highlight all the modifications with red color in the revised version of the manuscript and resubmit it along with a clean version of the revised manuscript. With these improvements, we sincerely hope that the revised version of our manuscript meets the journal's criteria for publication.

Sincerely,
Yaming Jiu, PhD, Professor,
Shanghai Institute of Immunity and Infection,
Chinese Academy of Sciences

Referee #1:

Cui and colleagues study the role of CDC42 in hepatitis B virus entry and conclude a role for the regulation of macropinocytosis-dependent internalization. When conducting virus internalization and trafficking studies it's essential to consider the purity of the viral inoculum and how this may influence the results. On reading the Methods section the authors used a PEG concentrated stock of HBV for their experiments that will by definition aggregate the particles and this is likely to bias the results to favor macropinocytic uptake pathways.

Response:

For the revision, we have prepared HBV stocks with heparin HiTrap columns according to Chakraborty et al (PMID: 32799415). We have used heparin-prepared HBV particles to infect HepG2-NTCP cells treated with wortmannin and EIPA (Fig. R1-1). We did not observe any discrepancies in the data from the experiments using PEG-prepared HBV particles. We present these results in the revised Fig. EV5J. We have added heparin purification in

the Methods.

Fig. R1-1. Heparin HiTrap columns were used to prepare HBV stock. HepG2-NTCP cells treated with macropinocytosis inhibitors wortmannin or EIPA were infected with heparin-prepared HBV particles. Cells were collected at indicated time points post-infection. HBV DNA was analyzed by qPCR. This figure is presented as revised Fig. EV5J.

We have performed all the infection experiments in primary human hepatocytes (PHHs) with heparin-purified HBV stocks. We have not observed any discrepancy in results compared with PEG-purified HBV stocks. We have specified this part in the Methods (chapter HBV production and titration).

General comment: to ensure replicate sample numbers and independent experiments are listed in figure legends. Overlaid symbols on bar charts are difficult to discern and suggest increasing their size.

Response:

We have added all the replicate sample numbers and independent

experiments in the figure legends of the revised manuscript. We have increased the size of the texts (Helvetica font, size 10 pt) in the figures.

Specific comments on the experimental design and conclusions are listed below:

Fig.1. In panel (A) the authors state that CDC42-CA expressing cells show a significant increase in GTP-bound CDC42 and the inverse phenotype with CDC42-DN cells. However, no information is provided on the replicate sample numbers that were used for quantitative analysis or stat test used to support this conclusion. Visual inspection of the blots does not support this conclusion.

Response:

We performed independently the experiments three times (Fig. R1-2). The quantification was performed using Image J. The middle image of Fig. R1-2 is presented as revised Fig. 1A. We have added in the legend “The ratio of the GTP-bound active CDC42 versus GAPDH is presented. The experiment was repeated three times and one representative result is displayed.”

Fig. R1-2. Detection of active CDC42 was assayed by GTP-bound GTPase pulldown and Western blotting. GAPDH was used as loading control. The ratio of the GTP-bound active CDC42 versus GAPDH is presented. Three independent experiments are presented.

Fig.2. Suggest authors present HBV cell binding data that is currently plotted as supp in main figure as this supports a role for CDC42 in the viral internalization process.

Response:

We accept this suggestion. We have presented HBV attachment data in revised Fig. 2A,B.

The authors currently use an 8 h inoculation and trypsinization protocol to measure internalized HBV particles by qPCR measurement of HBV DNA presented in (A). 8h is a long time window in which to study virus trafficking and this is particularly relevant given the asynchronous nature of the infection. Recommend using a synchronized temp-shift protocol as published in Chakraborty 2020 DOI: 10.1111/cmi.13250 and to perform time-dependent assays with heparin purified HBV that would increase the robustness of the data and define the role of CDC42 in regulating HBV internalization rate.

Response:

We have used the synchronized temp-shift protocol (PMID: 32799415) and

heparin-purified HBV particles to perform HBV internalization assays (Fig. R1-3). Activation of CDC42 significantly increased virus entry as early as 4 h post-infection, whereas virus entry was inhibited in CDC42-DN cells with similar timing. These data are consistent with the conclusion in the initial submission. We have presented this result in Fig. EV2B. We have added in the Methods (chapter HBV production and titration) that we compared PEG-purified particles with heparin-purified particles infecting HepG2-NTCP cells in HBV internalization assays and found no significant difference.

Therefore, we did not specify the method in HBV stock preparation in the description of each infection experiment.

Fig. R1-3. The synchronised temp-shift protocol (Carkraborty et al, 2020 DOI: 10.1111/cmi.13250) was used to conduct internalization assays with heparin-purified HBV particles. HBV DNA was quantified by qPCR at indicated time points post-infection. This figure is presented as Fig. EV2B in the revised version.

(C/E) Similar critique to single 8 h time window to image intracellular preS1 as evidence of internalization in asynchronous infection assay.

Response:

We have used the synchronized temp-shift protocol (PMID: 32799415) and heparin-purified HBV particles to image intracellular preS1 (Fig. R1-4). The results are presented in Fig. EV2C.

Fig. R1-4. The synchronised temp-shift protocol (Carkraborty et al, 2020 DOI: 10.1111/cmi.13250) was used to conduct internalization assays with heparin-purified HBV particles in HepG2-NTCP vector, CDC42-CA and CDC42-DN cells. HBV particles were stained with anti-preS1 antibody at indicated time points. Quantification of preS1-positive signals is presented. This figure is presented as Fig. EV2C in the revised version.

Fig.3/4: for all blots please ensure replicate values are provided in figure legend and how quantitative values are determined.

Response:

We have added the replicate values in the figure legends and details of WB quantification in the revised Methods (chapter Western blot).

Panels (B/D) the authors need to evaluate the functional activity of NTCP-EGFP in terms of bile acid uptake or HBV infection as this large fusion protein may show different localization and trafficking properties to the parental protein. Of note when the authors show HBV infection data they use HepG2-NTCP cells and not those expressing NTCP-EGFP and yet the data sets are presented back to back in the same figures prompting the reader to compare the read-outs.

Response:

HepG2-NTCP-EGFP cells were generated based on the methodology described in Stross et al. (PMID: 20539008), which demonstrated that the NTCP-EGFP fusion protein retains bile acid transport activity comparable to wild-type NTCP. We have added this point in the text (Results section, chapter CDC42 fosters the recruitment of NTCP to the cell surface).

For the revision, we have infected both HepG2-NTCP and HepG2-NTCP-EGFP cells with HBV with same MOI and measured HBeAg levels in the supernatant. ELISA results revealed that HBeAg levels in HepG2-NTCP-EGFP cells were slightly lower than those in HepG2-NTCP cells (Fig. R1-5). These results indicate that although NTCP-EGFP shows slightly reduced receptor activity, it can still be used as a surrogate to study NTCP in HBV infection. We have updated these results in the revised Fig. EV3D.

We have specified the cell lines in all the figures in Fig. 4.

Fig. R1-5. HBeAg released from infected HepG2-NTCP-EGFP and HepG2-NTCP cells was quantified by ELISA. This figure is presented as Fig. EV3D in the revised version.

The authors could have used a fluorescent labelled preS1 to track NTCP as previously reported.

Response:

We agree that fluorescent labelled preS1 can be used as a tool to track NTCP. In the current study, we opted for more direct tracking methods for NTCP. We used NTCP-EGFP fusion for imaging NTCP and immunoblotting with anti-NTCP antibody to detect NTCP protein expression on the plasma

membrane.

Fig.5. (E) Did the authors evaluate the antiviral activity of Dynasore and Pitstop in the CDC42-DN line, this may provide more reliable data given the potential off-target effects of ML141.

Response:

We have evaluated the antiviral activity of Dynasore and Pitstop in HepG2-NTCP CDC42-DN cells by analyzing HBeAg levels (Fig. R1-6A). We have also treated HepG2-NTCP cells with Dynasore and Pistop, and CDC42 inhibitor ML141 (Fig. R1-6B). These data support the conclusion that CDC42-mediated HBV internalization and clathrin-mediated endocytic uptake of HBV represent independent pathways for HBV entry.

Fig. R1-6. (A) HBeAg was analyzed by ELISA in HepG2-NTCP CDC42-DN cells treated with CME inhibitors Pitstop2 or Dynasore. This figure is presented as Fig. 6F in the revised version. (B) HepG2-NTCP cells were treated with CME inhibitors, then treated with ML141 and infected with HBV. HBeAg was quantified by ELISA. This figure is presented as Fig. 6H in the revised version.

Fig.6. Earlier critique of HBV internalization protocol mentioned above applies here. Encourage the authors to read the Chakraborty 2020 DOI: 10.1111/cmi.13250 paper that showed limited evidence for a macropinocytic uptake of purified HBV particles in two independent hepatoma cell lines.

Response:

We think that the discrepancies in macropinocytosis results between PMID: 32799415 and the current study may stem from variations in the duration of the viral entry analysis and the duration of drug treatments. Chakraborty et al analyzed viral entry at 6 hpi, while we did this analysis at 8 hpi. We pre-treated cells with wortmannin or EIPA for 2 h and continued to treat cells during HBV inoculation. We have discussed this point in the Discussion section.

Referee #2:

CDC42 supports hepatitis B virus entry by fostering NTCP plasma membrane location and macropinocytosis-dependent internalization Cui-S et al., The authors used the HBV-susceptible HepG2-NTCP cell line to investigate modes of HBV binding and entry.

Key Findings:

CDC42 positively regulates HBV infection through two distinct mechanisms

Promoting NTCP receptor trafficking to the plasma membrane

Enhancing macropinocytosis-mediated viral internalization

Points of concern.

The authors used only the human hepatoma cell line HepG2, artificially overexpressing NTCP, while HBV is adapted to highly differentiated hepatocytes, relying both on differentiation-dependent cellular factors for entry (NTCP) and expression of episomal cccDNA (hepatic nuclear factors, e.g. HNF4) for infection and replication. Either the authors use primary human hepatocytes (PHH) to show the observed effects in a differentiated cell, known as the gold standard for HBV binding and infectivity in vitro, or they should extend the title by including the cell line HepG2 in the title.

Response:

For the revision, we have purchased primary human hepatocytes (PHHs) and performed HBV entry experiments with PHHs. Treatment of PHHs with CDC42 activator bradykinin led to the increased levels of HBV entry, cccDNA, intracellular HBV DNA and HBeAg, whereas treatment of PHHs with CDC42 inhibitor ML141 decreased the levels of HBV entry, cccDNA, intracellular HBV DNA and HBeAg (Fig. R2-1). These results are in agreement with those from HepG2-NTCP cells. These data are presented in the revised Fig. 1I, 1H, 1J and 2E. We have added PHH experiments in the Methods.

Fig. R2-1. Primary human hepatocytes (PHHs) were treated with bradykinin or ML141 and infected with heparin-purified HBV particles. Cells were collected at 8 hpi for HBV internalization (A), at 3 dpi for cccDNA analysis (B) and at 7 dpi for intracellular HBV DNA analysis (C) by qPCR. Culture supernatant was quantified for HBeAg by ELISA at indicated time points (D). In the revised version, A is presented as Fig. 2E, B as Fig. 1I, C as Fig. 1H, D as Fig. 1J.

Fig. 1b: the higher replicative intermediates for HBV-infected HepG2-NTCP

cells 12 days after the infection, still overexpressed CDC42-CA could be derived from either increased HBV uptake, but also from increased activity of cccDNA (e.g. increased transcription, see increased HBeAg expression in Figure 1e). How do the authors really distinguish between these things in experimental results? Even the little increased amount of cccDNA in presence of constitutively active (CA) CDC42 (and decrease during presence of dominant-negative CDC42) shown in Fig. 1d could be a result of increased or decreased activity of cellular repair mechanism in the nucleus acting on incoming HBV rcDNA after uptake (via whichever HBV route of uptake). This is not clearly shown in the manuscript.

Response:

To address this question, we have performed following experiments.

i) In our study, we used both genetic (CDC42-CA or -DN) and chemical (bradykinin or ML141) approaches to manipulate CDC42 activity in cells. Consistent results were obtained from two approaches. While CDC42-CA and -DN expression are stable in cells, the treatment with bradykinin or ML141 is temporary. We have tested how long the CDC42 activator bradykinin and the inhibitor ML141 are effective. To do it, we have treated HepG2-NTCP cells with bradykinin or ML141 for 10 h, because in infection assays, we pre-treated cells with bradykinin or ML141 for 2 h, then infected cells with HBV for 8 h in the presence of these reagents, thus total treatment time was 10 h. Cells were then washed and collected 1 day post-treatment (dpt), 2 dpt and 3 dpt for GTPase pulldown assay. As shown in Fig. R2-2, the effects of these two reagents primarily lasted for 24 h. On the third day post-treatment, no difference was observed between treated- and untreated-cells, indicating temporary effects of bradykinin and ML141 on CDC42.

Fig. R2-2. HepG2-NTCP cells were treated with bradykinin or ML141 for 10 h. Cells were collected at indicated time points for GTPase pulldown assay. GTP-bound CDC42 was detected by Western blotting. This figure is presented in the revised version as Fig. EV1D.

ii) we have firstly infected HepG2-NTCP cells for 8 h, then treated infected cells with bradykinin or ML141 for 8 h. Cells were collected at 3 dpi for cccDNA analysis and at 5 dpi for pgRNA quantification. No significant difference was observed between treated- and untreated-cells (Fig. R2-3), suggesting that

CDC42 activation or inactivation have no effect on post-entry steps of viral replication.

Fig. R2-3. HepG2-NTCP cells were firstly infected with HBV for 8 h, then treated with bradykinin or ML141. cccDNA was quantified at 3 dpi. pgRNA was prepared and quantified at 5 dpi. cccDNA results are presented in revised Fig. 3A and pgRNA results in revised Fig. 3B.

iii) we have used cell models for HBV replication bypassing entry step, namely HepAD38 cells and HepG2 cells transfected with plasmid containing 1.3-mer HBV genome (pHBV1.3). We have treated HepAD38 cells with bradykinin or ML141 for 2 h in the presence of doxycycline to inhibit pgRNA transcription, then for 8 h in the absence of doxycycline, which is in line with the treatment protocol in infection assays. Seven days later, we have prepared encapsidated viral DNA and performed qPCR on encapsidated viral DNA. No difference was observed between treated- and untreated-cells (Fig. R2-4). We have described this experiment in the Methods. This figure is presented as revised Fig. 3C.

Fig. R2-4. HepAD38 cells were treated with bradykinin or ML141. Encapsidated viral DNA was prepared and quantified by qPCR. This figure is presented as Fig. 3C in the revised version.

iv) we have treated HepG2 cells with bradykinin or ML141, then transfected

cells with pHBV1.3, followed by the treatment with bradykinin or ML141 for 8 h. Encapsidated DNA was prepared at 48 h post-transfection and quantified by qPCR. No difference was observed between treated- and untreated-cells (Fig. R2-5). We have described this experiment in the Methods. This figure is presented as revised Fig. 3D.

Fig. R2-5. HepG2 cells were treated with bradykinin or ML141 for 2 h, then transfected with pHBV1.3. Cells were collected 48 h later and analyzed for encapsidated DNA by qPCR. This figure is presented as revised Fig. 3D.

In conclusion, CDC42 signaling mainly affects HBV entry.

Why did the authors have not used an inducible expression vector of regulated expressing CDC42 and its variants? Using this kind of regulation, the authors could have activated the CDC42 activity during infection, and shut it down after the infection (8 hours) till analysis of HBV replicative intermediates 12 days later (and vice versa)?

Response:

We agree that the inducible expression is a better system to manipulate CDC42 activation/inhibition in cells. After several attempts to establish CDC42 inducible cell lines, we unfortunately failed to obtain stable cell lines. Nevertheless, bradykinin and ML141 are commonly used as CDC42 agonist and inhibitor, respectively (PMID: 7891688; 23382385). ML141 serves as a non-competitive inhibitor of CDC42 with high specificity. Therefore, we employed both CDC42-CA/DN stable cell lines and chemical-induced/inhibited CDC42 as experimental systems to investigate CDC42 functions in HBV infection.

The effects of chemical activators (bradykinin) or inhibitors (ML141) on HBV replicative forms (fig. 1g), secreted virions (fig. 1h) and HBV cccDNA (fig. 1i)

for only 10 hours during infection is a nice approach to differentiate between uptake of HBV and replication of HBV under CDC42, but the controls are missing that it has no effect on HBV replication after 12 days, the day the above mentioned assays on HBV replication have been done.

Response:

The controls have been added for the revision. Fig. R2-2 shows temporary effects of bradykinin and ML141 on CDC42. Our new results show that treatment with bradykinin or ML141 has no effect on after-entry steps of HBV replication (Fig. R2-3, R2-4 and R2-5).

In this regard, the figure 1f is not convincing. An upper band, half visible (but half cut) in the untreated anti-CDC42 control is cut in the Bradykinin- and ML141-treated bands. Is it not there? What is it? The authors should explain and show larger section of the blot here.

Response:

We show here an example of uncropped gel blotted with anti-CDC42 antibody (Fig. R2-6. Top left). We have selected the band in the frame based on the molecular weight of CDC42 (21KD) (Fig. R2-6. Top right). The results of three independent experiments are presented (Fig. R2-6. Top right, bottom left and bottom right). We have replaced Fig. 1F with top right image.

Fig. R2-6. Detection of active CDC42 was assayed by GTP-bound GTPase pulldown and Western blotting. GAPDH was used as loading control. The ratio of the GTP-bound active CDC42 versus GAPDH is presented. The uncropped image (top left) is presented. Three independent experiments are presented.

Fig. 2c-f and Fig. 6b/c: the authors used an anti-preS1 to show viral particles

within cells using immunofluorescence. This statement cannot be made. The preS1-domain is part of the large HBV surface protein, LHBs and is indeed essential for HBV binding of hepatocytes via the NTCP. But the LHBs is also present in HBV subviral particles that are present in large excess both in sera of HBV-infected patients and also in any HBV-containing inoculate from cell culture derived HBV that is not purified by ultracentrifugation via density gradient, e.g. sucrose. Did the authors distinguish between HBV virions and subviral particles in their uptake assay using preS1-specific antibodies in IF? How can the authors argue that the preS1-dependent intracellular signal is derived from viral particles and not subviral particles? According to the material and methods section, the authors used HepAD38 cells that produce HBV but also large excess of subviral particles. The authors did not purify the viral particles in a way to separate virions from subviral particles, but used PEG-precipitation and low-speed centrifugation to pellet the virions-subviral particle precipitates. From the solubilized precipitates, only virions were quantified, but not the amount of subviral particles that contaminate the preparation and will for sure affect the preS1-dependent assay on virus (and subviral particle) uptake. The authors should determine the amount of preS1-containing subviral particles in their preparation and the ratio of virions to subviral particles in the inoculum.

Response:

We agree with you that preS1 staining cannot distinguish infectious particles with subviral particles. The current study is mainly focused on the functions of CDC42 in the regulation of NTCP and viral uptake. We are not intended to specify the entry of infectious particles or subviral particles, since both particles enter into hepatocytes via NTCP binding (PMID: 30952782; 35458456). The internalization experiments aimed to show viral entry, including both virions and subviral particles.

For the revision, HBV particles were also prepared with Heparin HiTrap columns (see the Methods section, chapter HBV production and titration). We compared PEG-purified particles with heparin-purified particles infecting HepG2-NTCP cells in HBV internalization assays and found no significant difference. Therefore, we did not specify the method in HBV stock preparation in the description of each infection experiment. We are aware that both heparin- and PEG-purified HBV particles contain both virions and subviral particles.

Minor:

Spelling:

Line 34: Rab11-dependent, instead of Rab11-depedent.

Response:

We have corrected this spelling in the revised manuscript.

Citation:

Line 185: A citation is needed for the argument that regulation of NTCP mRNA is mainly achieved by bile acids.

Response:

We have added the related citation (PMID:33932583) in the revised manuscript.

Referee #3:

In their manuscript entitled "CDC42 supports hepatitis B virus entry by fostering NTCP plasma membrane location and macropinocytosis-dependent internalization" Shuzi Cui and coworker investigate the impact of CDC42 on HBV life cycle. They describe that CDC42 is a prominent factor affecting HBV entry. In this context they provide evidence that CDC42-dependent macropinocytosis is relevant for HBV entry.

The authors present interesting observations but at the present stage of the manuscript there are still a number of open points, which must be addressed.

Specific points:

Is there a bias in the chosen assay system as GTP-loaded and GDP-loaded CDC42 differ with respect to their localization and due to association with membranes and cytoskeleton components in their solubility?

Response:

We used Active CDC42 Detection Kit (Cell Signaling Technology, #8819). According to the manufacturer's description, the assay system is not affected by the localization or solubility of CDC42-GTP and CDC42-GDP. The assay has been described in detail in the Methods.

The authors should include data describing the impact of CDC42-CA and CDC42-DN on cell polarity.

Response:

To address this question, we have performed immunostaining of MRP2, a bile acid transporter that is expressed on the apical membrane of hepatocytes, in HepG2-NTCP vector, CDC42-CA and CDC42-DN cells. We found that activation of CDC42 increased MRP2 expression, whereas its inactivation decreased MRP2 expression (Fig. R3-1). These results indicate the role of CDC42 in the regulation of cell polarity, and have been presented in revised Fig. EV3E.

Fig. R3-1. MRP2 was stained in HepG2-NTCP vector, CDC42-CA and CDC42-DN cells. MRP2-positive signals are presented in yellow and DAPI in blue. The scale bar = 10 μ m. MRP2 puncta within each cell was quantified. Data are shown as fold changes to the vector group. n = 10 views (40 \times /1.5 air objective). This figure is presented as revised Fig. EV3E.

Information about the mutations leading to CA or DN-activity of CDC42 should be included in the manuscript.

Response:

We described the CDC42-CA and CDC42-DN mutants in the Methods (chapter Cloning), as “Flag-tagged human wild-type (WT), constitutively active (CA, substitution of Gly¹² to Val) and dominant-negative (DN, substitution of Thr¹⁷ to Asp) CDC42 mutant fragments (kindly provided by Prof. Xueliang Zhu, State Key Laboratory of Cell Biology, CAS Center for Excellence in Molecular Cell Science, Shanghai Institute of Biochemistry and Cell Biology, Chinese Academy of Sciences) were cloned into pCDH-CMV-puro.”.

The authors should provide a quantification of fig 1a (GTP-bound CDC42).

Response:

For quantification, the intensity of GTP-bound active CDC42 was quantified by Image J and normalized to that of GAPDH. It has been described in the Methods (chapter Western blot). Three independent experiments were performed (Fig. R3-2). The middle image of Fig. R3-2 is presented as revised Fig. 1A.

Fig. R3-2. Detection of active CDC42 was assayed by GTP-bound GTPase pulldown and Western blotting. GAPDH was used as loading control. The ratio of the GTP-bound active CDC42 versus GAPDH is presented. Three independent experiments are presented.

IF or FACS data should be included investigating the impact of CDC42-CA or CDC42-DN overexpression on the number of HBV positive cells.

Response:

By IF, we have quantified the impact of CDC42-CA or CDC42-DN overexpression on the number of HBV positive cells by staining HBcAg at 7 dpi (Fig. R3-3). CDC42-CA or CDC42-DN overexpression did not affect the percentage of HBcAg positive cells. The intensity of HBcAg increased in HepG2-NTCP CDC42-CA cells and decreased in CDC42-DN cells. These results are in agreement with our findings that CDC42 mainly regulates NTCP cell surface expression, but not cellular susceptibility to HBV infection. These results are presented in revised Fig. 3E.

Fig. R3-3. HBcAg was stained in HepG2-NTCP vector, CDC42-CA and CDC42-DN cells. HBcAg-positive signals are presented in red and DAPI in blue. The scale bar = 10 μ m. HBcAg-positive cell was quantified. Data are shown as fold changes to the vector group. n = 10 views (40 \times /1.5 air objective). This figure is presented as revised Fig. 3E.

line 140: "Together, these data indicate that activation of CDC42 signaling facilitates HBV infection." At this point of the analysis this statement is an overinterpretation. Here the authors can conclude that CDC42 activation favours HBV replication.

Response:

We have modified the text as "Together, these data indicate that CDC42 activation favors HBV infection." in the revised manuscript.

fig2A. It is amazing that a moderate change in the level of GTP versus -GDP-bound CDC42 (fig1) is associated with a comparable on HBV entry effect as Myrcludex. The authors should comment on this.

Response:

The working concentration of Myrcludex B used in the initial study was 100 nM, while previous study (PMID: 32799415) used Myrcludex B at 200 nM. We have tested the inhibition of increasing doses of Myrcludex B on HBV entry and found that 200 nm of Myrcludex B further reduced HBV entry by half compared to its inhibitory effect at 100 nm (Fig. R3-4). This result indicates that a moderate dose of Myrcludex B (100 nM) was used, which may explain its comparable effect to CDC42 inactivation on HBV entry. To be in line with the previous study (PMID: 32799415), we have performed the experiments with 200 nM Myrcludex B in the revised version. Inhibition of HBV entry by the expression of CDC42-DN in HepG2-NTCP cells is 65% of that achieved by 200 nM Myrcludex B (Fig. 2C), while inhibition of HBV entry by the treatment with ML141 is 55% of that achieved by Myrcludex B (Fig. 2D). This result is presented in revised Fig. EV2A.

Fig. R3-4. Inhibitory effects on HBV entry by Myrcludex B in a dose-dependent manner. HepG2-NTCP and HepG2 cells were respectively used as positive and negative controls for HBV entry. This figure is presented as Fig. EV2A in the revised version.

What is the impact of CDC42-CA and CDC42-DN on taurocholate/bile acid uptake?

Response

We have tested taurocholate/bile acid uptake in HepG2-NTCP vector, CDC42-CA and CDC42-DN cells using the Total Bile Acid (TBA) kit (E003-2-1, Nanjing, China). The results show that activation of CDC42 significantly enhanced bile acids uptake, in contrast to the negative effects on bile acids uptake when CDC42 is inactivated (Fig. R3-5). These findings are consistent with the observations that CDC42 markedly regulates NTCP membrane expression. These data are presented as Fig. 4E in the revised version.

Fig. R3-5. Taurocholate (TC) uptake assay in HepG2-NTCP vector, CDC42-CA and CDC42-DN cells by ELISA. Data are shown as fold changes to the vector group. This figure is presented as Fig. 4E in the revised version.

The authors provide evidence that CDC42-CA led to an enhanced localization of NTCP on the plasma membrane. In differentiated cells NTCP-localization

shows polarity-is this observed in CDC42-CA overexpressing cells?

Response:

We did not observe polarity-dependent NTCP localization in HepG2-NTCP CDC42-CA cells.

To address this question, we have quantified fluorescence intensity of NTCP-EGFP on the basal and apical surfaces of HepG2-NTCP-EGFP vector, CDC42-CA and CDC42-DN cells and calculated the ratio of basal versus apical fluorescence intensity in each cell type. We found no significant difference (Fig. R3-6). Thus, CDC42 signaling regulates NTCP membrane expression but not its distribution relative to hepatocyte polarity. We have presented this result in Fig. EV3F in the revised version. In the Discussion section, we wrote that “we observed that NTCP is almost equally expressed on the basal and apical membranes in HepG2-NTCP cells. The reason why NTCP lacks the polarity pattern seen in the liver is not known. We speculate that this patterning may require other cell components and microenvironmental cues in the liver but absent in the cell culture system.”

Fig. R3-6. NTCP-EGFP fluorescence was quantified at the basal and apical surfaces of HepG2-NTCP vector, CDC42-CA and CDC42-DN cells. The values are presented as the ratio of EGFP intensity at the basal surface to that at the apical surface. n = 10 views (40×/1.5 air objective). This figure is presented as revised Fig. EV3F.

HBV-infection is characterized by "receptor down modulation " in infected cells. Is this affected by CDC42-activation?

Response:

We have performed new experiments to show the receptor down modulation by HBV infection in the context of CDC42 activation or inactivation. HepG2 NTCP vector, CDC42 CA and CDC 42DN cells were infected with HBV and cells were collected at 7 dpi. Total NTCP expression in infected and non-infected cells was revealed by Western blot. Consistent with previous report (PMID: 11119583), HBV infection resulted in slightly decrease of NTCP

level in HepG2 NTCP vector, CDC42 CA and CDC 42DN cells compared to their non-infected counterparts, but no significant difference was observed among these infected cells (Fig. R3-7). This result is consistent with our findings that CDC42 does not affect total NTCP expression in HepG2-NTCP cells. This figure is presented as revised Fig. EV3G.

Fig. R3-7. Total NTCP expression detected by Western blot in indicated cell lines infected or uninfected with HBV. Cells were collected at 7 dpi. GAPDH was used as loading control. The ratio of NTCP versus GAPDH is presented. This figure is presented as Fig. EV3G in the revised version.

The colocalization of CDC42 with Rab11 should be characterized in more detail with respect to the subcellular compartment(s) in which both proteins colocalize.

Response:

Both Rab11 and CDC42 have been reported enriched in the recycling endosome (PMID: 10811830; 17704769). However, immunoprecipitation assays show absence of interaction between Rab11 and CDC42 (Fig. 5H,I). No colocalization of CDC42 with Rab11 was found in this study.

What is the impact of CDC42-CA/DN in the experimental setting described here on membrane composition, lipid raft formation, especially cholesterol content and cholesterol uptake?

Response:

The active form of CDC42 has been shown to induce membrane ruffling (PMID: 9348306). It is possible that cells expressing CDC42-CA/DN exhibit altered membrane composition or lipid rafts. Cholesterol plays a role in the HBV internalization pathway (PMID: 32799415). The relationship between CDC42 signaling and cholesterol content on the plasma membrane can be an independent subject for studying cellular factors on HBV entry.

Minor points:

There are some typing errors and style and grammar could be slightly modified.

Response:

We have carefully checked typing errors and grammar in the revised manuscript.

Referee #4:

The manuscript presents compelling evidence that the level of active CDC42 in hepatocytes positively correlates with the entry capacity of the hepatitis B virus (HBV). This occurs via the promotion of the transport of the viral receptor, sodium taurocholate co-transporting polypeptide (NTCP), to the plasma membrane through a CDC42-driven, Rab11-dependent recycling endosomal pathway. Additionally, it demonstrates that macropinocytosis is essential for viral uptake.

General Comments:

The experiments are generally well designed, but some controls are needed and further experiments should be included to confirm the claims.

It is widely recognized that high multiplicities of infection (MOIs) of HBV are required in cell culture to achieve infection of a small number of cells with minimal amounts of covalently closed circular DNA (cccDNA), indicating that only a few particles reach the nucleus and convert relaxed circular DNA (rcDNA) to cccDNA. This suggests that most HBV particles taken up by cells do not lead to actual infection. It would be interesting to consider which entry pathways result in uptake and which lead to real infection. Observing that HepG2 cells without NTCP expression cannot be infected by HBV at all, likely due to reduced uptake and membrane fusion, the observed HBV uptake in these cells requires further explanation.

Response:

HepG2 cells were employed as negative control for HBV infection (Fig. R4-1). We believe that the weak signals in HepG2 cells were background, because treatment with bradykinin or ML141 did not reduce the signals.

Fig. R4-1. Quantification of HBV entry level with HepG2 cells as the negative control.

The question of which entry pathways result in uptake and which lead to real infection is very important. We discussed this point in the Discussion section. We wrote that “We show that inhibition of CME or macropinocytosis results in reduced internalization of HBV and decreased levels of HBeAg, indicating that

both pathways can lead to productive viral infection. These data indicate that HBV involves various cellular endocytic pathways to ensure the success of its infection. Which pathway is more efficient to establish HBV infection? Does HBV randomly utilize the endocytic pathways for its entry, or does it selectively use specific ones? What are the factors that influence the selection of the entry pathway by the virus? These questions remain unanswered. It appears that successfully complete endosomal escape is a key step in HBV infection. Both clathrin-coated vesicles (CCVs) derived from CME and macropinosomes generated via macropinocytosis undergo vesicular fusion to form late endosomes and proceed through maturation (PMID: 30967002). Further studies on the processes of CCVs and macropinosomes maturation may provide valuable insights into the detailed mechanisms involved in HBV entry.”

Major Comments:

Figure 1:

What is the transduction rate of the HepG2 cells? A Flag staining of the cells would be beneficial since the CDC42 constructs express a Flag-tag. The authors could then show flag staining in parallel to HBV-PreS1 staining of positively and not-transfected cells next to each other.

Response:

HepG2-NTCP CDC42-CA/Flag and CDC42-DN/Flag cell lines are stable cell lines (Fig. R4-2). The generation of these cell lines is described in the Methods (chapters Cloning and Generation of Cell Lines).

Fig. R4-2. Immunofluorescence images of HepG2-NTCP vector, CDC42-CA and CDC42-DN cells stained with anti-Flag antibody. Nucleus were stained with DAPI.

Southern blot should be labeled with each individual band correctly rather than "replication forms".

Response:

Four Southern blots are presented in this study (Fig. R4-3). We observed that the position of each band varied between experiments. We reasoned that the factors such as DNA conformation, enzyme digestion efficiency, and gel electrophoresis conditions might affect the migration of different forms of HBV DNA in individual experiment. To accurately label each band, additional experiments using specific probes or specific enzymatic treatments are

required. In the current study, Southern blot analysis was used to demonstrate changes in the overall levels of HBV replication, while changes in specific

Fig 1

Fig 5

Fig 7

forms were quantified by qPCR with specific primers. Consequently, we chose to label them as "replication forms" in Southern blots.

Fig. R4-3. Southern blots in the main figures.

Explanation is needed for the significant difference between the CA and DN controls in Fig. 1b, while Fig. 1d only shows a three-fold difference.

Response:

Fig. 1b shows the levels of encapsidated HBV DNA in HepG2-NTCP CDC42-CA and CDC42-DN cells by Southern blot analysis. Cells were collected at 14 dpi. Fig. 1d shows cccDNA levels by qPCR analysis. Cells were collected at 3 dpi. The differences between CA and DN in these two figures mainly due to the different post-infection time points and the use of different assays. The accumulation of encapsidated DNA is a time-dependent process in HepG2-NTCP/HBV infection system, allowing to observe long-term effects (14 dpi) of CDC42 activation/inactivation on HBV replication. The analysis of cccDNA in samples collected at 3 dpi is intended to examine input viral particles.

Figure 2:

While claiming the quantification of HBV entry, it is essential to discriminate between virus taken up and virus bound to the membrane. How do the authors differentiated this just by using PCR?

Response:

These assays have been performed based on the previous publications (PMID: 32799415; 24375637; 34728272). For the attachment assay, HepG2-NTCP cells were incubated with HBV at 4°C for 3 h and then collected after washing out free viruses. For the internalization assay, cells were firstly incubated with HBV at 4°C for 3 h. Culture medium was changed. Cells were cultured at 37°C for further 8h. Cells were then trypsinized and extensively washed to remove cell-associated non-internalized viral particles.

For the revision, we have used another method for internalization assay. Cells were inoculated with heparin-purified HBV for 1 h at 4°C and then transferred to 37°C for 2 h, 4 h or 6 h. Cells were subsequently trypsinized and extensively washed prior to quantifying the cellular HBV DNA.

The attachment assay and internalization assay have been described in detail in the Methods.

In Fig. 2c, the fluorescence image of anti-PreS1 labeled particles shows PreS1 in the nucleus, which is unexpected since PreS1 is a membrane protein. Confocal microscopy should be used to show PreS1 staining in the endosomes and not the nucleus.

Response:

Fig. 2c (Fig. 2F in the revised version) is the Z-axis projection of scanning Confocal microscopy. The PreS1 positive signal is not located within nucleus; instead, it is present in the cytoplasmic spaces above or below nucleus when viewed from a 3D perspective.

A heparin control should be included since it would prevent PreS1 attachment and thus any particle uptake.

Response:

We have added heparin control in attachment and internalization experiments and presented the results in revised Figs. 2A-D (Fig. R4-4). Heparin treatment has been added in the Methods.

Fig. R4-4. HBV attachment and internalization assays were performed in HepG2-NTCP vector, CDC42-CA and CDC42-DN cells. Heparin was used as control reagent to block HBV attachment in HepG2-NTCP cells. These figures are presented as revised Figs. 2A-D.

In Fig. 2g, HepG2 cells do not take up any HBV compared to HepG2-NTCP cells. This result needs clarification, given that MyrB-treated HepG2-NTCP should yield similar results in HBV uptake, which seems to contradict to Fig. 2b.

Response:

The working concentration of Myrcludex B used in the initial study is 100 nM, while previous study (PMID: 32799415) used Myrcludex B at 200 nM. We have tested increasing doses of Myrcludex B on HBV entry and found that 200 nm of Myrcludex B further reduced HBV entry by half compared to its inhibitory effect at 100 nm (Fig. R4-5). This result indicates that a moderate dose of Myrcludex B (100 nM) was used in this study, which may explain its incomplete inhibition of HBV entry. To be in line with the previous study (PMID: 32799415), we have performed the experiments with 200 nM Myrcludex B in the revised version. Inhibition of HBV entry by the expression of CDC42-DN in HepG2-NTCP cells is 65% of that achieved by 200 nM Myrcludex B (Fig. 2C), while inhibition of HBV entry by the treatment with ML141 is 55% of that achieved by Myrcludex B (Fig. 2D). This result is presented in revised Fig.

EV2A.

Fig. R4-5. Inhibitory effects on HBV entry by Myrcludex B in a dose-dependent manner. HepG2-NTCP and HepG2 cells were respectively used as positive and negative controls for HBV entry. This figure is presented as Fig. EV2A in the revised version.

Figure 3:

The quantification of the Western blot in the figure is unclear. It should be clarified to which blot the numbers 0.46, 0.67, etc., correspond, as they do not seem to correlate with PM-NTCP or PM TfR.

Response:

For quantification, intensities of bands of interests were calculated by Image J, subtracted the background signaling and normalized to loading controls. The method has been described in the Methods (chapter Western blot).

The plasma membrane protein transferrin receptor (PM-TfR) and GAPDH were used as loading control of plasma membrane proteins and total proteins, respectively. The ratio of the PM-NTCP versus PM-TfR and total NTCP versus GAPDH are presented. It is indicated in the figure legends.

Overall, the claim that CDC42 and Rab11 inhibit HBV uptake is supported by showing HBV infection parameters, Southern blot, and qPCR. Including experiments for up or down-regulation using plasmid transfection with an HBV1.3 plasmid would strengthen the study. Since HBV cannot spread in cell culture, all experiments should show no effect on the HBV parameters for HBV markers.

Response:

We agree with you. For the revision, we have used two cell culture systems to bypass the entry step. First, we have treated HepAD38 cells in the absence of doxycycline with bradykinin or ML141. The level of encapsidated HBV DNA was measured by qPCR. No significant difference was observed (Fig. R4-6, left panel). The result is presented in revised Fig. 3C. Second, we have used an HBV1.3 plasmid to transfect HepG2 cells and treated cells with bradykinin or ML141. Cells were collected to prepare encapsidated HBV DNA. Analysis by qPCR showed no significant changes after treatment (Fig. R4-6, right panel). This figure is presented as revised Fig. 3D.

Fig. R4-6. Encapsidated HBV DNA was measured by qPCR from HepAD38 cells and HepG2 cells transfected with 1.3-mer HBV plasmid.

Another potential experiment could be the knockdown after HBV infection at 1 dpi. Since this should not show any reduction in infection parameters. Same could be done by using HBV expressing cell lines (HepG2-HBV, HepAD38).

Response:

We have replaced the knockdown of CDC42 by treating cells with ML141. We have infected HepG2-NTCP cells for 8 h, then treated infected cells with bradykinin or ML141 for 8 h. Cells were collected at 3 dpi for cccDNA analysis and at 5 dpi for pgRNA quantification. No significant difference was observed between treated- and untreated-cells (Fig. R4-7). cccDNA results are presented in revised Fig. 3A. pgRNA results are presented in revised Fig. 3B.

Fig. R4-7. HepG2-NTCP cells were first infected with HBV for 8 h, then treated with bradykinin or ML141. cccDNA was quantified at 3 dpi. pgRNA was prepared and quantified at 5 dpi. cccDNA results are presented in revised Fig. 3A and pgRNA results in revised Fig. 3B.

Dear Yaming,

Thank you for submitting your revised manuscript. It has now been seen by two of the original referees. My apologies for the delay in getting back to you, which was due to the delay in receiving referee reports.

As you will see, referees find that the study is significantly improved during revision and recommend publication. However, referee #2 has remaining minor outstanding concerns. Please address them and provide a point-by-point response. Please let me know if you would like to discuss any of the points further.

Moreover, the editorial points below need to be addressed before I can accept the manuscript.

- Please remove the Author contributions section from the manuscript text.
- Please fill out and include an author checklist as listed in our online guidelines (<https://www.embopress.org/page/journal/14693178/authorguide>)
- The funding information needs to be complete in both the manuscript text and our manuscript tracking system. We note that the following is missing from the manuscript tracking system: 92354301, 92054104 for National Natural Science Foundation of China; R&D Program of Guangzhou National Laboratory (GZNL2023A03004); Natural Science Foundation of Shanghai (23ZR1470900); Key Research and Development Program, Ministry of Science and Technology of China (2022YFC2303502).
- We note that main figures provided in one PDF and EV figures in the other PDF. The main and EV figures need to be provided as individual production quality Figure files (in TIFF, PDF or EPS format).
- Please remove the following sentences from the Data Availability section "All data that support the findings of the study are available in the manuscript and the supplementary information files. Source data are provided with this paper."
- Please remove the Reagents & Tools table from the manuscript file and submit it as a separate word file.
- We note the following regarding the source data: a separate zip file needs to be uploaded for each figure. Moreover, the source data file uploaded for the PM-NTCP blot of Figure 5J seems blank. Please double check.
- The manuscript sections should be in the following order: Title page - Abstract & Keywords - Introduction - Results - Discussion - Methods - Data Availability - Acknowledgments - Disclosure Statement & Competing Interests - References - Figure Legends - (Main Tables with legends if applicable) - Expanded View Figure Legends.
- Our production/data editors have asked you to clarify several points in the figure legends - Figure Legends (main + EV):
 - o Please note that the exact p values are not provided in the legends of figures 1C, D, E, H-J; 2A, B, C, D, R, G, I, J; 3E, 4E, 5C, E, G; 6E, F, G, H; 7A, C, E, F, G, H, I, J; EV1 E-G; EV2 A-C; EV3 D-E; EV4 C-E; EV5 B, D, H, I, J, L, M.
 - o Please note that the white borders are not defined in the legend of figures 2F, H. This needs to be rectified.
- Papers published in EMBO Reports include a 'synopsis' and 'bullet points' to further enhance discoverability. Both are displayed on the html version of the paper and are freely accessible to all readers. The synopsis includes a short standfirst summarizing the study in 1 or 2 sentences (max 35 words) that summarize the paper and are provided by the authors and streamlined by the handling editor. I would therefore ask you to include your synopsis blurb and 3-5 bullet points listing the key experimental findings.
- In addition, please provide an image for the synopsis. This image should provide a rapid overview of the question addressed in the study but still needs to be kept fairly modest since the image size cannot exceed 550 (width) x 300-600 (height) pixels.

Thank you again for giving us to consider your manuscript for EMBO Reports, I look forward to your minor revision.

Kind regards,

Deniz

--

Deniz Senyilmaz Tiebe, PhD
Senior Scientific Editor
EMBO Reports

Referee #3:

The authors sufficiently addressed all points raised by the reviewer

Referee #4:

Reviewer Response

We would like to thank Cui and colleagues for their thorough revision of the manuscript. The authors have addressed the majority of the concerns raised by myself and the other reviewers in a satisfactory manner. Below, I highlight a few remaining

points which were originally raised by the different respective reviewer, that I believe could still benefit from further clarification or minor revision:

Reviewer 1:

The authors have emphasized the use of a new protocol for virus purification using a heparin column. In this context, they could cite this protocol publication (PMID: 34452368) if they used this method. Additionally, it would be helpful if the authors clearly indicate which experiments were performed with which virus stock and whether additional PEG was used in the infection medium for each case.

Reviewer 3:

The authors have added an experiment demonstrating the dose-dependency of MyrB on viral uptake. It appears that uptake is reduced by approximately 50-60% at 100 nM, while infection is known to be reduced by more than 95% at the same concentration. Do the authors see a contradiction here? Perhaps they could briefly comment on this point without the need for further experiments or manuscript changes.

Reviewer 4:

While the manuscript describes how the cell lines were generated, it lacks detail regarding the antibiotic resistance used for selection and the resulting positive rate. A brief addition to the Materials and Methods section would be sufficient to clarify this. Furthermore, I do not fully agree with the authors' response regarding the Southern blot labeling. As correctly stated in their reply, band migration can vary between experiments due to different factors. However, the different forms of the DNA should migrate consistently within the same experiment, allowing for more precise labeling of the bands without requiring additional experiments. For reference, see PMID: 23821267.

Dear Deniz,

Thank you very much for handling our manuscript (MS# EMBOR-2024-60966V2) entitled "CDC42 supports hepatitis B virus entry by fostering NTCP to plasma membrane and macropinocytosis". We have revised the manuscript according to the reviewer #4's comments.

We highlighted all these modifications with red color in the revised marked version of the manuscript, and resubmitted it along with a clean version of the revised manuscript. With these improvements, we hope that the current version of our manuscript meets the journal's standards for publication.

Sincerely,
Yaming Jiu, PhD, Professor,
Shanghai Institute of Immunity and Infection,
Chinese Academy of Sciences

- Please remove the Author contributions section from the manuscript text.

Response:

We have removed the Author contributions section.

- Please fill out and include an author checklist as listed in our online guidelines (<https://www.embopress.org/page/journal/14693178/authorguide>)

Response:

We have filled out the author checklist.

- The funding information needs to be complete in both the manuscript text and our manuscript tracking system. We note that the following is missing from the manuscript tracking system: 92354301, 92054104 for National Natural Science Foundation of China; R&D Program of Guangzhou National Laboratory (GZNL2023A03004); Natural Science Foundation of Shanghai (23ZR1470900); Key Research and Development Program, Ministry of Science and Technology of China (2022YFC2303502).

Response:

We have added all the funding information in the manuscript tracking system.

- We note that main figures provided in one PDF and EV figures in the other PDF. The main and EV figures need to be provided as individual production quality Figure files (in TIFF, PDF or EPS format).

Response:

We have provided figures and EV figures as individual PDFs.

- Please remove the following sentences from the Data Availability section "All data that support the findings of the study are available in the manuscript and the supplementary information files. Source data are provided with this paper."

Response:

We have removed the sentences.

- Please remove the Reagents & Tools table from the manuscript file and submit it as a separate word file.

Response:

We have removed the Reagents & Tools table from the manuscript file and submitted it as a separate word file.

- We note the following regarding the source data: a separate zip file needs to be uploaded for each figure. Moreover, the source data file uploaded for the PM-NTCP blot of Figure 5J seems blank. Please double check.

Response:

We have checked and upload the source data as separate zip file.

- The manuscript sections should be in the following order: Title page - Abstract & Keywords - Introduction - Results - Discussion - Methods - Data Availability - Acknowledgments - Disclosure Statement & Competing Interests - References - Figure Legends - (Main Tables with legends if applicable) - Expanded View Figure Legends.

Response:

We have reordered sections as you suggested.

- Our production/data editors have asked you to clarify several points in the figure legends - Figure Legends (main + EV):

- o Please note that the exact p values are not provided in the legends of figures 1C, D, E, H-J; 2A, B, C, D, R, G, I, J; 3E, 4E, 5C, E, G; 6E, F, G, H; 7A, C, E, F, G, H, I, J; EV1 E-G; EV2 A-C; EV3 D-E; EV4 C-E; EV5 B, D, H, I, J, L, M.

- o Please note that the white borders are not defined in the legend of figures 2F, H. This needs to be rectified.

Response:

We have added the exact p values in the legends.

We have defined the white borders in the legend of figures 2F.

- Papers published in EMBO Reports include a 'synopsis' and 'bullet points' to further enhance discoverability. Both are displayed on the html version of the paper and are freely accessible to all readers. The synopsis includes a short standfirst summarizing the study in 1 or 2 sentences (max 35 words) that summarize the paper and are provided by the authors and streamlined by the handling editor. I would therefore ask you to include your synopsis blurb and 3-5 bullet points listing the key experimental findings.

Response:

Synopsis blurb is “CDC42 regulates HBV entry through two mechanisms: it promotes the recruitment of NTCP receptor to the cell surface and enhances the macropinocytosis-mediated internalization of NTCP-bound virions, thereby facilitating HBV infection”.

Bullet points are:

- CDC42 promotes NTCP translocation to the plasma membrane via Rab11-related recycling endosomes
- Macropinocytosis is an important pathway for HBV internalization
- Macropinocytosis-mediated, but not clathrin-mediated, HBV internalization depends on CDC42

- In addition, please provide an image for the synopsis. This image should provide a rapid overview of the question addressed in the study but still needs to be kept fairly modest since the image size cannot exceed 550 (width) x 300-600 (height) pixels.

Response:

Referee #3:

The authors sufficiently addressed all points raised by the reviewer.

Response:

We thank you very much for your positive feedback on our manuscript.

Referee #4:

Reviewer Response

We would like to thank Cui and colleagues for their thorough revision of the manuscript. The authors have addressed the majority of the concerns raised by myself and the other reviewers in a satisfactory manner. Below, I highlight a few remaining points which were originally raised by the different respective reviewer, that I believe could still benefit from further clarification or minor revision:

Reviewer 1:

The authors have emphasized the use of a new protocol for virus purification using a heparin column. In this context, they could cite this protocol publication (PMID: 34452368) if they used this method. Additionally, it would be helpful if the authors clearly indicate which experiments were performed with which virus stock and whether additional PEG was used in the infection medium for each case.

Response:

The publication PMID: 34452368 has been cited in the chapter HBV Production and Titration in the Methods. We specified in this chapter which

infection experiment was performed with heparin-prepared or PEG-prepared HBV stocks.

Reviewer 3:

The authors have added an experiment demonstrating the dose-dependency of MyrB on viral uptake. It appears that uptake is reduced by approximately 50-60% at 100 nM, while infection is known to be reduced by more than 95% at the same concentration. Do the authors see a contradiction here? Perhaps they could briefly comment on this point without the need for further experiments or manuscript changes.

Response:

In the current study, we used MyrB exclusively in entry experiment and did not examine viral replication at post-entry steps following MyrB treatment throughout our study. We showed that MyrB can reduce viral entry by approximately 60% at 100 nM. We do not know the extent of reduction of cccDNA, encapsidated HBV DNA, pgRNA or HBeAg levels in our experimental systems.

Reviewer 4:

While the manuscript describes how the cell lines were generated, it lacks detail regarding the antibiotic resistance used for selection and the resulting positive rate. A brief addition to the Materials and Methods section would be sufficient to clarify this.

Response:

The generation and antibody selection of HepG2-NTCP and HepG2-NTCP-EGFP cell lines were described in the chapter Cell Culture in the Methods. The generation and antibody selection of the cell lines expressing CDC42 mutants or Rab11 were described in the chapters Cloning and Generation of Cell Lines in the Methods.

Furthermore, I do not fully agree with the authors' response regarding the Southern blot labeling. As correctly stated in their reply, band migration can vary between experiments due to different factors. However, the different forms of the DNA should migrate consistently within the same experiment, allowing for more precise labeling of the bands without requiring additional experiments. For reference, see PMID: 23821267.

Response:

We have added PMID: 23821267 in the References. We have labeled the bands in all Southern blots similar as in PMID: 23821267.

Prof. Yaming Jiu
Shanghai Institute of Immunity and Infection, Chinese Academy of Sciences
Key Laboratory of Molecular Virology and Immunology
Life Science Research Building 320 Yueyang Road, Xuhui District, Shanghai, China
Shanghai 200031
China

Dear Yaming,

Thank you for submitting your revised manuscript. I have now looked at everything and all is fine. Therefore, I am very pleased to accept your manuscript for publication in EMBO Reports.

Congratulations on a nice work!

Kind regards,

Deniz

--

Deniz Senyilmaz Tiebe, PhD
Senior Scientific Editor
EMBO Reports
